# Changes in Air Quality during the Period of COVID-19 in China

**DOI:** 10.3390/ijerph192316119

**Published:** 2022-12-02

**Authors:** Xin Xu, Shupei Huang, Feng An, Ze Wang

**Affiliations:** 1School of Economics and Management, China University of Geosciences (Beijing), Beijing 100083, China; 2Key Laboratory of Carrying Capacity Assessment for Resource and Environment, Ministry of Natural Resources, Beijing 100083, China; 3School of Economics and Management, University of Science and Technology Beijing, Beijing 100083, China; 4International Academic Center of Complex Systems, Beijing Normal University, Zhuhai 519087, China

**Keywords:** air quality, predictability, COVID-19 pandemic, ensemble empirical mode decomposition, recursive quantitative analysis

## Abstract

This paper revisits the heterogeneous impacts of COVID-19 on air quality. For different types of Chinese cities, we analyzed the different degrees of improvement in the concentrations of six air pollutants (PM2.5, PM10, SO_2_, NO_2_, CO, and O_3_) during COVID-19 by analyzing the predictivity of air quality. Specifically, we divided the sample into three groups: cities with severe outbreaks, cities with a few confirmed cases, and cities with secondary outbreaks. Ensemble empirical mode decomposition (EEMD), recursive plots (RPs), and recursive quantitative analysis (RQA) were used to analyze these heterogeneous impacts and the predictivity of air quality. The empirical results indicated the following: (1) COVID-19 did not necessarily improve air quality due to factors such as the rebound effect of consumption, and its impacts on air quality were short-lived. After the initial outbreak, NO_2_, CO, and PM2.5 emissions declined for the first 1–3 months. (2) For the cities with severe epidemics, air quality was improved, but for the cities with second outbreaks, air quality was first enhanced and then deteriorated. For the cities with few confirmed cases, air quality first deteriorated and then improved. (3) COVID-19 changed the stability of the air quality sequence. The predictability of the air quality index (AQI) declined in cities with serious epidemic situations and secondary outbreaks, but for the cities with a few confirmed cases, the AQI achieved a stable state sooner. The conclusions may facilitate the analysis of differences in air quality evolution characteristics and fluctuations before and after outbreaks from a quantitative perspective.

## 1. Introduction

The outbreak of COVID-19 had significant impacts on the lives of people and the operation of the economy [1,2,3,4,5,6]. Governments implemented various measures to prevent its spread, such as controls on travel [7], case isolation [8], and home quarantine [9,10,11,12]. These policies led to significant reductions in travel and industrial products [13] and affected air quality. However, these impacts on air quality could be heterogeneous for Chinese cities with different levels of epidemic severity. For the cities with severe outbreaks, the outbreak and strict lockdown measures could significantly improve air quality, but for cities less severely affected, the lifting of initial lockdown measures could lead to a rebound in consumption and production, worsening air quality. Furthermore, for some cities that experienced a second outbreak of COVID-19, their change in air quality presented a more complex picture. At present, there is no detailed empirical research on these differences. Beyond that, the economy of Chinese cities is highly connected. The outbreak and lockdown measures in one area may affect production activities in the surrounding areas and thus affect the air quality in the surrounding areas. Hence, this paper aims to evaluate and compare the heterogeneous impacts of different levels of epidemic outbreaks on air quality and the changes in the predictability and stability of air quality during this period. This could provide a new perspective from which to study changes in air quality during the outbreak resulting from these regulations [14,15,16] and the influence of measures for curbing the spread of COVID-19 on these changes and the improvement of environmental quality [17,18,19].

Most current studies believe COVID-19 improved air quality [20,21,22]. As one of the countries with the strictest control over the epidemic, China has received more attention [23]. An existing study analyzed changes in air quality in Shanghai, Wuhan, and Tangshan before and after the outbreak, finding that COVID-19 control measures improved air quality [24]. When variables such as temperature were included, the AQI was 12.2 percent lower in cities with lockdown policies [25]. The difference-in-differences (DID) model was used to estimate the impact of the COVID-19 pandemic on air quality, and a significant improvement in air quality during the COVID-19 pandemic was confirmed [26]. Another study used a quasi-difference-in-differences analysis to investigate the influence of lockdown measures during the COVID-19 outbreak on air pollution in 357 Chinese cities, finding that air quality improved significantly during the first outbreak [27]. The literature has also pointed out that energy use and carbon dioxide emissions have decreased during the COVID-19 pandemic, and this negative influence was more significant than that observed during previous economic downturns [18]. It has also been found that the reduction of emissions has varied according to the measures taken by each country to restrict movement. On the basis of indicators of transportation and power generation, air quality was significantly improved in China due to the pandemic in 2020 compared to that during the same period in 2019 [28].

However, as mentioned above, COVID-19 could have heterogeneous impacts on air quality and its predictability and stability that have not been considered in previous literature [28]. The autocorrelation of the air quality sequence in these cities makes it predictable. Under normal conditions, the time series of air quality has strong predictability, but the epidemic outbreak could cause significant effects and new information, breaking the original sequence pattern, and the predictability is reduced. In addition, the predictability of the air quality sequence was more affected by the two outbreaks. Therefore, we hold that the analysis of predictability can help to depict the heterogeneous effects of COVID-19 on air quality. This can be visually represented by RP diagrams. On the basis of this method, some quantitative indicators can be used to further quantify changes in predictability.

Beyond that, there are complex industrial and administrative relationships among Chinese cities. Economic development processes will be linked through various channels, COVID-19 lockdown measures will spill over, and air quality change patterns will be more complex. The effects of the epidemic on air quality are also time dependent. With the gradual increase in confirmed cases, residents’ attention to the epidemic and their travel demands has been further affected, which has also influenced air quality. Even if restrictions on movement are lifted, behavioral inertia and worry regarding the epidemic will alter the lives of residents and travel patterns over a certain period. Due to the rapid economic recovery in some regions, emissions will rebound, and air quality may be decreased. The above factors, including repeated outbreaks of COVID-19, sudden lockdown measures, and gradual resumption of production and work, can cause short-term perturbations in air quality data, which may lead to deviations in studies on the long-term impact of COVID-19 on air quality. Therefore, it is also important to discuss the long-term change in air quality without the interference of short-term redundant information. 

To be specific, this paper considered three types of Chinese cities under different epidemic levels (severe epidemics, less severe epidemics, and secondary outbreaks). We first focus on the evolution of air quality during surges and stable epidemic periods in these cities by adopting the EEMD method. The empirical process is as follows: (1) On the basis of data availability, this paper analyzed and compared trends in six pollutants (PM2.5, PM10, SO_2_, NO_2_, CO, and O_3_) in 348 cities between 2018–2019 and 2019–2020. (2) We selected 12 cities that experienced severe epidemic conditions in Hubei Province, 24 cities with a few confirmed cases from other provinces, and 5 cities with secondary outbreaks. The AQI was used to measure air quality from 1 January 2018 to 27 December 2020. This article adopted this method by mainly considering the following aspects: the characteristics of air pollutant data, the purpose of the research, and the advantages of the methods. Specifically, the data form of air pollutants was non-stationary daily time series data, and its sample size was greater than 300 for each year, meeting the data requirements of the EEMD method, which was suitable for non-stationary signal analysis. This paper aimed to analyze the long-term change characteristics of air pollutants, and EEMD can effectively extract these characteristics. The results of EMD may be aliasing, resulting in the unclear distinction of components in different time scales. The EEMD used in this paper could make up for the shortcoming of EMD and provide more accurate long-timescale information about the air quality.

Then, this paper adopted recurrence plots (RPs) and recurrence quantification analysis (RQA) to compare the characteristics of fluctuations in AQI before and after outbreaks, which could be used to describe the effect of the COVID-19 pandemic on the predictability of urban air quality. Specifically, in this paper, we first selected the three classes of cities mentioned previously: cities experiencing severe epidemics, cities with mild epidemic conditions, and cities experiencing secondary outbreaks. The recurrence plots of the mean values of AQI in these cities were reported to discuss the characteristics of fluctuation in AQI in these different kinds of cities. The recurrence plots and recurrence quantification analysis for the mean AQI values in 2018, 2019, and 2020 are listed for comparison. The empirical results prove that the impact of epidemic outbreaks on air quality was different in different cities. Air quality was significantly improved in severely affected areas, while the improvement effect was insignificant in less severely affected areas. In cities with severe COVID-19, new information generated by outbreaks decreased the predictability of air quality.

Compared with the existing literature, this paper has two main innovations. First, differing from the existing literature, which discussed the different levels of air quality before and after the outbreak and investigated the impact of COVID-19 lockdowns on air quality, this paper focused on the evolution of air quality during surges and stable periods of the epidemic for areas experiencing severe and mild epidemic conditions by adopting ensemble empirical mode decomposition (EEMD). We confirmed that the long-term impact of the epidemic on air quality was complex and heterogeneous. Whether the impact of COVID-19 on AQI will be weakened with future outbreaks and stabilization of the epidemic also attracted our interest. Hence, we further selected five cities, namely, Harbin, Dalian, Urumqi, Chengdu, and Beijing, to investigate this point and compare air quality changes before and after secondary outbreaks. The epidemic was controlled after the outbreak in these cities, but new confirmed cases were detected again a few months after the initial confirmed cases no longer increased, and the blockade measures in these cities were further strengthened.

Second, to our knowledge, whether the COVID-19 pandemic could affect the predictability of urban air quality in different cities has not been addressed in previous literature. Therefore, this paper adopted recurrence plots (RPs) and recurrence quantification analysis (RQA) to compare the characteristics of fluctuations in AQI before and after outbreaks, which could be used to describe the effect of the COVID-19 pandemic on the predictability of urban air quality visually and quantitatively. It was found that air quality changes were more dramatic for cities under severe pandemics, and predictability was significantly reduced. Comparatively, the air quality sequence of cities with less severe epidemics was more stable. For the cities with the second outbreak, the negative impact of the second outbreak on the stability of the sequence was weakened. This may help to predict urban air quality accurately, facilitate the analysis of differences in air quality evolution characteristics and fluctuations before and after outbreaks from a quantitative perspective, and formulate air control measures in advance.

The remainder of the paper is organized as follows. Section 2 presents the methodology and data; the empirical results and analysis are reported in Section 3; Section 4 discusses the findings; and the conclusions and policy implications are presented in Section 5.

## 2. Methods and Data

This paper first used a different method to calculate the change in air pollutants before and after the epidemic outbreak and then used the signal decomposition method to decompose its components. The intrinsic mode function (IMF) calculated by EEMD represents different frequency components in air quality variation. IMF is a kind of time series data, meeting two conditions: (1) In the whole data series, the number of extreme points $\left(N_{e}\right)$ and the number of zeros crossing $\left(N_{z}\right)$ must meet the following relations: $\left(N_{z}-1\right)\leq N_{e}\leq \left(N_{z}+1\right)$. (2) At any time, the average value of the upper envelope ($f_{\max}\left(t\right)$) formed by the local maximum point and the lower envelope ($f_{\min}\left(t\right)$) formed by the local minimum point is zero: $\left[f_{\max}\left(t\right)+f_{\min}\left(t\right)\right]/2=0$.

Second, in order to more intuitively show the changes in air quality in these three cities during the epidemic and the impact of the epidemic and lockdown measures, a recursive graph was used to allow visualization of the trajectory periodicity through phase space. By drawing a recursive graph, we studied some aspects of the m-dimensional phase space trajectory of the AQI through a two-dimensional representation, which reflected the predictability of air quality. Then, recursive quantitative analysis was used for the quantitative analysis of changes in predictability. 

### 2.1. Ensemble Empirical Mode Decomposition

For cities experiencing severe epidemic conditions and those with mild epidemic conditions, the difference in AQI between 2020 and 2019 minus the difference in AQI between 2019 and 2018 was evaluated in this paper. The change in trends for these pollutants caused by COVID-19 was calculated by
(1)$$Poll_{i,COVID-19}=\Delta Poll_{2020}-\Delta Poll_{2019},\;i=1,\ldots ,348$$
where $Poll_{i,COVID-19}$ refers to the changes in the concentration of pollutants in city i caused by the outbreak. $Poll_{2020}$, $Poll_{2019}$, and $Poll_{2018}$ represent the pollutants in 2020, 2019, and 2018, respectively. $\Delta Poll_{2020}$ is the changes in air pollutant concentrations from 2019 to 2020, and $\Delta Poll_{2020}=Poll_{2020}-Poll_{2019}$. $\Delta Poll_{2019}$ is the changes in air pollutant concentrations from 2018 to 2019, and $\Delta Poll_{2019}=Poll_{2019}-Poll_{2018}$. We averaged the change values of 348 cities to obtain a sequence. This sequence represents the overall change. Then, the EEMD method was adopted to obtain the long-term effect of COVID-19 on the AQI. In this paper, EEMD was used to analyze the differences in air pollutant changes. The other IMFs had their meanings separately, and IMF1, IMF2, …, and IMF12 indicated that the time scale decreased according to the calculation formula of the IMFs. Since air quality data are non-stationary and highly volatile, much short-term redundant information exists. Therefore, it is necessary to use EEMD to extract the long-term trend, which can reduce the interference of outliers and is more conducive to revealing the long-term essential relationship. Since the last IMF is the closest IMF to the long-term trend, it was used in this paper. Specifically, the EEMD algorithm steps were as follows [29]:

(1) The Gaussian white noise sequences, $\varepsilon_{i}\left(t\right),$ were randomly generated and then introduced into ΔAQI to obtain $X_{i}\;\left(t\right)$. The coefficients of the white noise during each iteration were 1, and $X_{i}\left(t\right)=\Delta AQI+\varepsilon_{i}\left(t\right),\;i=1,2,\ldots ,N$, where N represents the number of iterations.

(2) A cubic spline interpolation function was adopted to fit the upper and lower envelopes of the original data on the basis of the maximum and minimum points of ΔAQI.

(3) The mean values of the upper and lower envelopes $m_{i}\left(t\right)$ were subtracted from $X_{i}\left(t\right)$, following the equation
(2)$$c_{i}\left(t\right)=X_{i}\left(t\right)-m_{i}\left(t\right)$$

(4) If $c_{i}\left(t\right)$ contains a negative local maximum or positive local minimum, steps 2 and 3 need to be performed again.

(5) Finally, $X_{i}\left(t\right)$ can be decomposed into IMFs after multiple filter operations:(3)$$X_{i}\left(t\right)=\sum c_{i}^{j}\left(t\right)+e_{i}^{n}\left(t\right)$$
where $e_{i}^{n}\left(t\right)$ denotes the residuals after decomposition.

(6) Steps 1–5 were executed iteratively, and in each iteration, the white noise sequence was generated randomly. Then, the decomposition results were
(4)$$X\left(t\right)=\sum \sum c_{i}^{j}\left(t\right)+e_{i}^{m}\left(t\right)$$

The original sequence can be decomposed as follows:(5)$$X\left(t\right)=\sum_{j=1}^{N-1}c^{j}\left(t\right)+e^{N}\left(t\right)$$
where $c^{j}\left(t\right)=\frac{1}{N}\sum_{i=1}^{N}c_{i}^{j}\left(t\right)$. $c^{j}\left(t\right)$ is the *j*th IMF after decomposition, which can be represented by ${\mathrm{IMF}}_{j}$.

### 2.2. State Phase Reconstruct

The state phase reconstruction method was adopted in this paper to further analyze the AQI time series [30]. By selecting the appropriate time lag factor τ, the AQI time series was embedded in the m-dimensional phase space. The number of points in the reconstructed phase space was $n=N-\left(m-1\right)$. The phase space of the structure was $AQI={\left(AQI_{j},AQI_{j+\tau },\ldots ,AQI_{j+\left(m-1\right)\tau }\right)}^{T}$. Therefore, the chaotic attractor of the dynamic system contained in the original AQI sequence can be unfolded without folding in the high-dimensional reconstructed phase space, thus showing its implicit evolutionary pattern and describing its pattern of motion.

### 2.3. Recurrence Plots

The recurrence plots effectively depicted the characteristics of the AQI system, represented by a recurrence matrix. The elements in this matrix were
(6)$$R_{i,j}\left(\varepsilon \right)=\Theta \left(\varepsilon -∥\vec{AQI_{i}}-\vec{AQI_{j}}∥\right),i,j=1,\ldots ,N$$
where ε represents the threshold distance given in advance. $∥\vec{AQI_{i}}-\vec{AQI_{j}}∥$ is the Euclidean norm of $\left(\vec{AQI_{i}}-\vec{AQI_{j}}\right)$, and $\Theta \left(\cdot \right)$ is the Heaviside function, which is defined by
(7)$$\Theta \left(x\right)=\left\{\begin{array}{c} 0,\;x<0\\ 1,\;x\geq 0 \end{array}\right.$$

The value of $\Theta \left(x\right)$ corresponded to the recursive state under ε. We adopted different colors to represent the binary elements of the recurrence plots. When $R_{i,j}\left(\varepsilon \right)=1$, black dots were used, and when $R_{i,j}\left(\varepsilon \right)=0$, white dots were employed.

### 2.4. Recurrence Quantification Analysis (RQA)

To supplement the visual information provided by a recursive graph, RQA indicators were adopted to analyze the different characteristics of AQI according to the diagonal structure and other aspects of recurrence plots [30]. Specifically, the RQA indicators used in this paper included determinacy (DET), maximum diagonal length (LMAX), entropy (ENT), laminarity (LAM), and tripping time (TT).

(1) DET: The percentage of the number of recursive points of the diagonal line segment in all recursive points was defined by
(8)$$DET=\frac{\sum_{l=l_{min}}^{N-1}l\times P\left(l\right)}{\sum_{i,j=1}^{N}R_{i,j}}$$
where $P\left(l\right)$ is the number of segments of length l, only including segments with lengths exceeding the given $l_{min}$ before counting. The value of l is generally an integer no less than 2. $R_{i,j}$ indicates each element in this matrix. DET distinguishes the isolated recursive points in the recursive graph from the organized recursive points, which form continuous diagonal line segments. The denser the lines along the main diagonal in the recursive graph, the stronger the system’s certainty.

(2) LMAX: The maximum diagonal length parallel to the main diagonal in a recursive graph:(9)$$LMAX=\max\left({\left\{l_{i}\right\}}_{i=1}^{N}\right)$$

The larger LMAX is, the slower the bifurcation velocity.

(3) ENT: The increase in entropy corresponded to the enhancement of AQI sequence complexity, which is defined by
(10)$$ENT=-\sum_{l=l_{min}}^{N}p\left(l\right)\times \ln p\left(l\right)$$

(4) LAM: Laminarity is another important indicator that can be used to measure the stability of an AQI system, defined by
(11)$$LAM=\frac{\sum_{l=l_{min}}^{N}l\times P\left(l\right)}{\sum_{l=1}^{N}l\times P\left(l\right)}$$

(5) TT: Trapping time was adopted to estimate the duration of a system remaining in a characteristic state defined by $TT=\sum_{l=l_{min}}^{N}l\times p\left(l\right)$.

In order to make the recursive graph and recursive quantitative analysis method clearer, we took the RQ plot for 12 cities that experienced severe epidemic conditions as an example. The result is provided in Figure 1. The index of recursive quantitative analysis was based on the number of black and white dots.

### 2.5. Air Quality and COVID-19 Data

The AQI was selected in this paper to measure the air quality of cities in China on the basis of a previous study [20]. According to the definition provided by the China Meteorological Administration (http://www.cma.gov.cn/ (accessed on 8 March 2021)), AQI is a dimensionless index that quantitatively describes the air quality status without specific units. It usually takes sulfur dioxide (SO_2_), nitrogen dioxide (NO_2_), fine particulate matter (PM2.5), carbon monoxide (CO), particulate matter (PM10), and ozone (Q3) as accounting factors. It is on a scale of 0 to 500, with 0 being the best air quality and 500 being the worst. The daily air quality index (AQI) data from 1 January 2018 to 27 December 2020 were obtained from the Ministry of Ecology and Environment, PRC (https://www.mee.gov.cn/ (accessed on 8 March 2021)). The numbers of confirmed cases of COVID-19 in cities with severe epidemic conditions, with few confirmed cases and experiencing secondary outbreaks, were obtained from the Wind database (https://www.wind.com.cn/ (accessed on 8 March 2021)).

We considered three kinds of cities: cities with severe epidemic conditions, cities with few confirmed cases, and cities experiencing secondary outbreaks (Table 1). The data for this paper examined six kinds of pollutants, namely, PM2.5, PM10, SO_2_, NO_2_, CO, and O_3_, and the AQI of 348 cities (the data on 6 February 2019 were missing, and there were no data from 29 February 2019 and 2018; we deleted the data from these two days). To facilitate the presentation of empirical results and on the basis of the consideration of data availability, we collected data from three years and differentiated the samples by year. The sample covered the periods before and after the outbreak of COVID-19. Therefore, the sample size was sufficient for empirical analysis of the impacts of secondary outbreaks. Beyond that, the discussion on pollutant emissions was mainly based on the emissions monitored on that day and did not involve in-depth discussion on the causes and chemical reactions of pollutants.

The numbers of confirmed cases of COVID-19 in cities with severe epidemic conditions, with few confirmed cases and experiencing secondary outbreaks, were obtained from the Wind database. We chose these cities mainly because, compared with other Chinese regions, COVID-19 first broke out in Wuhan, Hubei Province. Therefore, these 12 cities in Hubei Province, including Wuhan, where the epidemic was relatively serious, were included as a group of samples. According to the statistics from the Hubei Provincial Health Commission (https://wjw.hubei.gov.cn/ (accessed on 8 March 2021)), these 12 cities have experienced severe outbreaks, with more than 100 confirmed cases in the sample period, which is representative. However, the confirmed number of people in other cities in Hubei, such as Tianmen City and Shennongjia City, was less than 100, so they were not used as samples in this paper. Cities with few confirmed epidemic cases in the same period as the second group of samples come from different provinces, and there were almost no cases of infection in these cities. In addition, for Harbin, Dalian, Urumqi, Chengdu, and Beijing, according to the National Health Commission, PRC (http://www.nhc.gov.cn/ (accessed on 8 March 2021)), and media reports from the cities, there have been apparent secondary outbreaks in these cities in China. The number of confirmed cases in these cities rose again after a period of stability. These cities are the third group.

## 3. Empirical Results

This paper aimed to analyze and compare the evolution of levels of different pollutants during the periods before and after the COVID-19 pandemic. The EEMD method was adopted to describe the differences in evolutionary trajectories of these pollutants by extracting the last IMF, which indicates the change in trends of different pollutants caused by COVID-19. We also considered and compared the last IMF of these pollutants in cities with severe epidemic conditions, fewer confirmed cases, and repeated COVID-19 outbreaks for more robust results. CEEMDAN was also adopted for the robustness test, and the results are reported in the Appendix A. In addition, to further investigate the effect of the COVID-19 epidemic on air quality in cities with different COVID-19 conditions, the RP and RQA methods were employed to analyze the predictability and stability of the AQI before and after the COVID-19 pandemic.

### 3.1. Effects of Epidemic Context on Different Pollutants in 348 Cities

We adopted EEMD to extract the last IMF values of six pollutants and AQI during the COVID-19 pandemic. The mean values of the last IMF for each month are also reported in Table 2 and in Figure 2 and Appendix A.

As shown in Table 2 and Appendix A, within two months (Jan and Feb) after the start of the COVID-19 outbreak, the emissions of CO, NO_2_, PM10, and PM2.5 were reduced, but the emissions of O_3_ and SO_2_ were elevated. Considering that O_3_ is a secondary pollutant, and its emissions could be affected by meteorological factors such as lightning, it is difficult to evaluate whether the negative impact of the outbreak on O_3_ emissions is significant. From Apr to Jun 2020, strict home isolation and business shutdown policies (such as the traffic control measures issued in response to the epidemic) may have begun to be successful. Therefore, the rates of growth of SO_2_ levels began to decrease. After Aug 2020, the emissions of most pollutants declined. During this stable period, although the epidemic in China was effectively controlled, the government did not relax control over transportation. In Figure 2, different colors represent the increase or decrease in the last IMF of various pollutant emissions. For example, yellow represents the increased IMFs, indicating that pollutants had an upward trend on a long-term time scale. In addition, the sizes of color boxes represent degrees of change. During the initial outbreak period (January, February, March, and April), PM2.5, PM10, and NO_2_ declined, but changes in CO emissions were not noticeable. As shown in Appendix A, the ozone concentration was increased, and the concentration of other pollutants changed according to our conclusions.

Furthermore, the percentages of cities with improved air quality for different pollutants were analyzed in this paper, and the results are provided in Table 3 and Appendix A. Beyond that, the averages of the monthly city-wide air pollutant emission data for primary pollutants are provided in Appendix A, and the time trends of the monthly city-wide mean values for different primary pollutants are reported in Appendix A.

As shown in Table 3 and Appendix A, the proportion of cities with improved air quality varied by pollutant. Specifically, the percentages of cities with decreasing O_3_ and SO_2_ were based on less than 30% of the emissions monitored on that day instead of considering their chemical reactions. Nevertheless, the percentages of cities showing improvement in other pollutants exceeded 50%, and there was no noticeable change in the percentage of cities with decreasing AQI. This indicates that at the beginning of the COVID-19 outbreak, improvements in air quality were limited. However, the harm caused by the epidemic generated widespread concern [14,15,16], and the travel behavior of residents in many cities has not been limited. In fact, according to the data released by the National Bureau of Statistics in China (https://data.stats.gov.cn/easyquery.htm?cn=A01 (accessed on 8 March 2021)), the average monthly passenger flows of railway, highway, water transportation, and shipping in 2020 decreased by 39.80, 47.1, 45.03, and 36.71 percent, respectively, compared with 2019. The situation may be more complicated because of international trade [31,32]. Beyond that, in 2020, the number of motor vehicles in China reached 372 million, and 33.28 million new motor vehicles were registered, an increase of 1.14 million over 2019 according to the statistics of China’s Ministry of Public Security (https://www.mps.gov.cn/ (accessed on 8 March 2021)). According to the travel data released by Didi Travel (https://www.didiglobal.com/ (accessed on 8 March 2021)) in 2020, due to the epidemic, the demand for online car-hailing reached a trough in February 2020, but only four months later, user travel demand in June returned to the level of the same period in 2019. To avoid being infected using shared means of transport, some residents may have preferred to drive private cars instead of public transport, resulting in increased air pollutant emissions. This result is similar to existing literature [33,34] that confirms that the increase in air pollution is due to the shift of the preferred mode of travel from public transport to personal motor vehicles during weekdays. Beginning in May 2020, strict home isolation and travel control policies began to show success. In August 2020, over 70% of urban air quality improved, and in the stable period from September to December, this ratio was maintained above 60%.

### 3.2. Trends in AQI in Different Classes of Cities

Considering that trends in AQI may differ in different classes of COVID-19 outbreaks in Chinese cities, it is difficult to draw valuable conclusions on the basis of analyzing pollutant emissions averaged across all cities. Specifically, we selected three classes of cities: 12 cities in Hubei that have experienced severe COVID-19 epidemic conditions, 24 cities where very few cases of COVID-19 have been confirmed, and 5 cities with repeated COVID-19 outbreaks.

#### 3.2.1. The Evolution of Trends in AQI in the Cities in Hubei Experiencing Severe COVID-19 Epidemic Conditions

Similar to the process of calculation described above, we adopted the EEMD method to obtain the last IMF for the evolution of trends in AQI in these cities and calculated the mean values of the last IMF in each month (shown in Appendix A). In Appendix A, yellow and blue represent the overall rise and fall, respectively, of the AQI, and the extent of each rise and fall is indicated by the size of the corresponding block. Among 12 cities in Hubei experiencing severe COVID-19 epidemic conditions, air quality improved from January 2020 to April 2020 in most cities, including Xiangyang, Huangshi, Yichang, Jingmen, Xianning, and Shiyan. From June 2020 to September 2020, air quality also improved in Wuhan, Xiaogan, Huanggang, and Jingzhou, which may have resulted from the implementation of a comprehensive urban blockade and home isolation policy. However, with the gradual stabilization of the COVID-19 epidemic, air quality has declined in most of the 12 cities. This phenomenon may have been caused by the retaliatory consumption of residents and the weakening of blockade policy implementation.

In Appendix A, red and blue represent the changes in the evolution of AQI trends during the initial COVID-19 outbreak and the stable period in the cities in Hubei Province, respectively. As shown in Appendix A, urban air quality in Hubei improved to varying degrees during the initial COVID-19 outbreak (from January 2020 to May 2020). In the stable period (from April 2020 to December 2020), air quality improvement trends were slightly weakened in most of these 12 cities, and air quality deteriorated in Shiyan. On the whole, in regions experiencing severe COVID-19 outbreaks, air quality was improved.

#### 3.2.2. The Evolution of Trends in AQI in Cities with Few Confirmed COVID-19 Cases

In addition to considering the evolution of trends in AQI in cities experiencing severe COVID-19 outbreaks, 24 cities from different provinces with few confirmed COVID-19 cases are also discussed in this paper, which could help us understand the evolution in AQI trends in regions in which the COVID-19 epidemic is under control. The evolution of AQI trends in these cities is described in Appendix A.

In Appendix A, blue and yellow represent the improvement and deterioration of air quality, respectively. Air quality deteriorated from Jan 2020 to May 2020, corresponding to the initial outbreak of COVID-19, in most of these 24 cities, including Heyuan, Aksu, and Baise. However, the degree of deterioration in air quality in these cities weakened after August 2020. In addition, air quality in most cities with few confirmed COVID-19 cases has not improved during the pandemic. In fact, many cities have not implemented rigorous traffic control, business shutdown, or home isolation policies because the epidemic is well controlled and the number of infected individuals is minimal (on the basis of relevant information on Chinese government websites, https://www.gov.cn/ (accessed on 8 March 2021)). Daily economic operations are less affected in these cities. Therefore, improvements in air quality were found only in areas experiencing severe epidemic conditions. Similarly, the evolution of air quality trends in cities with few confirmed COVID-19 cases during the outbreak and stable periods is illustrated in Appendix A.

As shown in Appendix A, during the initial outbreak of COVID-19, air quality was improved only in approximately 40% of these cities, including Fushan, Zhangjiajie, Xuancheng, Binzhou, and Linfen. With the epidemic situation gradually under effective control, the level of air quality improvement in these cities during the stable period also declined.

#### 3.2.3. The Evolution of Trends in AQI in Cities Experiencing Secondary COVID-19 Outbreaks

To analyze the change in urban air quality trends before and after the second COVID-19 outbreak, five cities (Yichun, Baicheng, Fushun, Heyuan, and Zhoushan) where new infection cases appeared again after the epidemic had stabilized for a period were chosen. In Appendix A, purple regions represent the months during which COVID-19 outbreaks recurred. The mean differences in the last IMF values between these two periods are also provided in a chart.

After the initial outbreak period, the values of the changes in the last IMF were negative in Dalian, Harbin, and Urumqi, indicating that during this period, air quality improved in these cities by approximately 12.91, 6.28, and 0.36, respectively. However, air quality deteriorated in Beijing and Chengdu during this period by approximately 0.77 and 5.04, respectively. During the second outbreak of COVID-19, the values of changes in the last IMF were less than zero in Harbin (−1.07), Dalian (−1.64), Chengdu (−5.93), and Beijing (−4.05), but air quality deteriorated in Urumqi (1.98). The degree of change in the last IMF before and after the COVID-19 outbreak was reduced, indicating that the sensitivity of air quality to epidemic conditions decreased.

This paper further provides the long-term changes in each month of these three types of cities in Figure 3. As shown in Figure 3, there was a significant difference during COVID-19. For the cities with a few confirmed cases (24 cities), the AQI increased in about six months, indicating that air pollution could be deteriorated. Then, air quality may be improved after six months. However, for the cities with a serious COVID-19 pandemic (12 cities), air quality was improved significantly, especially in the first month and eighth months after the outbreak of COVID-19. The air quality also improved for the cities with a second outbreak, but the improvement was not as significant as in cities with more severe outbreaks. The lag effect appeared at around half a year after the outbreak. It is worth noting that these curves do not represent changes in the stability of air quality; they reflect changes in the monthly mean of air quality. In fact, the impact of the epidemic on air quality is complex. From the perspective of predictability, there should be a better way to characterize the impact of the epidemic on the stability of air quality.

## 4. The Predictability of the Evolution of AQI Trends Caused by COVID-19

Previous studies involving air quality prediction used econometric or numerical simulation methods [35,36] or aimed to provide specific air quality estimates or CO_2_ for the next few days or weeks. Considering new information makes the system less predictable [37]. This paper does not intend to estimate the concentration of air pollutants in the future and pays more attention to whether the outbreak of COVID-19 could affect the varying intensity of the AQI. In other words, it aimed to evaluate whether the epidemic outbreak reduces the difficulty and accuracy of estimating the future air quality value. Namely, there are differences in the evolution of air quality before and after the outbreak of COVID-19, which may lead to a change in the regularity of air quality data. Differing from regression models or networks [38,39,40], the recursive graph contains rich information for the quantitative comparison of air quality before and after the beginning of the epidemic; this paper further adopted recurrence quantification analysis to depict the complex mechanisms underlying air quality dynamics. Some indicators were used to represent the characteristic properties of recursive graphs. These indicators of recursive quantitative analysis were used to compare the dynamic behavior of the phase space of the AQI in 2018, 2019, and 2020. First, we calculated the mean values of various RQA indicators, including DET, LMAX, ENT, LAM, and TT. All indicators are reported in Figure 4.

### 4.1. Twelve Cities That Experienced Severe Epidemic Conditions in Hubei

Temporal trends in the mean value of the AQI in the 12 cities in Hubei with severe COVID-19 epidemic conditions from 2018 to 2020 are presented in Appendix A. Compared with the mean values from January to April 2019, the AQI declined from January to April 2020. Although air pollutants appeared to increase during the Spring Festival, in the first few months, home isolation measures and travel restrictions restrained the deterioration of air quality in cities with severe COVID-19 outbreaks. However, compared with the air quality in the second and third quarters of 2019, the air quality was not improved in the second and third quarters of 2020. It can be deduced that the effect of restrictive measures on improving air quality began to weaken as the epidemic was gradually controlled. The values of RQ indicators and RQ plots for the 12 cities that experienced severe epidemic conditions are reported in Figure 1 and Figure 4.

In Figure 1, the white part indicates that the trend of AQI sequence differed greatly from the value of the preceding and following sequences, and the fluctuation was more violent and difficult to predict. Black indicates stable parts of the sequence. The color meaning of the recursion graph for the remaining two types of cities in the supplement was the same. In early 2020, the AQI sequence showed small fluctuations and decreased predictability. However, the amplitude was small, and it is difficult to clearly see the detailed fluctuation of air quality in three years from the recursive graph. Therefore, we compared three years of recursive indicators. In Figure 4, the vertical axis is the calculation result of each index, which is only a numerical value and does not include units. The histogram on the right represents the average value of this measure for each year in such cities. The mean values of all RQA indicators showed increasing trends from 2018 to 2019. However, in 2020, the epidemic broke out in these cities. Specifically, (1) the mean values of DET and ENT in 2020 were less than those in 2019, indicating that the certainty and stability of the system declined from 2019 to 2020. Although restrictions during the epidemic period restricted residents’ daily travel and consumption, with the passage of time and the stability of the epidemic, air pollutants emitted by residents did not decrease steadily. (2) The mean value of LMAX for 2020 was less than that in 2019, indicating that the rate of change in AQI declined, which is consistent with the above conclusion. (3) The mean value of TT in 2020 was less than that in 2019, indicating that the duration of slow signal change was continuously decreasing. Therefore, it could also be deduced that the stability of the AQI system decreased. 

### 4.2. Twenty-Four Cities Where Very Few Cases of COVID-19 Were Confirmed

In addition to the cities that experienced severe epidemic conditions, we also considered the characteristics of fluctuations in air quality parameters in cities where the epidemic was not serious. The temporal trends in AQI and the recurrence plots of the evolution of trends in AQI in these cities are presented in Appendix A.

In the 24 cities with few COVID-19 cases, air quality improved overall because of restrictive epidemic control policies or environmental control measures. Areas of uncertainty were observed in the middle of the sample, indicating that retaliatory consumption led to the increasing fluctuation of the impact of the epidemic on air quality (Appendix A). However, in the stable stage of the epidemic, the long-term change trend in the AQI gradually stabilized, indicating that residents’ retaliatory consumption lasted for only a few months before the long-term trend in air quality became regionally stable. The RQA indicators for cities where the epidemic situation was not serious are illustrated in Figure 4 and Appendix A. As shown in Figure 4 and Appendix A, the changes in RQA indicator trends in 24 cities from 2019 to 2020 where the epidemic was not serious differed from those in 12 cities that experienced severe epidemic conditions. The RQA indicators DET, LMAX, ENT, and LAM decreased from 2018 to 2019 but increased from 2019 to 2020, indicating that the certainty and stability of the system improved in cities where the epidemic was not serious.

### 4.3. Five Cities Experiencing Secondary Outbreaks

To investigate the evolution of air quality trends resulting from the COVID-19 pandemic in cities with secondary outbreaks, temporal trends in AQI were analyzed, and recurrence plots of the evolution of trends in AQI in these cities were constructed (shown in Appendix A).

Although the trend of deteriorating air quality was reduced, it still existed. The long-term change in AQI was stable in early 2020, corresponding to the outbreak stage, indicating that during the outbreak stage, the deterioration of air quality was slow (Appendix A). However, the AQI during the first epidemic was stable. Even in the second outbreak, the fluctuation of the air quality trend was not affected. The RQA indicators for the recurrence plot of the evolution of trends in AQI are reported in Figure 4. As shown in Figure 4, the changes in RQA indicator trends from 2018 to 2020 for five cities with second COVID-19 outbreaks were complex and differed from those of the 12 cities that experienced severe or mild epidemics. Specifically, the mean values of DET and LAM showed steady downward trends, implying that the predictability and stability of the AQI signal both decreased. The mean value of TT showed a downward trend and then an upward trend, indicating that the duration of maintaining a certain state in the long-term trend of air pollutants first decreased and then increased.

## 5. Conclusions

This paper revisited the impact of COVID-19 on air quality in China. We not only investigated the changes in six air pollutants before and after COVID-19 outbreaks in 348 cities in China by extracting the differences in the growth trends between 2018–2019 and 2019–2020 but also applied recurrence plots and recursive quantitative analysis to compare the changes in the predictability of AQI. It was confirmed that due to the initial outbreak and restrictive mitigation policies, the concentrations of different pollutants changed, and the predictability of air quality was also affected. Interestingly, we found a phenomenon that differs from the existing literature, namely, that the air quality in the outbreak area does not necessarily improve, and it is possible that the air quality deteriorates with the reduction of the severity of the outbreak and the exposure to lockdown policies. This may be due to the rebound effect of production and consumption as well as the indirect industrial correlation effect between adjacent regions.

In the three to four months after the outbreak, there was an increasing proportion of cities with reduced O_3_ concentrations. We hold that the government should formulate corresponding control policies for different air pollutants in the post-epidemic era. Changes in trends in the impacts of the COVID-19 pandemic on AQI were found to show obvious differences among different classes of cities. Compared with the cities that did not experience severe epidemic conditions, air quality noticeably improved in the cities in Hubei, and for the cities with secondary outbreaks, air quality first improved and then deteriorated. During the second stable period, the sensitivity of air quality to epidemic conditions decreased. Therefore, it is suggested that in cities with severe epidemic conditions, it is necessary to strengthen the management of residents’ travel, maintain the effect of improving air quality, and ensure the smooth operation of the economy.

In addition, according to the results of RPs, in the stable period of the epidemic, there was instability in the cities with severe COVID-19 outbreaks, indicating that the impact of COVID-19 is challenging to predict. Nevertheless, AQI trends stabilized more rapidly in cities with fairly effective epidemic control. According to the RQA results, the predictability and stability of AQI were improved in cities where the epidemic was not serious, but in cities with severe COVID-19 outbreaks in Hubei, the opposite was true. This implies that in cities where the epidemic was not serious, the government can estimate the future AQI index on the basis of past air quality to formulate relevant policies to effectively improve regional air quality in the post-epidemic era.

## 6. Limitations and Future Research Directions

We hold that this study has three main deficiencies. (1) Some pollutants are secondarily produced in the atmosphere, and the way in which to distinguish them from the evolution of primary pollutants on the basis of COVID-19 conditions could not be solved. Second, the pollutants emitted in a city are transported to other regions within a few hours and days, which corresponds to the spatial effect of air pollutants. The EEMD method could not effectively depict this spatial effect. Therefore, in subsequent work, we intend to further improve the EEMD method and introduce the spatial weight matrix to investigate this spatial effect. (2) Recursive quantitative analysis is not an econometric regression model, and the relationship between multiple variables, such as meteorological factors, cannot be considered simultaneously. Therefore, in the follow-up empirical analysis, we intend to construct a flexible regression model to depict its effects. (3) When an outbreak occurs, local lockdown measures can lead to changes in air quality in neighboring areas with industrial links. We found that it seems possible to analyze production linkages between cities through lockdowns during the pandemic and changes in air quality across cities.

## Figures and Tables

**Figure 1 ijerph-19-16119-f001:**
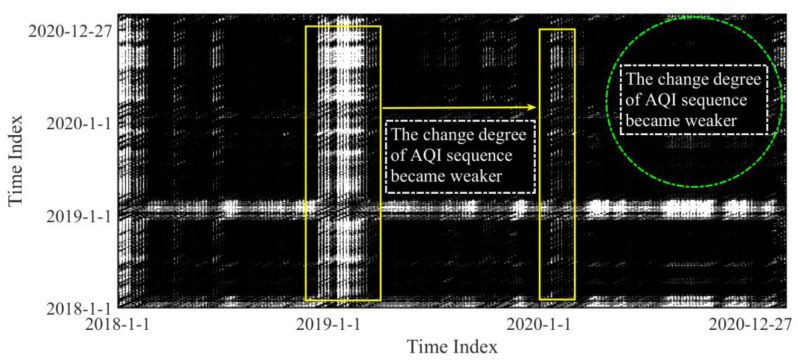
RQ plot for 12 cities that experienced severe epidemic conditions. (Note: black and white regions in the recursion graph represent the stability and instability of AQI sequence, respectively).

**Figure 2 ijerph-19-16119-f002:**
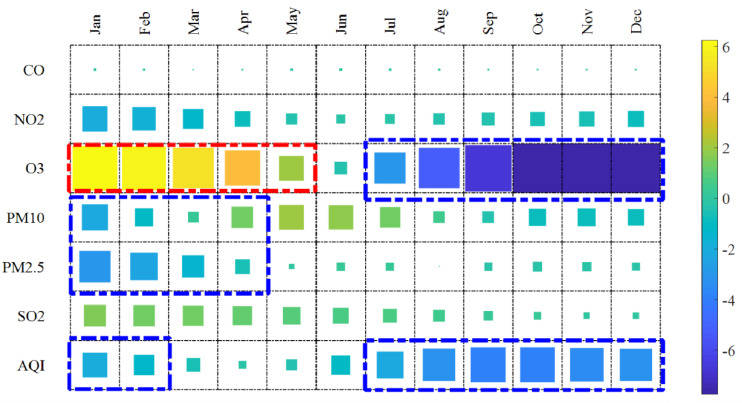
The trends in the mean values of the last IMF each month. (Note: The Y−axis indicates different pollutants, and the X−axis indicates the month. The square size represents the degree of change in air quality, and the larger the square size, the greater the degree. Blue and yellow represent the improvement and deterioration of air quality, respectively. The numbers corresponding to the colors also represent changes in air quality. A positive (negative) number indicates deterioration (improvement) in air quality. The smaller the number, the better the air quality improvement).

**Figure 3 ijerph-19-16119-f003:**
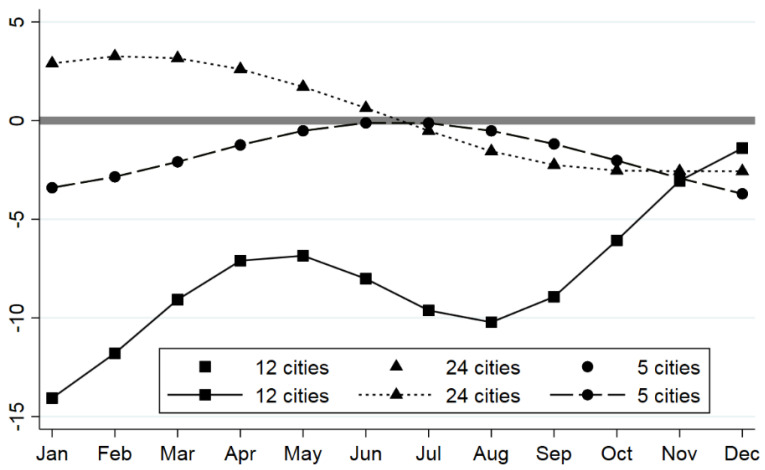
Trends of long-term changes in AQI in three types of cities. (Note: A total of 12 cities had severe outbreaks, 24 cities had few confirmed cases, and 5 cities had secondary outbreaks. The Y−axis indicates the value of the long-term change of AQI, and the X−axis indicates the month).

**Figure 4 ijerph-19-16119-f004:**
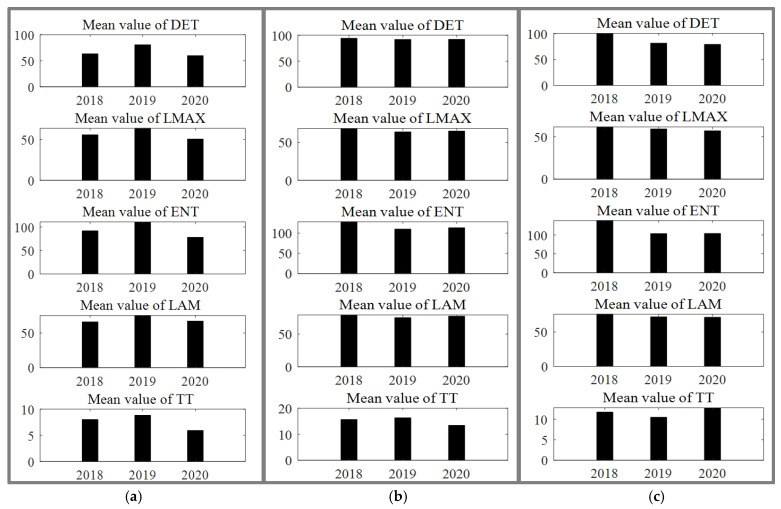
RQA indicators for cities under different epidemic conditions. (Note: (**a**) represents 12 cities with severe epidemics; (**b**) represents 24 cities with few confirmed cases; (**c**) represents 5 cities with secondary outbreaks. The Y−axis indicates the value of the RQA index, and the X−axis indicates the year).

**Table 1 ijerph-19-16119-t001:** Three types of cities studied in this paper.

City Type	Cities
Cities with severe epidemics	Ezhou, Wuhan, Xiaogan, Huanggang, Jingzhou, Suizhou, Xiangyang, Huangshi, Yichang, Jingmen, Xianning, and Shiyan
Cities with a few confirmed cases	Yichun, Baicheng, Fushun, Heyuan, Zhoushan, Zhangjiajie, Xuancheng, Jingdezhen, Aksu, Binzhou, Leshan, Zhenjiang, Longyan, Chengde, Yingkou, Wuhai, Baise, Lincang, Linfen, Jinchang, Anshun, Shizuishan, Xining, and Lhasa
Cities with a second outbreak	Harbin, Dalian, Urumqi, Chengdu, and Beijing

**Table 2 ijerph-19-16119-t002:** The mean values of the last IMF in each month ($\mu g/m^{3}$).

Month	CO	NO_2_	O_3_	PM10	PM2.5	SO_2_	AQI
January	−0.0164	−1.9292	6.2502	−2.1166	−3.0636	1.4610	−1.8965
February	−0.0113	−1.6811	5.9609	−0.9717	−2.3733	1.4046	−1.2603
March	−0.0032	−1.2551	5.2130	0.3340	−1.5065	1.2950	−0.5688
April	0.0064	−0.7515	3.8568	1.4051	−0.6651	1.1419	−0.1671
May	0.0139	−0.3742	1.9033	1.9118	−0.0777	0.9621	−0.3569
June	0.0173	−0.2283	−0.4794	1.8356	0.1985	0.7713	−1.1219
July	0.0161	−0.2735	−2.9691	1.2578	0.1855	0.5866	−2.2217
August	0.0115	−0.3951	−5.1941	0.3959	−0.0006	0.4132	−3.2504
September	0.0072	−0.5072	−6.7457	−0.3994	−0.1935	0.2669	−3.8183
October	0.0056	−0.6100	−7.5386	−0.8662	−0.2771	0.1658	−3.8209
November	0.0062	−0.7152	−7.7390	−0.9436	−0.2360	0.1149	−3.4874
December	0.0072	−0.8028	−7.6278	−0.8053	−0.1792	0.1084	−3.2060

**Table 3 ijerph-19-16119-t003:** The percentages of cities with improved air quality for different pollutants.

Month	CO	NO_2_	O_3_	PM10	PM2.5	SO_2_	AQI
January	58.62%	69.54%	18.39%	52.87%	58.91%	27.01%	55.17%
February	55.75%	68.68%	21.26%	52.01%	57.47%	26.72%	52.59%
March	53.45%	66.38%	26.15%	47.70%	53.74%	26.44%	51.72%
April	49.71%	61.49%	31.90%	45.69%	51.72%	25.00%	50.86%
May	47.13%	56.61%	43.10%	43.97%	50.29%	26.44%	54.02%
June	45.40%	55.17%	55.17%	45.11%	48.85%	28.16%	60.06%
July	45.11%	54.31%	68.10%	50.00%	51.15%	33.91%	66.09%
August	46.84%	54.89%	77.87%	50.86%	51.44%	35.06%	75.57%
September	45.98%	56.32%	82.18%	53.16%	56.03%	43.39%	76.44%
October	44.83%	57.18%	83.05%	54.60%	54.60%	44.25%	72.70%
November	43.68%	55.46%	82.76%	54.02%	54.02%	46.55%	70.69%
December	42.53%	55.46%	83.05%	54.60%	53.45%	47.41%	68.39%

## Data Availability

Not applicable.

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
