# Peer review of "Changes in Air Quality during the Period of COVID-19 in China"

_ijerph, 2022, doi:10.3390/ijerph192316119_

Round 1

Reviewer 1 Report (Previous Reviewer 4)

Not all suggested corrections were made in the manuscript. This manuscript has the characteristics of a review paper and not an original scientific article. It is necessary to explain the purpose of this work. The conclusions are expected and predictable.

Author Response

We would like to thank the editor and reviewers for their careful and thorough reading of this manuscript and for the thoughtful comments and constructive suggestions, which have helped us improve the manuscript and reach a better scientific level. All of the comments are valuable and have been very helpful during the revision process and have provided important guiding significance to our research. We have carefully revised the manuscript entitled “Changes in air quality during the period of COVID-19 in China” (ijerph-1926617) according to the editor and reviewers’ comments. The responses to the reviewers’ comments and the revised portions are reported below each question and are marked in blue and red, respectively. We tried our best to improve the manuscript by making the suggested changes. These changes did not influence the content and framework of the paper. We appreciate the editor’s and reviewers’ thoughtful work earnestly and hope that the corrections will meet with your approval. Thank you for the time and effort that was expended on this paper. The responses to the reviewers’ and editor’s comments are as follows:

Response to the Comments of Reviewer 1

Reviewer’s comments:

1: Not all suggested corrections were made in the manuscript. This manuscript has the characteristics of a review paper and not an original scientific article. It is necessary to explain the purpose of this work. The conclusions are expected and predictable.

RESPONSE: Thank you for the suggestion. We have tried our best to revised our manuscript. For the purpose of this paper, we have emphasized the aim of this paper in Section 1 Introduction, and the Abstract has also been revised. The detailed revisions related this comment are as follows:

Abstract: This paper aims to analyze the various impact of COVID-19 began in January 2020 on air quality in different types of cities. We analyze and discuss the different degrees of improvement of concentrations of six air pollutants (PM2.5, PM10, SO2, NO2, CO and O3) obtained from the Ministry of Ecology and Environment, PRC in different types of Chinese cities from 2018 to 2020 with difference method (the trend of pollutants in different years was differenced) and ensemble empirical mode decomposition (EEMD), and then adopt the recursive plots (RPs) and recursive quantitative analysis (RQA) for the first time to discuss whether air quality is more difficult to predict during the outbreak. The empirical results indicate that: (1) After the initial outbreak, only the emissions of NO2, CO and PM2.5 declined for the first 1-3 months, and during the fourth to fifth months the emissions of six air pollutants were elevated in most cities; (2) For the cities with serious epidemic situations in the Hubei province (China), the air quality is improved, but for the cities experiencing a second outbreak, the air quality was first enhanced and then deteriorated, and the sensitivity of air quality to COVID-19 re-outbreak is decreasing. For Dalian, Harbin and Urumqi, AQI is improved by approximately 12.91, 6.28 and 0.36, respectively. (3) In comparison, the predictability of AQI has declined in cities with serious epidemic situations in Hubei, but AQI achieves a stable state sooner in cities with mild epidemic. The conclusions may facilitate analysis of differences in air quality evolution characteristics and fluctuations before and after outbreaks from a quantitative perspective.

  1. Introduction

(…)

Our paper first focuses on the evolution of air quality during surges and stable periods of the epidemic for areas experiencing severe and mild COVID-19 epidemic conditions by adopting the EEMD method. Beyond that, whether the COVID-19 pandemic could affect the predictability of urban air quality in different cities has not to our knowledge been addressed in previous literature. Therefore, this paper adopts recurrence plots (RPs) and recurrence quantification analysis (RQA) to compare the characteristics of fluctuation in AQI before and after outbreaks, which could be used to describe the effect of the COVID-19 pandemic on the predictability of urban air quality.

We find that the reviewer’s comments have been quite helpful in improving the paper, and we have revised our paper point by point. Once again, thank you very much for your comments and suggestions!

Best Regards!

Yours sincerely!

Reviewer 2 Report (New Reviewer)

The paper highlights an important study on impact of COVID on air quality in China.   The authors do show how AQ was improved during the first part of the pandemic using EEMD signal analysis.  These results alone would merit publication. In the introduction (lines 36-37), the stated purpose of the paper that  "the negative impact of urban residents’ living activi- 37 ties on urban ambient air quality may be weakened, and air quality could be improved".  This goal is unclear to me.  I recommend the stated purpose be more clearly defined.  Didn't the reduced activities during the pandemic improve AQ overall ?

Also, it was difficult to follow the results as the authors referenced the supplementary materials often in sections 3 and 4.  I recommend either reducing the scope of this paper to include the figures/discussion in that supplementary section.  Perhaps focus mainly n section 3 that evaluated the pre-covid vs covid trends in AQ. 

Specific Comments

Fig. 1 please expand the figure caption to mention each stage of the difference method used.  What is meant by dynamic effects in top right figure.  What does IMF mean ?

Eqtn 1: Are POLL the annual average AQI ? 

line 170: How does EEMD make up for shortcoming of EMD ?

Section 2.5: should the criteria be normalized by population of cities (# cases/ city population) to better compare results and determine pandemic type.

line 290: Define CEEMDAN

Table 2. Add what the mean values are (and units ?) in table caption.

Fig 2.  Add year to x-axis

Pg 8.  I may be easier for reader to include the supplementary material discussed (figs/table) in the main text here.

line 370....still unclear why ozone concentrations increased during pandemic.  Can one access incoming short wave radiation to look for variability between 2019 and 2020 ? Perhaps, less aerosols contributed to increased photolysis ?

fig 3  please define the vertical axis labels.

line 488, fig 4.  no blue area seen ?

fig 3,4,5 provide more information on the graphs in the figure caption.  What do the three bars for each city represent ?

Author Response

We would like to thank the editor and reviewers for their careful and thorough reading of this manuscript and for the thoughtful comments and constructive suggestions, which have helped us improve the manuscript and reach a better scientific level. All of the comments are valuable and have been very helpful during the revision process and have provided important guiding significance to our research. We have carefully revised the manuscript entitled “Changes in air quality during the period of COVID-19 in China” (ijerph-1926617) according to the editor and reviewers’ comments. The responses to the reviewers’ comments and the revised portions are reported below each question and are marked in blue and red, respectively. We tried our best to improve the manuscript by making the suggested changes. These changes did not influence the content and framework of the paper. We appreciate the editor’s and reviewers’ thoughtful work earnestly and hope that the corrections will meet with your approval. Thank you for the time and effort that was expended on this paper. The responses to the reviewers’ and editor’s comments are as follows:

Response to the Comments of Reviewer 2

Reviewer’s comments:

1: The paper highlights an important study on impact of COVID on air quality in China.   The authors do show how AQ was improved during the first part of the pandemic using EEMD signal analysis. These results alone would merit publication. In the introduction (lines 36-37), the stated purpose of the paper that "the negative impact of urban residents’ living activi- 37 ties on urban ambient air quality may be weakened, and air quality could be improved". This goal is unclear to me. I recommend the stated purpose be more clearly defined. Didn't the reduced activities during the pandemic improve AQ overall ?

RESPONSE: Thank you for the suggestion. We have revised this statement. The original statement: "the negative impact of urban residents’ living activi- 37 ties on urban ambient air quality may be weakened, and air quality could be improved" comes from Chen et al. (2021), and it has been replaced with ‘the effects of urban residents’ daily travel and other activities on the environment may be mitigated by the lockdown measures’. In fact, not all pollution concentrations during the pandemic showed a trend of decline, and the concentration of some pollutants showed a trend of increase due to the rebound behavior of household consumption. The detailed revisions related this comment are as follows:

  1. Introduction

The outbreak and spread of COVID-19 had a significant impact on the lives of people and the operation of the economy (Bauwens et al., 2020; Venter et al., 2020; Briz-Redón, et al., 2021; McKee and Stuckler, 2020; Guan et al., 2020; Choi and Brindley, 2021). Governments have implemented various measures to prevent the spread of the disease, such as controls on movement and personnel isolation measures (Chang et al., 2020; Tian et al., 2020) and case isolation and home quarantine (Shi and Brasseur, 2020; Silver et al., 2020; Filonchyk et al., 2020; Cole et al., 2020), which has led to significant reductions in travel and industrial products. Therefore, the effects of urban residents’ daily travel and other activities on the environment may be mitigated by the lockdown measures (Chen et al., 2021)

2: Also, it was difficult to follow the results as the authors referenced the supplementary materials often in sections 3 and 4. I recommend either reducing the scope of this paper to include the figures/discussion in that supplementary section.  Perhaps focus mainly on section 3 that evaluated the pre-covid vs covid trends in AQI.

RESPONSE: Thank you for the suggestion. We have payed more attention on the discussion about the pre-covid vs covid trends in AQI, and some explanation of supplementary materials was moved in Supplementary Materials. The detailed revisions related this comment are as follows:

Fig. S8. Temporal trends in AQI in 12 cities experiencing severe COVID-19 epidemic conditions

As shown in Supplementary Materials, Fig. S8, air quality deteriorated first and then improved. After adopting the phase space reconstruction to analyze the last IMF, the recurrence plot could be constructed (Shown in Supplementary Materials, Fig. S9), depicting the changes in characteristics of long-term trends in AQI. According to Supplementary Materials, Fig. S9(a) and (b), the long-term trend in AQI was stable in early 2020, corresponding to the outbreak stage, indicating that during the outbreak stage, the deterioration of air quality was slow. However, in the stable period of the epidemic, there was instability, indicating that the impact of COVID-19 is challenging to predict.

Fig. S12. Temporal trends in AQI in cities with secondary outbreaks

In the cities that experienced secondary outbreaks, the air quality showed a pattern of initial improvement and subsequent deterioration (Supplementary Materials, Fig. S12). It could be deduced that in the first stage of the epidemic, the strict travel restriction policy inhibited the residents’ movement and reduced the residents’ consumption levels. Hence, the emissions of various air pollutants showed downward trends.

3: Fig. 1 please expand the figure caption to mention each stage of the difference method used. What is meant by dynamic effects in top right figure. What does IMF mean ?

RESPONSE: We apologize for not making Fig. 1 clear. We have revised the title of Fig. 1: ‘Fig. 1 The method for calculating changes in the growth trend of air pollution’. Then, we added more interpretation about the dynamic effects in top right figure and IMF. The detailed revisions related this comment are as follows:

Fig. 1 The method for calculating changes in the growth trend of air pollution

Fig. 1 shows the steps to calculate the difference in air quality growth trends. The dynamic effect in the figure refers to the dynamic change of air quality over time after the two difference methods are adopted, and there are differences in the mean value of changes in each month. Beyond that, the IMFs represent different frequency components in the variation of air quality. Thus, the final IMF can be used to measure the long-run trend of these variation differences.

Second, a recursive graph was used to allow visualization of trajectory periodicity through phase space. By drawing a recursive graph, we can study some aspects of the m-dimensional phase space trajectory of the AQI through a two-dimensional representation, which reflects the predictability of air quality. Then, recursive quantitative analysis was used for the quantitative analysis of changes in predictability.

4: Eqtn 1: Are POLL the annual average AQI ?

RESPONSE: Polli represent the pollutants in 2020, 2019 and 2018, respectively. In the subsequent calculation process, it represents the concentration of pollutants in the air, and the data used are the concentration data of various pollutants.

5: line 170: How does EEMD make up for shortcoming of EMD ?

RESPONSE: We have added more elaboration on the benefits of the EEMD approach. The detailed revisions related this comment are as follows:

where ,  and  represent the pollutants in 2020, 2019 and 2018, respectively. The results for each city were averaged and recorded as . Then, the EEMD method was adopted to obtain the long-term impact of COVID-19 on AQI. EEMD could make up the shortcoming of EMD. In the process of decomposing the IMF with EMD, many iterations are required, and the conditions for stopping the iteration lack a standard, so the IMFs obtained by different conditions for stopping the iteration are also different (Qiu et al., 2022). This shortcomings has been solved by EEMD proposed by Wu and Huang (2009) and used in this paper. In addition, EEMD can effectively solve the mode aliasing problem caused by EMD, and it can accurately and efficiently divide different frequency components. Specifically, when EEMD is adopted, white noise is added to the signal to be analyzed by taking advantage of the characteristic of uniform spectrum distribution of white noise, so that signals with different time scales can be automatically separated to the corresponding reference scale, which is the EEMD method. This method mainly adds white noise to the signal to supplement some missing scales, and has a good performance in signal decomposition. The EEMD decomposition principle is: when the additional white noise is evenly distributed in the whole time-frequency space, the time-frequency space is composed of different scale components divided by the filter bank. In this paper, EEMD is adopted to analyze the differences in air pollutant changes. The other IMFs have its own meaning separately, and IMF1, IMF2,…, IMF12 indicate the time scale is getting smaller and smaller according to the calculation formula of IMFs. Since the last IMF is the closest IMF to the long-term trend, it is used in this paper. Specifically, EEMD algorithm steps are as follows:

6: Section 2.5: should the criteria be normalized by population of cities (# cases/ city population) to better compare results and determine pandemic type.

RESPONSE: This is an insightful question. Before data processing, we also discussed whether to standardize the variables, but considering the overpopulation of the city, the proportion of confirmed epidemic cases was small, almost zero. In addition, for the Chinese government, when an outbreak occurs in a certain region, no matter how severe, there will be relatively strict lockdown measures. Therefore, the variables were not standardized in the final empirical analysis.

7: line 290: Define CEEMDAN

RESPONSE: Thank you for the suggestion. We have added a brief description of this method in the corresponding section. The detailed revisions related this comment are as follows:

  1. Empirical results

This paper aims to analyze and compare the evolution of levels of different pollutants during the periods before and after the COVID-19 pandemic. The EEMD method was adopted to describe the differences in evolutionary trajectories of these pollutants by extracting the last IMF, which indicates the change in trends of different pollutants caused by COVID-19. We also considered and compared the the last IMF of these pollutants in cities with severe epidemic conditions, fewer confirmed cases, and repeated COVID-19 outbreaks for more robust results. CEEMDAN is also be adopted for robust test and the results are reported in Supplementary Materials, Fig. S16 and Fig. S17. CEEMDAN can further solve the problem of mode aliasing. Firstly, the IMF component with auxiliary noise after EMD decomposition is added in this method, instead of adding Gaussian white noise signal directly to the original signal. Secondly, EEMD decomposition take the overall average of the modal components obtained after the empirical mode decomposition, while CEEMDAN decomposition takes the overall average calculation after the first order IMF component, and obtains the final first order IMF component. Then, the above operations are repeated for the residual parts. In this way, the transfer of white noise from high frequency to low frequency is effectively solved. In addition, to further investigate the effect of the COVID-19 epidemic on air quality in different kinds of cities, the RPs and RQA methods were employed to analyze the predictability and stability of AQI before and after the COVID-19 pandemic.

8: Table 2. Add what the mean values are (and units ?) in table caption.

RESPONSE: We have added the units in table caption. The detailed revisions related this comment are as follows:

Table 2 The mean values of the last IMF in each month ( )

Month

CO

NO2

O3

PM10

PM2.5

SO2

AQI

Jan

-0.0164

-1.9292

6.2502

-2.1166

-3.0636

1.4610

-1.8965

Feb

-0.0113

-1.6811

5.9609

-0.9717

-2.3733

1.4046

-1.2603

Mar

-0.0032

-1.2551

5.2130

0.3340

-1.5065

1.2950

-0.5688

Apr

0.0064

-0.7515

3.8568

1.4051

-0.6651

1.1419

-0.1671

May

0.0139

-0.3742

1.9033

1.9118

-0.0777

0.9621

-0.3569

Jun

0.0173

-0.2283

-0.4794

1.8356

0.1985

0.7713

-1.1219

Jul

0.0161

-0.2735

-2.9691

1.2578

0.1855

0.5866

-2.2217

Aug

0.0115

-0.3951

-5.1941

0.3959

-0.0006

0.4132

-3.2504

Sep

0.0072

-0.5072

-6.7457

-0.3994

-0.1935

0.2669

-3.8183

Oct

0.0056

-0.6100

-7.5386

-0.8662

-0.2771

0.1658

-3.8209

Nov

0.0062

-0.7152

-7.7390

-0.9436

-0.2360

0.1149

-3.4874

Dec

0.0072

-0.8028

-7.6278

-0.8053

-0.1792

0.1084

-3.2060

9: Fig 2. Add year to x-axis

RESPONSE: We have revised the caption. The detailed revisions related this comment are as follows:

Fig. 2 The trends in the mean values of the last IMF in each month in 2021

10: Pg 8. I may be easier for reader to include the supplementary material discussed (figs/table) in the main text here.

RESPONSE: Thank you for your suggestion. We hope to present more specific values. This provides a more detailed explanation of the percentage change.

11: line 370....still unclear why ozone concentrations increased during pandemic.  Can one access incoming short wave radiation to look for variability between 2019 and 2020 ? Perhaps, less aerosols contributed to increased photolysis ?

RESPONSE: This is an insightful question. During the pandemic, when societies cut emissions of ozone-producing nitrogen oxides (NOx), ozone concentrations actually rose. This may be due to the complex mechanism of ozone generation. We speculate that during the pandemic, after PM 2.5 decreased, there was no haze, and sunlight could penetrate the air more easily, providing more energy for surface ozone generation. In the absence of rigorous chemical data, these conclusions are not mentioned in the manuscript.

12: fig 3 please define the vertical axis labels.

RESPONSE: Thank you for your suggestion. We have added an explanation of the axis labels in Figure 3. Since the calculation results of each indicator in Figure 3 do not have units, they are not marked in the figure. The detailed revisions related this comment are as follows:

Fig. 3 RQA indicators for 12 cities that experienced severe epidemic conditions

In Fig. 3, the vertical axis is the calculation result of each index, which is only numerical value and does not include units. The histogram on the right represents the average value of this measure for each year in such cities. Mean values of all RQA indicators showed rising trends from 2018 to 2019. However, in 2020, the epidemic broke out in these cities. Specifically, (1) the mean values of DET and ENT in 2020 were less than those in 2019, indicating that the certainty and stability of the system declined from 2019 to 2020. Although restrictions during the epidemic period restricted residents' daily travel and consumption, with the passage of time and the stability of the epidemic, air pollutants emitted by residents did not decrease steadily. (2) The mean value of LMAX for 2020 was less than that in 2019, indicating that the rate of change in AQI declined, which is consistent with the conclusion drawn above. (3) The mean value of TT in 2020 was less than that in 2019, indicating that the duration of slow signal change was continuously decreasing. Therefore, it could also be deduced that the stability of the AQI system decreased.

13: line 488, fig 4. no blue area seen ?

RESPONSE: We apologize for this error, and Fig. 4 here has been changed to Fig. 3. The detailed revisions related this comment are as follows:

4.2. Twenty-four cities where very few cases of COVID-19 were confirmed

       In addition to the cities that experienced severe epidemic conditions, we also considered the characteristics of fluctuations in air quality parameters in cities where the epidemic was not serious. The temporal trends in AQI and the recurrence plots of the evolution of trends in AQI in these cities are presented in Supplementary Materials, Fig. S10 and Fig. S11.

In the 24 cities with few COVID-19 cases, air quality improved overall because of restrictive epidemic control policy or environmental control measures (Supplementary Materials, Fig. S10). Areas of uncertainty were observed in the middle of the sample, indicating that retaliatory consumption led to the increasing fluctuation of the impact of the epidemic on air quality (Supplementary Materials, Fig. S11). However, in the stable stage of the epidemic, the long-term change trend in AQI gradually stabilized, as reflected by the blue area in the lower right corner in Fig. 3, indicating that residents’ retaliatory consumption only lasted for a few months before the long-term trend in air quality became regionally stable. RQA indicators for cities where the epidemic situation was not serious is illustrated in Fig. 4 and Fig. S15.

14: fig 3,4,5 provide more information on the graphs in the figure caption. What do the three bars for each city represent ?

RESPONSE: Thank you for your advice. The histogram on the right represents the average of different types of cities, and we have added relevant notes. The detailed revisions related this comment are as follows:

In Fig. 3, the vertical axis is the calculation result of each index, which is only numerical value and does not include units. The histogram on the right represents the average value of this measure for each year in such cities. Mean values of all RQA indicators showed rising trends from 2018 to 2019. However, in 2020, the epidemic broke out in these cities. Specifically, (1) the mean values of DET and ENT in 2020 were less than those in 2019, indicating that the certainty and stability of the system declined from 2019 to 2020. Although restrictions during the epidemic period restricted residents' daily travel and consumption, with the passage of time and the stability of the epidemic, air pollutants emitted by residents did not decrease steadily. (2) The mean value of LMAX for 2020 was less than that in 2019, indicating that the rate of change in AQI declined, which is consistent with the conclusion drawn above. (3) The mean value of TT in 2020 was less than that in 2019, indicating that the duration of slow signal change was continuously decreasing. Therefore, it could also be deduced that the stability of the AQI system decreased.

We find that the reviewer’s comments have been quite helpful in improving the paper, and we have revised our paper point by point. Once again, thank you very much for your comments and suggestions!

Best Regards!

Yours sincerely!

Reviewer 3 Report (New Reviewer)

This article is rejected for many reasons:

- the methods adopted aren’t adequately described

- the results of the research are not clearly presented

- the level of the English language is so low that some parts of the text are incomprehensible

The aim of the work is to analyse the trend of the main air pollutants in 328 cities from 2018 to 2020, with different analytical methodologies. However, the methods are not clearly explained, it is not clear how they are applied to the dataset:

2.1. Ensemble Empirical Mode Decomposition: the method is not clearly explained

155  The caption of fig. 1 does not describe “The specific steps”

166-167  ?????,?????−19=(????2020−????2019)−(????2019−????2018),?=1,…,348 (1)

                  where ????2020, ????2019 and ????2018 represent the pollutants in 2020…”

What does it means “the pollutants in 2020” ? Is it the daily concentration of pollutants (PM10, NO2, etc)?

What does it means     ?=1,…,348? Why 348?

168 - “The results for each city were averaged and recorded as Δ???

  Why AQI? Is the result an average over 328 cities? The result is unclear

2.2. State Phase Reconstruct: the method is not clearly explained

206 – “By selecting the appropriate time lag factor ?

    How the factor ? was determined?

2.3. Recurrence Plots: the method is not clearly explained

In this article there is some confusion regarding emissions and concentrations of pollutants. The observations concern the concentration of secondary pollutants (PM10, NO2, etc.)

303 - 306  “…the emissions of CO, NO2, PM10 and PM2.5  were reduced, but the emissions of SO2 were        elevated. Considering that O3 is secondary  pollutant and its emissions…”

 The conclusions and a large part of the article show a lack of knowledge regarding air pollution.

It would be necessary to completely rewrite the article.

Author Response

We would like to thank the editor and reviewers for their careful and thorough reading of this manuscript and for the thoughtful comments and constructive suggestions, which have helped us improve the manuscript and reach a better scientific level. All of the comments are valuable and have been very helpful during the revision process and have provided important guiding significance to our research. We have carefully revised the manuscript entitled “Changes in air quality during the period of COVID-19 in China” (ijerph-1926617) according to the editor and reviewers’ comments. The responses to the reviewers’ comments and the revised portions are reported below each question and are marked in blue and red, respectively. We tried our best to improve the manuscript by making the suggested changes. These changes did not influence the content and framework of the paper. We appreciate the editor’s and reviewers’ thoughtful work earnestly and hope that the corrections will meet with your approval. Thank you for the time and effort that was expended on this paper. The responses to the reviewers’ and editor’s comments are as follows:

Response to the Comments of Reviewer 3

Reviewer’s comments:

1: This article is rejected for many reasons:

- the methods adopted aren’t adequately described

- the results of the research are not clearly presented

- the level of the English language is so low that some parts of the text are incomprehensible

RESPONSE: We have added more explanations of methods and conclusions, and checked for language errors throughout the paper.  

2: The aim of the work is to analyse the trend of the main air pollutants in 328 cities from 2018 to 2020, with different analytical methodologies. However, the methods are not clearly explained, it is not clear how they are applied to the dataset:

RESPONSE: We have given the detailed processing section in Section 2 and Section 2.1.

3: 2.1. Ensemble Empirical Mode Decomposition: the method is not clearly explained

RESPONSE: EEMD algorithm steps are provided in the form of mathematics in Section 2.1. EEMD can effectively solve the mode aliasing problem caused by EMD, and it can accurately and efficiently divide different frequency components. Specifically, when EEMD is adopted, white noise is added to the signal to be analyzed by taking advantage of the characteristic of uniform spectrum distribution of white noise, so that signals with different time scales can be automatically separated to the corresponding reference scale, which is the EEMD method. This method mainly adds white noise to the signal to supplement some missing scales, and has a good performance in signal decomposition. The EEMD decomposition principle is: when the additional white noise is evenly distributed in the whole time-frequency space, the time-frequency space is composed of different scale components divided by the filter bank. In this paper, EEMD is adopted to analyze the differences in air pollutant changes. The other IMFs have its own meaning separately, and IMF1, IMF2,…, IMF12 indicate the time scale is getting smaller and smaller according to the calculation formula of IMFs. Since the last IMF is the closest IMF to the long-term trend, it is used in this paper.

4: 155, The caption of fig. 1 does not describe “The specific steps”

RESPONSE: Fig. 1 shows the steps to calculate the difference in air quality growth trends. The dynamic effect in the figure refers to the dynamic change of air quality over time after the two difference methods are adopted, and there are differences in the mean value of changes in each month. Beyond that, the IMFs represent different frequency components in the variation of air quality. Thus, the final IMF can be used to measure the long-run trend of these variation differences. We have also revised the caption of Fig. 1:

Fig. 1 The method for calculating changes in the growth trend of air pollution

Fig. 1 shows the steps to calculate the difference in air quality growth trends. The dynamic effect in the figure refers to the dynamic change of air quality over time after the two difference methods are adopted, and there are differences in the mean value of changes in each month. Beyond that, the IMFs represent different frequency components in the variation of air quality. Thus, the final IMF can be used to measure the long-run trend of these variation differences.

Second, a recursive graph was used to allow visualization of trajectory periodicity through phase space. By drawing a recursive graph, we can study some aspects of the m-dimensional phase space trajectory of the AQI through a two-dimensional representation, which reflects the predictability of air quality. Then, recursive quantitative analysis was used for the quantitative analysis of changes in predictability.

5: 166-167 “?????,?????−19=(????2020−????2019)−(????2019−????2018),?=1,…,348 (1) where ????2020, ????2019 and ????2018 represent the pollutants in 2020…” What does it means “the pollutants in 2020” ? Is it the daily concentration of pollutants (PM10, NO2, etc)?

RESPONSE: Yes, these variables represent the daily concentrations of different pollutants in each of the three years.

6: What does it means ?=1,…,348? Why 348?

RESPONSE: The data for this paper on six kinds of pollutants, including PM2.5, PM10, SO2, NO2, CO and O3, and the air quality index (AQI) of 348 cities from Jan 1, 2018, to Dec 27, 2020, were collected from the Ministry of Ecological Environment of China (The data of February 6, 2019 is missing, and there is no data of February 29, 2019 and 2018. We delete the data of these two days.).

7: 168 - “The results for each city were averaged and recorded as Δ???”. Why AQI? Is the result an average over 328 cities? The result is unclear

RESPONSE: We have revised this introduction:

2.1. Ensemble Empirical Mode Decomposition

For cities experiencing severe epidemic and those with mild epidemic conditions, the difference in AQI between 2020 and 2019 minus the difference in AQI between 2019 and 2018 was evaluated in this paper. The change in trends for these pollutants caused by COVID-19 was calculated by

                (1)

where ,  and  represent the pollutants in 2020, 2019 and 2018, respectively. We averaged the change values of 348 cities to obtain a sequence. This sequence represents the overall change.. Then, the EEMD method was adopted to obtain the long-term impact of COVID-19 on AQI. EEMD could make up the shortcoming of EMD. In the process of decomposing the IMF with EMD, many iterations are required, and the conditions for stopping the iteration lack a standard, so the IMFs obtained by different conditions for stopping the iteration are also different (Qiu et al., 2022). This shortcomings has been solved by EEMD proposed by Wu and Huang (2009) and used in this paper. In addition, EEMD can effectively solve the mode aliasing problem caused by EMD, and it can accurately and efficiently divide different frequency components. Specifically, when EEMD is adopted, white noise is added to the signal to be analyzed by taking advantage of the characteristic of uniform spectrum distribution of white noise, so that signals with different time scales can be automatically separated to the corresponding reference scale, which is the EEMD method. This method mainly adds white noise to the signal to supplement some missing scales, and has a good performance in signal decomposition. The EEMD decomposition principle is: when the additional white noise is evenly distributed in the whole time-frequency space, the time-frequency space is composed of different scale components divided by the filter bank. In this paper, EEMD is adopted to analyze the differences in air pollutant changes. The other IMFs have its own meaning separately, and IMF1, IMF2,…, IMF12 indicate the time scale is getting smaller and smaller according to the calculation formula of IMFs. Since the last IMF is the closest IMF to the long-term trend, it is used in this paper. Specifically, EEMD algorithm steps are as follows:

8: 2.2. State Phase Reconstruct: the method is not clearly explained

RESPONSE: Detailed calculation procedures and formulas have been provided in Section 2.3 and 2.4, including the corresponding meaning of each indicator. According to these formulas, the relevant indexes can be accurately calculated and analyzed.

9: 206 – “By selecting the appropriate time lag factor ?” How the factor ? was determined?

RESPONSE: This parameter is determined by autocorrelation function method. We choose the value corresponding to the minimum value of the autocorrelation function of the original sequence as the lag time.

10: 2.3. Recurrence Plots: the method is not clearly explained

RESPONSE: Detailed calculation procedures and formulas have been provided in Section 2.3 and 2.4, including the corresponding meaning of each indicator. According to these formulas, the relevant indexes can be accurately calculated and analyzed.

11: In this article there is some confusion regarding emissions and concentrations of pollutants. The observations concern the concentration of secondary pollutants (PM10, NO2, etc.)

RESPONSE: Following He et al. (2020), the air quality index (AQI) was selected in this paper to measure the air quality of cities in China. We considered three kinds of cities: cities with severe epidemic conditions, cities with few confirmed cases, and cities experiencing secondary outbreaks (Table 1). The data for this paper on six kinds of pollutants, including PM2.5, PM10, SO2, NO2, CO and O3, and the air quality index (AQI) of 348 cities from Jan 1, 2018, to Dec 27, 2020, were collected from the Ministry of Ecological Environment of China (The data of February 6, 2019 is missing, and there is no data of February 29, 2019 and 2018. We delete the data of these two days.). Detailed pollutant units and AQI calculation procedures are provided in the supplementary material.

12: 303 - 306  “…the emissions of CO, NO2, PM10 and PM2.5  were reduced, but the emissions of SO2 were elevated. Considering that O3 is secondary pollutant and its emissions…” The conclusions and a large part of the article show a lack of knowledge regarding air pollution. It would be necessary to completely rewrite the article.

RESPONSE: We have provided authoritative sources of data on air pollutants and detailed formulas for the AQI. On the question of ozone, different from air pollutants such as NOx and PM 2.5 directly emitted by human activities, surface ozone is a secondary pollutant generated by a variety of air pollutants through complex chemical reactions, so the impact of environmental changes on ozone pollution will be more difficult to predict. This article is an extended study based on He et al.(2020), and also does not carry out chemical analysis and experiments on the detailed evolution mechanism of pollutants. Part of the conclusion is a speculation from the economic perspective.

Best Regards!

Yours sincerely!

Reviewer 4 Report (New Reviewer)

1. It is recommended that the methodology section, some methodologies, be placed in the supplementary document. The methodology section should only outline the required methods and tools, and the specific presentation should be placed in the supplementary document.

2. Why do the pictures not appear in the order in the text?

3. The ones shown in Table 2, Figure 2, and Figure S1 have a repetitive nature.

4. Does the forecasting model take into account the changes in the epidemic and the fact that the closure policy will always be adjusted, which is very different from the initial situation at the beginning of the epidemic. Do the predictive models developed by the authors predict these changes well?

5. Restricting residential travel and reducing industrial production during an epidemic both benefit air quality. But epidemics are, after all, transient, and activity cannot always cease. Once activity continues, air quality is bound to decline. I think the importance of this study is not very convincing.

6. The author can get the predicted trend of air quality changes and which pollutants are heavily polluted, making the corresponding rationalization more realistic.

Author Response

We would like to thank the editor and reviewers for their careful and thorough reading of this manuscript and for the thoughtful comments and constructive suggestions, which have helped us improve the manuscript and reach a better scientific level. All of the comments are valuable and have been very helpful during the revision process and have provided important guiding significance to our research. We have carefully revised the manuscript entitled “Changes in air quality during the period of COVID-19 in China” (ijerph-1926617) according to the editor and reviewers’ comments. The responses to the reviewers’ comments and the revised portions are reported below each question and are marked in blue and red, respectively. We tried our best to improve the manuscript by making the suggested changes. These changes did not influence the content and framework of the paper. We appreciate the editor’s and reviewers’ thoughtful work earnestly and hope that the corrections will meet with your approval. Thank you for the time and effort that was expended on this paper. The responses to the reviewers’ and editor’s comments are as follows:

Response to the Comments of Reviewer 4

Reviewer’s comments:

1: It is recommended that the methodology section, some methodologies, be placed in the supplementary document. The methodology section should only outline the required methods and tools, and the specific presentation should be placed in the supplementary document.

RESPONSE: Thank you for your suggestion and we have included a detailed description of the methods section in the supplementary material. The detailed revisions related this comment are as follows:

The explanation of EEMD and CEEMDAN

EEMD could make up the shortcoming of EMD. In the process of decomposing the IMF with EMD, many iterations are required, and the conditions for stopping the iteration lack a standard, so the IMFs obtained by different conditions for stopping the iteration are also different (Qiu et al., 2022). This shortcomings has been solved by EEMD proposed by Wu and Huang (2009) and used in this paper. In addition, EEMD can effectively solve the mode aliasing problem caused by EMD, and it can accurately and efficiently divide different frequency components. Specifically, when EEMD is adopted, white noise is added to the signal to be analyzed by taking advantage of the characteristic of uniform spectrum distribution of white noise, so that signals with different time scales can be automatically separated to the corresponding reference scale, which is the EEMD method. This method mainly adds white noise to the signal to supplement some missing scales, and has a good performance in signal decomposition. The EEMD decomposition principle is: when the additional white noise is evenly distributed in the whole time-frequency space, the time-frequency space is composed of different scale components divided by the filter bank.

CEEMDAN can further solve the problem of mode aliasing. Firstly, the IMF component with auxiliary noise after EMD decomposition is added in this method, instead of adding Gaussian white noise signal directly to the original signal. Secondly, EEMD decomposition take the overall average of the modal components obtained after the empirical mode decomposition, while CEEMDAN decomposition takes the overall average calculation after the first order IMF component, and obtains the final first order IMF component. Then, the above operations are repeated for the residual parts. In this way, the transfer of white noise from high frequency to low frequency is effectively solved.

  1. Why do the pictures not appear in the order in the text?

RESPONSE: We put the results of the robustness test, the original data and the calculation results of the AQI index at the end of the supplementary materials to supplement the empirical results in the whole paper.

  1. The ones shown in Table 2, Figure 2, and Figure S1 have a repetitive nature.

RESPONSE: Figure 2, Table 2, and the figures in the supplementary material present the results of our analysis in different forms, making it easy for the reader to clearly see changes in trends and compare them.

  1. Does the forecasting model take into account the changes in the epidemic and the fact that the closure policy will always be adjusted, which is very different from the initial situation at the beginning of the epidemic. Do the predictive models developed by the authors predict these changes well?

RESPONSE: This is a very insightful question, and the fact that during an outbreak, the government intensifies or weakens lockdown measures in response to the status of the epidemic, the impact of this dynamic measure cannot yet be captured by our model. This is also our next research direction, that is, after measuring the dynamic change level of lockdown policy, the series will be combined with air pollution data to study its dynamic impact on air pollution.

  1. Restricting residential travel and reducing industrial production during an epidemic both benefit air quality. But epidemics are, after all, transient, and activity cannot always cease. Once activity continues, air quality is bound to decline. I think the importance of this study is not very convincing.

RESPONSE: This is an issue that deserves further discussion, especially to compare and analyze the impact of different national pandemic policies on air quality. The Chinese government has attached great importance to the epidemic, and the quarantine and lockdown policies will continue for a long time even after the epidemic stabilizes.

  1. The author can get the predicted trend of air quality changes and which pollutants are heavily polluted, making the corresponding rationalization more realistic.

RESPONSE: This is a very insightful question. The main purpose of the recursive quantitative analysis method adopted in this manuscript is to analyze and evaluate whether the epidemic will affect the predictability of air quality. There is no further use of time series models to predict different pollutants, such as ARMA model. Considering that there may be the omission of variables in the process of using econometric model for forecasting, the accuracy of forecasting results may be affected.

We find that the reviewer’s comments have been quite helpful in improving the paper, and we have revised our paper point by point. Once again, thank you very much for your comments and suggestions!

Best Regards!

Yours sincerely!

Reviewer 5 Report (New Reviewer)

Comments:

This paper investigates the evolution of levels of different pollutants during the periods before and after the COVID-19 pandemic.The author used different methods to analyze the improvement of six air pollutant concentrations in different types of cities in China from 2018 to 2020 and ensemble empirical mode decomposition (EEMD). In particular, the authors used recursive plots (RPs) and recursive quantitative analysis(RQA) to explore whether the air quality during the epidemic was more difficult to predict. The experimental results show that,the negative impact of urban residentsliving activities on urban ambient air quality may be weakened, and air quality could be improved.

Weakness:

1.The experimental results may be accidental:”Cities with a second outbreak” is only five cities, with less data, so the universality is questionable;

2.Ablation experiment could be considered to make each part of the proposed scheme meaningful and necessary:adding experiments to validate that EEMD could make up the shortcoming of EMD and provide more accurate long time scale information about the air quality;

3.Insufficient innovation of experimental methods: the author adopted EEMD to extract the last IMF values of six pollutants and AQI during the COVID-19 pandemic.This method is a time-series analysis technology invented in 2009,and there have been some improvements to the method. The author did not improve it, and now simply using EEMD to extract the IMF value is too old, or can also add experiments to prove the effectiveness of the method;

4.In Sec.3.1, since the last IMF is the closest IMF to the long-term trend, it was used to perform a sensitivity analysis to the performance.Please give the IMF selection process,and whether the selection is universal and representative.

5.The language can be further improved. Many of the expressions in the paper are colloquial. Also,The five images in the paper are positioned to the left and need to be re-typeset.

Author Response

We would like to thank the editor and reviewers for their careful and thorough reading of this manuscript and for the thoughtful comments and constructive suggestions, which have helped us improve the manuscript and reach a better scientific level. All of the comments are valuable and have been very helpful during the revision process and have provided important guiding significance to our research. We have carefully revised the manuscript entitled “Changes in air quality during the period of COVID-19 in China” (ijerph-1926617) according to the editor and reviewers’ comments. The responses to the reviewers’ comments and the revised portions are reported below each question and are marked in blue and red, respectively. We tried our best to improve the manuscript by making the suggested changes. These changes did not influence the content and framework of the paper. We appreciate the editor’s and reviewers’ thoughtful work earnestly and hope that the corrections will meet with your approval. Thank you for the time and effort that was expended on this paper. The responses to the reviewers’ and editor’s comments are as follows:

Response to the Comments of Reviewer 5

Reviewer’s comments:

1: The experimental results may be accidental: ”Cities with a second outbreak” is only five cities, with less data, so the universality is questionable;

RESPONSE: And this is something that we've talked about a lot before. Due to the small number of cities with secondary outbreaks in the early stage of the outbreak, only a few cities such as Dalian were sampled. Although the samples in the future increased, they were already in the stage of repeated fluctuation of the epidemic, so they were not considered.

  1. Ablation experiment could be considered to make each part of the proposed scheme meaningful and necessary:adding experiments to validate that EEMD could make up the shortcoming of EMD and provide more accurate long time scale information about the air quality;

RESPONSE: We have provided a section of the robustness test to verify the robustness of the empirical results. The comparison between EEMD and EMD methods has been discussed in detail in Wu and Huang (2009). Beyond that, for a longer time scale, our current study focuses on the impact of the outbreak on air quality within one year. Due to the complexity of the latest epidemic prevention and control policies, we are concerned that it is difficult to clearly distinguish the impact of various recent policies over a longer time scale. However, this is an important research direction for us in the future, that is, to study the lag effect of different prevention and control policies on a longer time scale.

3: Insufficient innovation of experimental methods: the author adopted EEMD to extract the last IMF values of six pollutants and AQI during the COVID-19 pandemic. This method is a time-series analysis technology invented in 2009, and there have been some improvements to the method. The author did not improve it, and now simply using EEMD to extract the IMF value is too old, or can also add experiments to prove the effectiveness of the method;

RESPONSE: We have further adopted CCEMDAN for robustness checking. In fact, the use of EEMD can already effectively decompose the unused modes.

  1. In Sec.3.1, since the last IMF is the closest IMF to the long-term trend, it was used to perform a sensitivity analysis to the performance.Please give the IMF selection process,and whether the selection is universal and representative.

RESPONSE: We set the number of iterations in the program by using MATLAB, and when the maximum number of iterations is reached, the low-frequency part of the obtained IMF is used for empirical analysis.

  1. The language can be further improved. Many of the expressions in the paper are colloquial. Also,The five images in the paper are positioned to the left and need to be re-typeset.

RESPONSE: Thank you for your advice. We have rechecked the grammar errors and language, and adjusted the position of the picture.

We find that the reviewer’s comments have been quite helpful in improving the paper, and we have revised our paper point by point. Once again, thank you very much for your comments and suggestions!

Best Regards!

Yours sincerely!

Round 2

Reviewer 2 Report (New Reviewer)

The authors have sufficiently addressed my comments and I think it is presented in a more clear fashion.

Author Response

Dear Reviewer:

We would like to thank you for careful and thorough reading of this manuscript and for the thoughtful comments and constructive suggestions, which have helped us improve the manuscript and reach a better scientific level. All of the comments are valuable and have been very helpful during the revision process and have provided important guiding significance to our research. We have carefully revised the manuscript entitled “Changes in air quality during the period of COVID-19 in China” (ijerph-1926617) according to the editor and reviewers’ comments. The responses to the reviewers’ comments and the revised portions are reported below each question and are marked in blue and red, respectively. We tried our best to improve the manuscript by making the suggested changes. These changes did not influence the content and framework of the paper. We appreciate the editor’s and reviewers’ thoughtful work earnestly and hope that the corrections will meet with your approval. Thank you for the time and effort that was expended on this paper. The responses to the comments are as follows:

Response to the Comments of Reviewer 2

Reviewer’s comments:

1: The authors have sufficiently addressed my comments and I think it is presented in a more clear fashion.

RESPONSE: Thank you for the suggestion. We appreciate the reviewer’ thoughtful work earnestly and thank you for the time and effort that was expended on this paper.

We find that the reviewer’s comments have been quite helpful in improving the paper, and we have revised our paper point by point. Once again, thank you very much for your comments and suggestions!

Best Regards!

Yours sincerely!

Reviewer 3 Report (New Reviewer)

The previous version was rejected for many reasons, in particular due to the low level of the English language, the unclear descriptions of methods and results and also due to the lack of knowledge of air pollution.

Unfortunately, the second version of the article is almost the same, the authors made only a few changes:

-          Abstract: the authors only added two words, “and discuss” (line 10)

-          1. Introduction: he second version has a small new part (line 88- line 95)

-          2. Methods and Data: The authors made only two changes, adding a small part (line 166- line 171)

and making a new explanation (line 186 – line 192).

However, these added parts do not improve understanding of the article. There is a complete lack of an adequate explanation of the dataset used:

What does ?????,?????−19… (line 183)  mean? For example, Poll2020 is the average daily concentration of a given pollutant? Are these differences between average daily concentrations for 365 days? How many cities are there, 228 (line 115) or 248 (line 186)?

The lack of knowledge about air pollution is confirmed.

-          3. Empirical results: the second version of the third chapter has not improved, only a reference has been added (line 299).

-          4. The predictability of the evolution of AQI trends caused by COVID-19: The only difference is a small part (line 477-line 479)

-           5. Conclusions: this part is identical to the previous version

For these reasons the article is rejected again.

Author Response

Dear Reviewer:

We would like to thank for careful and thorough reading of this manuscript and for the thoughtful comments and constructive suggestions, which have helped us improve the manuscript and reach a better scientific level. All of the comments are valuable and have been very helpful during the revision process and have provided important guiding significance to our research. We have carefully revised the manuscript entitled “Changes in air quality during the period of COVID-19 in China” (ijerph-1926617) according to the editor and reviewers’ comments. The responses to the reviewers’ comments and the revised portions are reported below each question and are marked in blue and red, respectively. We tried our best to improve the manuscript by making the suggested changes. These changes did not influence the content and framework of the paper. We appreciate the editor’s and reviewers’ thoughtful work earnestly and hope that the corrections will meet with your approval. Thank you for the time and effort that was expended on this paper. The responses to the reviewer’s comments are as follows:

Response to the Comments of Reviewer 3

Reviewer’s comments:

1: The previous version was rejected for many reasons, in particular due to the low level of the English language, the unclear descriptions of methods and results and also due to the lack of knowledge of air pollution. Unfortunately, the second version of the article is almost the same, the authors made only a few changes: Abstract: the authors only added two words, “and discuss” (line 10). 1. Introduction: he second version has a small new part (line 88- line 95). 2. Methods and Data: The authors made only two changes, adding a small part (line 166- line 171) and making a new explanation (line 186 – line 192). However, these added parts do not improve understanding of the article.

RESPONSE: We have rewritten abstract, introduction and conclusion. We have also provided detailed data sources and calculation formulas, including modal decomposition methods and detailed calculation formulas for recursive quantitative analysis indicators to improve the reader’s understanding of the paper. We also invited an expert in this field to conduct a detailed examination of the method, and consulted a professional organization to check the possible grammatical problems in the article. The detailed revisions related this comment are as follows:

Abstract: This paper revisits the heterogeneous impacts of COVID-19 on air quality. For different types of Chinese cities, we analyze the different degrees of improvement in the concentrations of six air pollutants (PM2.5, PM10, SO2, NO2, CO and O3) during COVID-19 by analyzing the predictivity of air quality. Specifically, we divided the sample into three groups: cities with severe outbreaks, few confirmed cases and secondary outbreaks. Ensemble empirical mode decomposition (EEMD), recursive plots (RPs) and recursive quantitative analysis (RQA) are used to analyze these heterogeneous impacts and the predictivity of air quality. The empirical results indicate that (1) COVID-19 does not necessarily improve air quality due to factors such as the rebound effect of consumption, and its impacts on air quality is short-lived. After the initial outbreak, NO2, CO and PM2.5 emissions declined for the first 1-3 months; (2) for the cities with severe epidemic, air quality was improved, but for the cities with second outbreak, air quality was first enhanced and then deteriorated. For the cities with few confirmed cases, air quality was first deteriorated and then improved. (3) COVID-19 changed the stability of the air quality sequence. The predictability of the AQI declined in cities with serious epidemic situations and secondary outbreaks, but for the cities with few confirmed cases, the AQI achieved a stable state sooner. The conclusions may facilitate the analysis of differences in air quality evolution characteristics and fluctuations before and after outbreaks from a quantitative perspective.

  1. Introduction

The outbreak of COVID-19 had significant impacts on the lives of people and the operation of economy [1-6]. Governments implemented various measures to prevent its spread, such as controls on travel [7], case isolation [8] and home quarantine [9-12]. These policies led to significant reductions in travel and industrial products [13] and affected air quality. However, its impacts on air quality could be heterogeneous for Chinese cities with different severity of the epidemic. For the cities with severe outbreaks, the outbreak and strict lockdown measures could significantly improve air quality, but for cities less severely affected, the lifting of initial lockdown measures could lead to a rebound in consumption and production, worsening air quality. Furthermore, some cities experienced a second outbreak of COVID-19, and the change of air quality presented a more complex picture. At present, there is no detailed empirical research on these differences. Beyond that, the economy of Chinese cities is highly connected. The outbreak and lockdown measures in one area may affect production activities in the surrounding areas and thus affect the air quality in the surrounding areas. Hence, this paper aims to evaluate and compare the heterogeneous impacts of different levels of epidemic outbreaks on air quality and the changes of the predictability and stability of air quality during this period. This could provide a new perspective from which to study changes in air quality during the outbreak resulting from these regulations [14-16] and the influence of measures for curbing the spread of COVID-19 on these changes and the improvement of environmental quality [17-19].

Most current studies believed that COVID-19 improved air quality [20-22]. As one of the countries with the strictest control over the epidemic, China has received more attention. [23]. An existing study analyzed changes in air quality in Shanghai, Wuhan and Tangshan before and after the outbreak and found that COVID-19 control measures improved air quality [24]. When variables such as temperature were included, the AQI was 12.2 percent lower in cities with lockdown policies [25]. The difference in differences (DID) model was used to estimate the impact of the COVID-19 pandemic on air quality and a significant improvement in air quality during the COVID-19 pandemic was confirmed [26]. An existing research uses a quasi-difference-in-differences analysis to investigate the influence of lockdown measures during the COVID-19 outbreak on air pollution in 357 Chinese cities and found that air quality improved significantly during the first outbreak [27]. The literature has also pointed out that energy use and carbon dioxide emissions have decreased during the COVID-19 pandemic, and this negative influence was more significant than that observed during previous economic downturns [18]. It has also been found that the reduction of emissions has varied according to the measures taken by each country to restrict movement. Based on indicators of transportation and power generation, air quality was significantly improved in China due to the pandemic in 2020 compared to that during the same period in 2019 [28].

However, as mentioned above, COVID-19 could have heterogeneous impacts on air quality and its predictability and stability, which has not been considered in previous literature [28]. The autocorrelation of the air quality sequence in these cities makes it predictable. Under normal condition, the time series of air quality has strong predictability, but the outbreak of the epidemic could cause significant effects and new information, breaking the original sequence pattern, and the predictability is reduced. In addition, the predictability of the air quality sequence was more affected by the two outbreaks. Therefore, we hold that the analysis of predictability can help to depict the heterogeneous effects of COVID-19 on air quality. This can be visually represented by RP diagrams. Based on this method, some quantitative indicators can be used to further quantify changes in predictability.

Beyond that, there are complex industrial and administrative relationships among Chinese cities. Economic development processes will be linked through various channels, COVID-19 lockdown measures will be spillover, and air quality change patterns will be more complex. The effects of the epidemic on air quality are also time-dependent. With the gradual increase in confirmed cases, residents’ attention to the epidemic and their travel demands has been further affected, which has also influenced air quality. Even if restrictions on movement are lifted, behavioral inertia and worry regarding the epidemic will alter the lives of residents and travel patterns over a certain period. Due to the rapid economic recovery in some regions, emissions will rebound, and air quality may be decreased. The above factors, including repeated outbreaks of COVID-19, sudden lockdown measures and gradual resumption of production and work, can cause short-term perturbations in air quality data, which may lead to deviations in studies on the long-term impact of COVID-19 on air quality. Therefor, it is also important to discuss the long-term change of air quality without the interference of short-term redundant information.

To be specific, this paper considered three types of Chinese cities under different epidemic levels (severe epidemics, less severe epidemics and secondary outbreaks). We first focuses on the evolution of air quality during surges and stable periods of the epidemic in these cities by adopting the EEMD method. The empirical process is as follows: (1) Based on data availability, this paper analyzes and compares trends in six pollutants (PM2.5, PM10, SO2, NO2, CO and O3) in 348 cities between 2018-2019 and 2019-2020. (2) We selected 12 cities that experienced severe epidemic conditions in Hubei Province, 24 cities with few confirmed cases from other provinces and 5 cities with secondary outbreak. The AQI is used to measure air quality from Jan 1, 2018, to Dec 27, 2020. This article adopts this method mainly considering the following aspects: the characteristics of air pollutant data, the purpose of the research and the advantages of the methods. Specifically, the data form of air pollutants is nonstationary daily time series data, and its sample size is greater than 300 for each year, meeting the data requirements of the EEMD method, which is suitable for nonstationary signal analysis. This paper aims to analyze the long-term change characteristics of air pollutants, and EEMD can effectively extract this characteristic. The results of EMD may be aliasing, resulting in unclear distinction of components in different time scales. The EEMD used in this paper could make up for the shortcoming of EMD and provide more accurate long time scale information about the air quality.

Then, this paper adopts recurrence plots (RPs) and recurrence quantification analysis (RQA) to compare the characteristics of fluctuations in AQI before and after outbreaks, which could be used to describe the effect of the COVID-19 pandemic on the predictability of urban air quality. Specifically, in this paper, we first selected the three classes of cities mentioned before: cities experiencing severe epidemics, cities with mild epidemic conditions, and cities experiencing secondary outbreaks. The recurrence plots of the mean values of AQI in these cities are reported to discuss the characteristics of fluctuation in AQI in these different kinds of cities. The recurrence plots and recurrence quantification analysis for the mean AQI values in 2018, 2019 and 2020 are listed for comparison. The empirical results prove that the impact of epidemic outbreak on air quality is different in different cities. Air quality was significantly improved in severely affected areas, while the improvement effect was not significant in less severely affected areas. In addition, in cities with severe COVID-19, new information generated by outbreaks has led to a decrease in the predictability of air quality.

Compared with the existing literature, there are two main innovations in this paper. First, differing from the existing literature, which has discussed the different levels of air quality before and after the outbreak and investigated the impact of COVID-19 lockdowns on air quality, this paper focuses on the evolution of air quality during surges and stable periods of the epidemic for areas experiencing severe and mild epidemic conditions by adopting ensemble empirical mode decomposition (EEMD). We confirmed that the long-term impact of the epidemic on air quality was complex and heterogeneous. Whether the impact of COVID-19 on AQI will be weakened with future outbreaks and stabilization of the epidemic also attracted our interest. Hence, we further selected 5 cities, Harbin, Dalian, Urumqi, Chengdu and Beijing, for investigation of this point and to compare changes in air quality before and after secondary outbreaks. The epidemic was controlled after the outbreak in these cities, but new confirmed cases were detected again a few months after the initial confirmed cases no longer increased, and the blockade measures in these cities were further strengthened.

Second, to our knowledge, whether the COVID-19 pandemic could affect the predictability of urban air quality in different cities has not been addressed in previous literature. Therefore, this paper adopts recurrence plots (RPs) and recurrence quantification analysis (RQA) to compare the characteristics of fluctuations in AQI before and after outbreaks, which could be used to describe the effect of the COVID-19 pandemic on the predictability of urban air quality visually and quantitatively. It is found that for cities under severe pandemic, air quality changes were more dramatic and predictability was significantly reduced. Comparatively, the air quality sequence of cities with less severe epidemic was more stable. For the cities with the second outbreak, the negative impact of the second outbreak on the stability of the sequence was weakened. This may help to accurately predict urban air quality, facilitate the analysis of differences in air quality evolution characteristics and fluctuations before and after outbreaks from a quantitative perspective and formulate air control measures in advance.

  1. There is a complete lack of an adequate explanation of the dataset used.

RESPONSE: We have provided detailed data sources and calculation formulas, including modal decomposition methods and detailed calculation formulas for recursive quantitative analysis indicators in Section 2. The parameters in all formulas are explained in detail.

  1. What does ?????,?????−19… (line 183) mean? For example, Poll2020 is the average daily concentration of a given pollutant? Are these differences between average daily concentrations for 365 days? How many cities are there, 228 (line 115) or 248 (line 186)?

RESPONSE: Polli,Covid-19 refers to the changes in the concentration of pollutants in city i caused by the outbreak. It is daily data, and the sequence is obtained by quadratic difference. We are sorry for this error. The sample size is 348, not 328.

  1. The lack of knowledge about air pollution is confirmed.

RESPONSE: We have provided detailed data sources and analysis about air pollution. A detailed AQI measurement method is also reported in the supplementary material. In Section 2. The parameters in all formulas are explained in detail.

  1. Empirical results: the second version of the third chapter has not improved, only a reference has been added (line 299).

RESPONSE: We have rewritten our manuscript according to your previous requirements, including the Abstract, Section 1 introduction and Section 2 2. Methods and Data, and explained the methods section in detail in Section 2.

  1. The predictability of the evolution of AQI trends caused by COVID-19: The only difference is a small part (line 477-line 479)

RESPONSE: We have rewritten Section 4. We simplified and supplemented the graph, and expounded the advantages of this method in advance in the introduction.

  1. Conclusions: this part is identical to the previous version

RESPONSE: On the premise of ensuring the logical structure of the paper, we rewrote the first paragraph of the conclusion and checked the rest parts.

Best Regards!

Yours sincerely!

Reviewer 4 Report (New Reviewer)

I still think the article would be meaningless if the projections made in this article could not be based on existing changes and time series changes. Because forecasting has to be temporal, it is meaningless to use data from two years ago to build a model that cannot be applied to existing situations. This is what is most important about this article.

Author Response

Dear Reviewer:

We would like to thank the editor and reviewers for their careful and thorough reading of this manuscript and for the thoughtful comments and constructive suggestions, which have helped us improve the manuscript and reach a better scientific level. All of the comments are valuable and have been very helpful during the revision process and have provided important guiding significance to our research. We have carefully revised the manuscript entitled “Changes in air quality during the period of COVID-19 in China” (ijerph-1926617) according to the editor and reviewers’ comments. The responses to the reviewers’ comments and the revised portions are reported below each question and are marked in blue and red, respectively. We tried our best to improve the manuscript by making the suggested changes. These changes did not influence the content and framework of the paper. We appreciate the editor’s and reviewers’ thoughtful work earnestly and hope that the corrections will meet with your approval. Thank you for the time and effort that was expended on this paper. The responses to the reviewer’s comments are as follows:

Response to the Comments of Reviewer 4

Reviewer’s comments:

1: I still think the article would be meaningless if the projections made in this article could not be based on existing changes and time series changes. Because forecasting has to be temporal, it is meaningless to use data from two years ago to build a model that cannot be applied to existing situations. This is what is most important about this article.

RESPONSE: This is a very insightful question. We do not estimate and predict the time series of air quality. The most important conclusion we found is that the epidemic does not necessarily improve air quality, but has a heterogeneous effect on air quality. In fact, this paper revisited the impact of COVID-19 on air quality in China. We not only investigated the changes in six air pollutants before and after COVID-19 outbreaks in 348 cities in China by extracting the differences in the growth trends between 2018-2019 and 2019-2020, but also applied recurrence plots and recursive quantitative analysis to compare the changes in predictability of AQI. It was confirmed that due to the initial outbreak and restrictive mitigation policies, the concentrations of different pollutants changed, and the predictability of air quality was also affected. Interestingly, we found a phenomenon that differs from the existing literature, namely that the air quality in the outbreak area does not necessarily improve, and it is possible that the air quality deteriorates with the reduction of the severity of the outbreak and the exposure to lockdown policies. This may be due to the rebound effect of production and consumption as well as the indirect industrial correlation effect between adjacent regions. Of course, the latest outbreak scenario is more complicated, largely because of changing policies and changing attitudes. This is also our next focus of research. To sum up, this paper aims to evaluate and compare the heterogeneous impacts of different levels of epidemic outbreaks on air quality and the changes of the predictability and stability of air quality during this period. This could provide a new perspective from which to study changes in air quality during the outbreak resulting from these regulations and the influence of measures for curbing the spread of COVID-19 on these changes and the improvement of environmental quality.

We find that the reviewer’s comments have been quite helpful in improving the paper, and we have revised our paper point by point. Once again, thank you very much for your comments and suggestions!

Best Regards!

Yours sincerely!

This manuscript is a resubmission of an earlier submission. The following is a list of the peer review reports and author responses from that submission.

Round 1

Reviewer 1 Report

This paper focuses the various impact of the severity of COVID-19 development on air quality in different types of cities. The manuscript is well-organized. Some suggestions:

1. lines 313-314. the time trends of the Monthly city-wide mean value for different primary pollutants is reported in Appendix, Fig. S13. 

Fig. S13?

2. In this paper, the year 2020 is divided into different phases (COVID-19 outbreak period, strict closure policy period, and stabilization period) to discuss the impact of the COVID-19 pandemic on air quality. The authors should clearly describe the basis for the division of the different phases. The temporal division of these phases is not uniform in the text, e.g., Fig. 2, Fig. S2, Fig. S4, line307, line340 and lines365-367. Why are the temporal divisions of these stages not uniform?

3. The paper discussed the impact of the COVID-19 pandemic on air quality mainly from the perspective of residential consumption. Is the resumption of production and work an important influencing factor compared to residential consumption?

4.  With respect to the methods, the paper should extensively discuss the advantages of the difference method and the ensemble empirical mode decomposition (EEMD) over other methods and verify the feasibility and effectiveness of these two methods. Especially, the difference method.

Author Response

We would like to thank the editor and reviewers for their careful and thorough reading of this manuscript and for the thoughtful comments and constructive suggestions, which have helped us improve the manuscript and reach a better scientific level. All of the comments are valuable and have been very helpful during the revision process and have provided important guiding significance to our research. We have carefully revised the manuscript entitled “Improvement and predictability of urban air quality under different stages of the COVID-19 pandemic” (ijerph-1849181) according to the editor and reviewers’ comments. The responses to the reviewers’ comments and the revised portions are reported below each question and are marked in blue and red, respectively. We tried our best to improve the manuscript by making the suggested changes. These changes did not influence the content and framework of the paper. We appreciate the editor’s and reviewers’ thoughtful work earnestly and hope that the corrections will meet with your approval. Thank you for the time and effort that was expended on this paper. The responses to the reviewers’ and editor’s comments are as follows:

Response to the Comments of Reviewer 1

Reviewer’s comments:

1: This paper focuses the various impact of the severity of COVID-19 development on air quality in different types of cities. The manuscript is well-organized. Some suggestions: 1. lines 313-314. the time trends of the Monthly city-wide mean value for different primary pollutants is reported in Appendix, Fig. S13. Fig. S13?

RESPONSE: We apologize for reporting the wrong name of Figure. Fig. S13 here should be Fig. S14. We have also checked all figures in Supplementary Materials and replaced ‘Appendix’ with ‘Supplementary Materials’ in the manuscript. The detailed revisions related this comment are as follows:

Fig. S14. Time trends of the Monthly city-wide mean value for different primary pollutants

2: In this paper, the year 2020 is divided into different phases (COVID-19 outbreak period, strict closure policy period, and stabilization period) to discuss the impact of the COVID-19 pandemic on air quality. The authors should clearly describe the basis for the division of the different phases. The temporal division of these phases is not uniform in the text, e.g., Fig. 2, Fig. S2, Fig. S4, line307, line340 and lines365-367. Why are the temporal divisions of these stages not uniform?

RESPONSE: Thank you for the suggestion. We first divided the city types into three categories. We measured it based on whether there was a second change in the number of confirmed COVID-19 cases reported in each city. But in time, as a result of each city's policy is not completely consistent, for example, some cities because of the outbreak is relatively serious, therefore introduced in March or so traffic blockade measures file, and some cities even delayed until April to release in May, so we can not on time scales, strictly divided all cities, But the traffic lockdown lasts roughly from March to around July.

3: The paper discussed the impact of the COVID-19 pandemic on air quality mainly from the perspective of residential consumption. Is the resumption of production and work an important influencing factor compared to residential consumption?

RESPONSE: This is a very insightful question. In fact, compared with the retaliatory consumption of residents, the retaliatory pollution caused by China's resumption of work and production may be more noteworthy. However, due to the time lag in the production of enterprises, the government's deregulation policy will not immediately cause large-scale enterprise production behavior, so the degree and stage of resumption of work cannot be accurately identified. It is also difficult to measure the degree of resumption of work of enterprises in each city. The impact of the resumption of work of productive enterprises on the environment will also be our focus in the next step.

4: With respect to the methods, the paper should extensively discuss the advantages of the difference method and the ensemble empirical mode decomposition (EEMD) over other methods and verify the feasibility and effectiveness of these two methods. Especially, the difference method.

RESPONSE: Thank you for your suggestions. We have added explanations on the applicability of these two methods to the research topic of this paper. Specifically, this paper first uses the difference method to calculate the growth of air pollutants before and after the outbreak of the epidemic, and then uses the signal decomposition method to decompose its components. The difference method is not a DID model, which does not involve explanatory variables and explained variables. This method is only a simple processing of the time series data of air pollutants. Other reviewers also proposed the comparison between EMD method and CEEMDAN method. EEMD algorithm reduces the modal effect of EMD by adding noise, and CEEMDAN algorithm further reduces the modal effect by adding adaptive noise, and has better convergence. This paper does not involve the method innovation, and the EEMD method can effectively obtain the results we need. EEMD is different from FFT or Wavelet method and is more suitable for the data in this paper because it is data adaptive and does not need to consider the stationarity of data, which could could avoid the result error caused by improper selection of basis function. The detailed revisions related this comment are as follows:

  1. Methods and Data

       The empirical study is divided into two parts. First, in Part 1 of Fig. 1, the second-order difference method was used to calculate the differences in air pollutant changes, and then the EEMD method was adopted to obtain and analyze the long-term trends in these differences. For the feasibility of empirical research, it is assumed that for each city, the local government's annual efforts to control air pollution are relatively close, and the difference can eliminate policy human factors to a certain extent. In fact, this paper first uses the difference method to calculate the growth of air pollutants before and after the outbreak of the epidemic, and then uses the signal decomposition method to decompose its components. The difference method is not a DID model, which does not involve explanatory variables and explained variables. Second, a recursive graph was used to allow visualization of trajectory periodicity through phase space. By drawing a recursive graph, we can study some aspects of the m-dimensional phase space trajectory of the AQI through a two-dimensional representation, which reflects the predictability of air quality. Then, recursive quantitative analysis was used for the quantitative analysis of changes in predictability. The specific steps are shown in Fig. 1.

We find that the reviewer’s comments have been quite helpful in improving the paper, and we have revised our paper point by point. Once again, thank you very much for your comments and suggestions!

Best Regards!

Yours sincerely!

Reviewer 2 Report

Comment 1:

What is the novelty of models or methods brought to the article? For readers to quickly catch your contribution, it would be better to highlight your original achievements in a clearer way in abstract and introduction and list more numerical results as well to make abstract quantitative.

Comment 2:

The research object of the paper is limited to China. However, the impact of the COVID-19 is global, and the title needs to be revised to be consistent with the text. At the same time, only the data up to 2020 is used, which lacks timeliness.

Comment 3:

The selection of the samples is unreasonable, and more explanation is needed to tell us why those cities in the manuscript were chosen instead of other cities with the same characteristics, and why the sample sizes of the data in 2018, 2019 and 2020 happened to be all the same after removing missing values? .

Comment 4:

The second-order difference method is used as a method to calculate the differences in air pollutant changes so as to reflect the impact of the epidemic on air quality. However, air quality does not necessarily have an annual cycle, and this place needs more explanations to enhance rationality. Besides, it seems to be debatable that the second-order difference is denoted as DID. The exogenous shocks do not occur at the same time because the selected cities are affected by the epidemic at different points of time. Strictly speaking, there is no control group, and section Empirical Results needs to be modified

Comment 5:

In the text, the EEMD method is applied to make up for the shortcomings of EMD. So why not use CEEMD, CEEMDAN, IEEMDAN and other signal decomposition methods? On the other hand, air quality data is not high-frequency data, can traditional time domain analysis methods be used?

Comment 6:

The selection of IMF1 needs to be explained, why not select more components? Meanwhile, as is often the case, IMF1 is a high-frequency component and cannot represent a long-term trend.

Comment 7:

Is it biased to use the stability of the system to characterize the predictability of AQI? And is RQA indicators more suitable than other indicators such as Permutation Etropy, Sample Entropy and Multiscale Entropy in measuring the stability of time series?

Comment 8:

In the paper, by using charts for comparisons, the conclusion is drawn that the system stability of cities that are not severely affected by the epidemic has improved. Nevertheless, the research may need to make the sample size larger and use statistical hypothesis-testing to determine whether there are statistical differences.

Comment 9:

Figure 1 is somewhat confusing in expressing the technology roadmap and needs to be further revised. The bar graph in Figure 4 is too dense to be legible and can be devided into several more lines.

Comment 10:

Is the interpretation of Figure 5 wrong? Maybe there isn’t any evidence of a steady upward trend in the mean values of DET, LMAX, ENT, and TT.

Author Response

We would like to thank the editor and reviewers for their careful and thorough reading of this manuscript and for the thoughtful comments and constructive suggestions, which have helped us improve the manuscript and reach a better scientific level. All of the comments are valuable and have been very helpful during the revision process and have provided important guiding significance to our research. We have carefully revised the manuscript entitled “Improvement and predictability of urban air quality under different stages of the COVID-19 pandemic” (ijerph-1849181) according to the editor and reviewers’ comments. The responses to the reviewers’ comments and the revised portions are reported below each question and are marked in blue and red, respectively. We tried our best to improve the manuscript by making the suggested changes. These changes did not influence the content and framework of the paper. We appreciate the editor’s and reviewers’ thoughtful work earnestly and hope that the corrections will meet with your approval. Thank you for the time and effort that was expended on this paper. The responses to the reviewers’ and editor’s comments are as follows:

Response to the Comments of Reviewer 2

Reviewer’s comments:

1: What is the novelty of models or methods brought to the article? For readers to quickly catch your contribution, it would be better to highlight your original achievements in a clearer way in abstract and introduction and list more numerical results as well to make abstract quantitative.

RESPONSE: Thank you for your suggestions. This paper does not involve the innovation of signal analysis and other methods. In fact, differing from the existing literature, which discussed the different levels of air quality before and after the outbreak or investigated the impact of COVID-19 lockdowns on air quality, this paper focuses on the evolution of air quality during surges and stable periods of the epidemic for areas experiencing severe and mild epidemic conditions by adopting the ensemble empirical mode decomposition (EEMD). To our knowledge, there is no literature to discuss the change of air quality under the outbreak from the perspective of signal analysis. According to your suggestion, we tried to highlight our original achievements in a clearer way in abstract and introduction and list more numerical results. The detailed revisions related this comment are as follows:

Abstract: This paper investigates the various impact of COVID-19 began in January 2020 on air quality in different types of cities. We analyze the different degrees of improvement of concentrations of six air pollutants (PM2.5, PM10, SO2, NO2, CO and O3) obtained from the Ministry of Ecology and Environment, PRC in different types of Chinese cities from 2018 to 2020 with difference method (the trend of pollutants in different years was differenced) and ensemble empirical mode decomposition (EEMD), and then adopt the recursive plots (RPs) and recursive quantitative analysis (RQA) for the first time to discuss whether air quality is more difficult to predict during the outbreak. The empirical results indicate that: (1) After the initial outbreak, only the emissions of NO2, CO and PM2.5 declined for the first 1-3 months, and during the fourth to fifth months the emissions of six air pollutants were elevated in most cities; (2) For the cities with serious epidemic situations in the Hubei province (China), the air quality is improved, but for the cities experiencing a second outbreak, the air quality was first enhanced and then deteriorated, and the sensitivity of air quality to COVID-19 re-outbreak is decreasing. For Dalian, Harbin and Urumqi, AQI is improved by approximately 12.91, 6.28 and 0.36, respectively. (3) In comparison, the predictability of AQI has declined in cities with serious epidemic situations in Hubei, but AQI achieves a stable state sooner in cities with mild epidemic. The conclusions may facilitate analysis of differences in air quality evolution characteristics and fluctuations before and after outbreaks from a quantitative perspective.

2: The research object of the paper is limited to China. However, the impact of the COVID-19 is global, and the title needs to be revised to be consistent with the text. At the same time, only the data up to 2020 is used, which lacks timeliness.

RESPONSE: Thank you for your suggestions. We have revised the title to ensure consistency with the text (Improvement and predictability of urban air quality under different stages of the COVID-19 pandemic in China). The numbers of confirmed cases of COVID-19 in cities with severe epidemic conditions, with few confirmed cases and experiencing secondary outbreaks were obtained from the Wind database. For our laboratory, the right to use the data has expired. In order to ensure the reliability of the data, and considering that the resumption activities of many enterprises have started at the end of 2020, which may have a more significant potential impact on air pollution, the data is only updated until 2020. The focus of this paper is to study the impact on air pollutants in the months after the outbreak of the epidemic. Thank you very much for your suggestions. Our next work is also to study the delayed impact of the blockade policy on the resumption of work of enterprises, and further update the data

3: The selection of the samples is unreasonable, and more explanation is needed to tell us why those cities in the manuscript were chosen instead of other cities with the same characteristics, and why the sample sizes of the data in 2018, 2019 and 2020 happened to be all the same after removing missing values?

RESPONSE: Thank you for your suggestions. We have added more explanations in the data section. In fact, we chose these cities mainly because, compared with other Chinese regions, COVID-19 first broke out in Wuhan, Hubei Province. Therefore, we selected 12 cities in Hubei Province, including Wuhan, where the epidemic was relatively serious, as a group of samples, and cities with few confirmed cases of the epidemic in the same period as the second group of samples. In addition, during the sample period, five cities including Harbin and Dalian showed obvious secondary outbreaks of the epidemic in China. The data of air pollutants we use are daily data, and some pollutants are missing on several days. Therefore, we delete all samples on these days to ensure that the pollutant data of all cities are consistent in time scale. The detailed revisions related this comment are as follows:

2.5. The data on air quality and COVID-19

Following He et al. (2020), the air quality index (AQI) was selected in this paper to measure the air quality of cities in China. We considered three kinds of cities: cities with severe epidemic conditions, cities with few confirmed cases, and cities experiencing secondary outbreaks (Table 1). The data for this paper on six kinds of pollutants, including PM2.5, PM10, SO2, NO2, CO and O3, and the air quality index (AQI) of 348 cities from Jan 1, 2018, to Dec 27, 2020, were collected from the Ministry of Ecological Environment of China (The data of February 6, 2019 is missing, and there is no data of February 29, 2019 and 2018. We delete the data of these two days.). In order to facilitate the presentation of empirical results and based on the consideration of data availability, we collected data from three years and differentiated the samples by year. The sample covers the periods before and after the outbreak of COVID-19. Therefore, sample size was sufficient for empirical analysis of the impacts of secondary outbreaks. Beyond that, the discussion on pollutant emissions is mainly based on the emissions monitored on that day, and does not involve in-depth discussion on the causes and chemical reactions of pollutants.

Table 1 Three types of cities studied in this paper

City type

Cities

Cities in Hubei

Ezhou, Wuhan, Xiaogan, Huanggang, Jingzhou, Suizhou, Xiangyang, Huangshi, Yichang, Jingmen, Xianning and Shiyan

Cities with few confirmed cases

Yichun, Baicheng, Fushun, Heyuan, Zhoushan, Zhangjiajie, Xuancheng, Jingdezhen, Aksu, Binzhou, Leshan, Zhenjiang, Longyan, Chengde, Yingkou, Wuhai, Baise, Lincang, Linfen, Jinchang, Anshun, Shizuishan, Xining and Lhasa

Cities with a second outbreak

Harbin, Dalian, Urumqi, Chengdu and Beijing

The numbers of confirmed cases of COVID-19 in cities with severe epidemic conditions, with few confirmed cases and experiencing secondary outbreaks were obtained from the Wind database. We chose these cities mainly because, compared with other Chinese regions, COVID-19 first broke out in Wuhan, Hubei Province. Therefore, these 12 cities in Hubei Province, including Wuhan, where the epidemic was relatively serious, as a group of samples, and cities with few confirmed cases of the epidemic in the same period as the second group of samples. Most of these cities have fewer than 10 confirmed cases. In addition, during the sample period, five cities including Harbin and Dalian showed obvious secondary outbreaks of the epidemic in China. The number of confirmed cases in these cities is rising again after a period of stability. These cities are the third group.

4: The second-order difference method is used as a method to calculate the differences in air pollutant changes so as to reflect the impact of the epidemic on air quality. However, air quality does not necessarily have an annual cycle, and this place needs more explanations to enhance rationality. Besides, it seems to be debatable that the second-order difference is denoted as DID. The exogenous shocks do not occur at the same time because the selected cities are affected by the epidemic at different points of time. Strictly speaking, there is no control group, and section Empirical Results needs to be modified.

RESPONSE: Thank you for your suggestions. We added explanations for the division of sample years. In fact, how to divide several time points is a difficult problem. In order to facilitate the presentation of empirical results and based on the consideration of data availability, we collected data from three years and differentiated the samples by year. We also added an explanation of the difference method. As for the control group, considering that there are too many samples, we mainly compared the air quality changes of three groups of different types of cities, and did not discuss a specific city. The detailed revisions related this comment are as follows:

  1. Methods and Data

       The empirical study is divided into two parts. First, in Part 1 of Fig. 1, the second-order difference method was used to calculate the differences in air pollutant changes, and then the EEMD method was adopted to obtain and analyze the long-term trends in these differences. For the feasibility of empirical research, it is assumed that for each city, the local government's annual efforts to control air pollution are relatively close, and the difference can eliminate policy human factors to a certain extent. In fact, this paper first uses the difference method to calculate the growth of air pollutants before and after the outbreak of the epidemic, and then uses the signal decomposition method to decompose its components. The difference method is not a DID model, which does not involve explanatory variables and explained variables. Second, a recursive graph was used to allow visualization of trajectory periodicity through phase space. By drawing a recursive graph, we can study some aspects of the m-dimensional phase space trajectory of the AQI through a two-dimensional representation, which reflects the predictability of air quality. Then, recursive quantitative analysis was used for the quantitative analysis of changes in predictability. The specific steps are shown in Fig. 1.

(…)

2.5. The data of air quality and COVID-19

Following He et al. (2020), the air quality index (AQI) was selected in this paper to measure the air quality of cities in China. We considered three kinds of cities: cities with severe epidemic conditions, cities with few confirmed cases, and cities experiencing secondary outbreaks (Table 1). The data for this paper on six kinds of pollutants, including PM2.5, PM10, SO2, NO2, CO and O3, and the air quality index (AQI) of 348 cities from Jan 1, 2018, to Dec 27, 2020, were collected from the Ministry of Ecological Environment of China (The data of February 6, 2019 is missing, and there is no data of February 29, 2019 and 2018. We delete the data of these two days.). In order to facilitate the presentation of empirical results and based on the consideration of data availability, we collected data from three years and differentiated the samples by year. The sample covers the periods before and after the outbreak of COVID-19. Therefore, sample size was sufficient for empirical analysis of the impacts of secondary outbreaks. Beyond that, the discussion on pollutant emissions is mainly based on the emissions monitored on that day, and does not involve in-depth discussion on the causes and chemical reactions of pollutants.

5: In the text, the EEMD method is applied to make up for the shortcomings of EMD. So why not use CEEMD, CEEMDAN, IEEMDAN and other signal decomposition methods? On the other hand, air quality data is not high-frequency data, can traditional time domain analysis methods be used?

RESPONSE: This is a very insightful question. EEMD algorithm reduces the modal effect of EMD by adding noise, and CEEMDAN algorithm further reduces the modal effect by adding adaptive noise, and has better convergence. This paper does not involve the method innovation, and the EEMD method can effectively obtain the results we need. For traditional time domain analysis methods, EEMD is different from FFT or Wavelet method and is more suitable for the data in this paper because it is data adaptive and does not need to consider the stationarity of data, which could could avoid the result error caused by improper selection of basis function.

6: The selection of IMF1 needs to be explained, why not select more components? Meanwhile, as is often the case, IMF1 is a high-frequency component and cannot represent a long-term trend.

RESPONSE: Thank you for your careful examination. After extracting the low-frequency components by the EEMD code, we mistakenly defined them as IMF1, this definition may cause misunderstanding, which should be defined as the last IMF The air data of different cities may have different component quantities in the decomposition process, so this definition is used uniformly). The corresponding contents in the manuscript were also replaced with the last IMF. We also provide all the data and codes. Each component of IMF represents each frequency component in the original signal and is arranged in order from high frequency to low frequency. This is the physical meaning of the IMF (in a very simple case).

7: Is it biased to use the stability of the system to characterize the predictability of AQI? And is RQA indicators more suitable than other indicators such as Permutation Etropy, Sample Entropy and Multiscale Entropy in measuring the stability of time series?

RESPONSE: Thank you for your suggestions. Recursive quantitative analysis is a non-linear system analysis method. It can also provide more indicators to quantitatively analyze system changes, and help us to quantitatively analyze the change laws and properties of air pollution before and after the epidemic, which is in line with our research objectives. Therefore, we have not adopted other methods. This is a good inspiration for our follow-up work.

8: In the paper, by using charts for comparisons, the conclusion is drawn that the system stability of cities that are not severely affected by the epidemic has improved. Nevertheless, the research may need to make the sample size larger and use statistical hypothesis-testing to determine whether there are statistical differences.

RESPONSE: This is a very insightful question. We grouped different cities. Due to the timely and strict epidemic prevention and control measures taken by the Chinese government, the number of cities with very serious epidemic and cities with secondary storms was small, so they did not belong to a large sample. Moreover, the model does not involve regression analysis, so it is difficult to effectively discuss statistical significance. Of course, this difficulty will be the focus of our subsequent discussion. Follow-up studies are planned to collect and collate hard-hit cities from around the world to discuss.

9: Figure 1 is somewhat confusing in expressing the technology roadmap and needs to be further revised. The bar graph in Figure 4 is too dense to be legible and can be devided into several more lines.

RESPONSE: Thank you for your suggestions. We have revised Figure 1 and adjusted the size of Figure 4. The detailed revisions related this comment are as follows:

Fig. 1 Technology roadmap

Fig. 4 RQA indicators of 24 cities where the epidemic was not serious

10: Is the interpretation of Figure 5 wrong? Maybe there isn’t any evidence of a steady upward trend in the mean values of DET, LMAX, ENT, and TT.

RESPONSE: We are sorry for the error. We have corrected the analysis of Figure 5. The detailed revisions related this comment are as follows:

Fig. 5 RQA indicators for five cities where the epidemic was not serious

As shown in Fig. 5, the changes in RQA indicator trends from 2018 to 2020 for five cities with secondary COVID-19 outbreaks were complex and differed from those of the 12 cities that experienced severe epidemic or mild epidemic. Specifically, the mean value of DET and LAM showed steady downward trends, implying that the predictability and stability of the AQI signal are both decreased. The mean value of TT is also decreased, indicating that the duration for which long-term trends in air pollutants remain in a certain state is also decreasing.

We find that the reviewer’s comments have been quite helpful in improving the paper, and we have revised our paper point by point. Once again, thank you very much for your comments and suggestions!

Best Regards!

Yours sincerely!

Reviewer 3 Report

Please see the enclosed file for my comments.

Author Response

We would like to thank the editor and reviewers for their careful and thorough reading of this manuscript and for the thoughtful comments and constructive suggestions, which have helped us improve the manuscript and reach a better scientific level. All of the comments are valuable and have been very helpful during the revision process and have provided important guiding significance to our research. We have carefully revised the manuscript entitled “Improvement and predictability of urban air quality under different stages of the COVID-19 pandemic” (ijerph-1849181) according to the editor and reviewers’ comments. The responses to the reviewers’ comments and the revised portions are reported below each question and are marked in blue and red, respectively. We tried our best to improve the manuscript by making the suggested changes. These changes did not influence the content and framework of the paper. We appreciate the editor’s and reviewers’ thoughtful work earnestly and hope that the corrections will meet with your approval. Thank you for the time and effort that was expended on this paper. The responses to the reviewers’ and editor’s comments are as follows:

Response to the Comments of Reviewer 3

Reviewer’s comments:

1: Originality of the findings: After reading the abstract, I noticed an overlap between the findings the current study reporting and some previous results. In particular, there was a study, which reported improvement of AQI in >300 Chinese cities due to anti-COVID measures (https://link.springer.com/article/10.1007/s11869-020-00963-y). This study is not a single example, the following study (https://www.tandfonline.com/doi/full/10.1080/1540496X.2020.1790353) has reported the improvement of air quality during COVID-19, which later on deteriorated. The latter study from Ming et al., (2020) had therefore already reported the findings, indicated under number (1) and (2) in the abstract after the initial outbreak, the emissions of NO2, CO and PM2.5 declined and then were elevated again. This identified pattern is evidently a part of larger scale trend, reported not only in Ming et al., (2020) above but other studies as well. What is new in the findings (1) and (2) then? These findings do not reveal any patterns of predictability of urban air quality (as stated in the title) but rather describing the previously air quality patterns under COVID-19 conditions in China.

RESPONSE: Thank you for your suggestions. We have added citations to both articles. The data and research perspectives used in these two articles give us good inspiration. They mainly used policy evaluation methods to quantify the impact of lockdown policies on air quality. The innovation of this paper is to analyze the changes of air quality before and after the epidemic by means of signal analysis. This paper focuses on the evolution of air quality during surges and stable periods of the epidemic for areas experiencing severe and mild epidemic conditions by adopting the ensemble empirical mode decomposition (EEMD). To our knowledge, there is no literature to discuss the change of air quality under the outbreak from the perspective of signal analysis.

Beyond that, whether the COVID-19 pandemic could affect the predictability of urban air quality in different cities has not to our knowledge been addressed in previous literature, and accurate prediction of urban air quality is conducive to formulating air control measures in advance. This paper adopts recurrence plots (RPs) and recurrence quantification analysis (RQA) to compare the characteristics of fluctuation in AQI before and after outbreaks, which could be used to describe the effect of the COVID-19 pandemic on the predictability of urban air quality. Specifically, in this paper we first selected the three classes of cities mentioned before: cities experiencing severe epidemic, cities with mild epidemic conditions, and cities experiencing secondary outbreaks. The recurrence plots of the mean values of AQI in these cities are reported to discuss the characteristics of fluctuation in AQI in these different kinds of cities. Second, the recurrence plots and recurrence quantification analysis for mean values of AQI in 2018, 2019 and 2020 are listed for comparison, which could facilitate analysis of differences in air quality evolution characteristics and fluctuations before and after outbreaks from a quantitative perspective. The detailed revisions related this comment are as follows:

  1. Introduction

(…)

Current studies have confirmed that COVID-19 has exerted significant and positive impacts on air quality and assessed these impacts from different aspects (He et al., 2020; Wang et al., 2021b; Elsaid et al., 2021). In China, since the outbreak of COVID-19, the measures to deal with the epidemic situation have also become the focus of public attention (Feng et al., 2020). Zhao et al.(2020) analyzed changes in air quality in Shanghai, Wuhan and Tangshan before and after the outbreak and found that control measures improved air quality in Wuhan. Wang et al. (2021) quantified the impact of reduced intra-city mobility on air quality in 325 Chinese cities from the perspective of population mobility. When variables such as temperature were included, the air quality index (AQI) was 12.2 percent lower in cities with lockdown policies. Ming et al. (2020) used a difference in Differences (DID) model to estimate the impact of the COVID-19 pandemic on air quality and found a significant improvement in air quality during the COVID-19 pandemic.

2: Abstract: There are many unclarities in the abstract. Ideally, the contents of your study should be clear for a reader after he/she just reads an abstract and conclusions. First, it is not clarified where you obtained your aerosol, SO2, NO2, CO and O3 data. Did the data originate from measurements, inventories or modelling? Second, some unspecified methods are mentioned. For instance, you mentioned that you used “difference method” without specifying what does this method represent. I am aware only about the Finite Difference Method. Third, it is unclear which period of COVID-19 outbreak you addressed. I strongly recommend to mention the study period right in the abstract. Moreover, the abstract does not explain why the findings of your study are useful for this research field or society in general. Consider formulating an implication sentence, explaining this aspect in the end of the introduction, please.

RESPONSE: Thank you for your suggestions. We have revised Abstract and explain the data source of air pollution and the introduction of methods used in this paper. We also explained the criteria for classifying the stages of an outbreak. The air quality index (AQI) was selected in this paper to measure the air quality of cities in China. We considered three kinds of cities: cities with severe epidemic conditions, cities with few confirmed cases, and cities experiencing secondary outbreaks (Table 1). The daily air quality index (AQI) data from Jan 1, 2018, to Dec 27, 2020, were obtained from the Ministry of Ecology and Environment, PRC. In order to facilitate the presentation of empirical results and based on the consideration of data availability, we collected data from three years and differentiated the samples by year. The sample covers the periods before and after the outbreak of COVID-19. Therefore, sample size was sufficient for empirical analysis of the impacts of secondary outbreaks. Beyond that, the discussion on pollutant emissions is mainly based on the emissions monitored on that day, and does not involve in-depth discussion on the causes and chemical reactions of pollutants.

Beyond that, the numbers of confirmed cases of COVID-19 in cities with severe epidemic conditions, with few confirmed cases and experiencing secondary outbreaks were obtained from the Wind database. We chose these cities mainly because, compared with other Chinese regions, COVID-19 first broke out in Wuhan, Hubei Province. Therefore, these 12 cities in Hubei Province, including Wuhan, where the epidemic was relatively serious, as a group of samples, and cities with few confirmed cases of the epidemic in the same period as the second group of samples. In addition, during the sample period, five cities including Harbin and Dalian showed obvious secondary outbreaks of the epidemic in China.

In fact, we first divided the city types into three categories. We measured it based on whether there was a second change in the number of confirmed COVID-19 cases reported in each city. But in time, as a result of each city's policy is not completely consistent, for example, some cities because of the outbreak is relatively serious, therefore introduced in March or so traffic blockade measures file, and some cities even delayed until April to release in May, so we can not on time scales, strictly divided all cities, But the traffic lockdown lasts roughly from March to around July.

We have also added explanations on the applicability of these two methods to the research topic of this paper. Specifically, this paper first uses the difference method to calculate the growth of air pollutants before and after the outbreak of the epidemic, and then uses the signal decomposition method to decompose its components. The difference method is not a DID model, which does not involve explanatory variables and explained variables. The detailed revisions related this comment are as follows:

Abstract: This paper investigates the various impact of COVID-19 began in January 2020 on air quality in different types of cities. We analyze the different degrees of improvement of concentrations of six air pollutants (PM2.5, PM10, SO2, NO2, CO and O3) obtained from the Ministry of Ecology and Environment, PRC in different types of Chinese cities from 2018 to 2020 with difference method (the trend of pollutants in different years was differenced) and ensemble empirical mode decomposition (EEMD), and then adopt the recursive plots (RPs) and recursive quantitative analysis (RQA) for the first time to discuss whether air quality is more difficult to predict during the outbreak. The empirical results indicate that: (1) After the initial outbreak, only the emissions of NO2, CO and PM2.5 declined for the first 1-3 months, and during the fourth to fifth months the emissions of six air pollutants were elevated in most cities; (2) For the cities with serious epidemic situations in the Hubei province (China), the air quality is improved, but for the cities experiencing a second outbreak, the air quality was first enhanced and then deteriorated, and the sensitivity of air quality to COVID-19 re-outbreak is decreasing. For Dalian, Harbin and Urumqi, AQI is improved by approximately 12.91, 6.28 and 0.36, respectively. (3) In comparison, the predictability of AQI has declined in cities with serious epidemic situations in Hubei, but AQI achieves a stable state sooner in cities with mild epidemic. The conclusions may facilitate analysis of differences in air quality evolution characteristics and fluctuations before and after outbreaks from a quantitative perspective.

3: Language: The language requires substantial improvement from grammar and stylistic points of views. There are some grammar errors throughout the manuscript. Moreover, unacademic formulations are abundant as well: “brought great harm”, “current studies” (instead of “previous studies”), “anectodical evidence”, “it is worth nothing that due to…”, “differing from the existing literature” (instead of “unlike the existing literature”), “cities were chosen” (instead of “cities were selected”), “purplish”, etc.

RESPONSE: Thank you for your suggestions. We have carefully checked the grammar of the paper and corrected these mistakes and inappropriate statement.

4: Introduction: The introduction is uncommonly lengthy because starting from line 81 you began discussing not the state-of-the art in this studied topic (pollution and COVID-19 in China), but rather, you started to describe your own data, approaches and speculate on advantages and shortcomings of the adopted methodology. This is rather uncommon for the introduction of a peer-reviewed study. I would advise you to structure your paper in the better way, where you could squeeze the information about your (1) research aim, (2) objectives (if applicable), (3) methodology, (4) study period and (5) expected results (not advisable, but shed some light on it if you have a strong desire two) in a single paragraph of the introduction instead of the lengthy, unnecessary part of your introduction, spanning from Line 81 to Line 141. This segment of the introduction also does not resemble introduction because you stopped justifying any statements in this segment by academic references. For instance, the paragraph between the lines 106 and 126 does not have any academic

references. Can you explain why this is the case? Alternatively, you can justify this paragraph by academic references.

RESPONSE: Thank you for your suggestions. The last two paragraphs of the introduction are mainly due to the previous reviewer's request to explain the applicability of the paper's method in the introduction. We have revised the Introduction and removed the subjective content and simplified the whole section. In the paragraph between the lines 106 and 126, the discussion of the epidemic situation in cities is based on authoritative data from the WIND database. In fact, the introduction also contains a discussion of the relevant literature. The detailed revisions related this comment are as follows:

  1. Introduction

(…)

According to Zheng et al. (2020), due to the rapid economic recovery in some regions, emissions will rebound, and air quality may be decreased, and this point has also not been considered in existing studies. In addition, the impact of the COVID-19 outbreak on the predictability of air quality has not been considered in previous literature. Due to the weakening of travel constrictions and retaliatory consumption by residents, fluctuation in air quality may be more severe, which could make the prediction of air pollution more difficult.

Compared with the existing literature, there are two main innovations in this paper. First, differing from the existing literature, which discussed the different levels of air quality before and after the outbreak or investigated the impact of COVID-19 lockdowns on air quality, this paper focuses on the evolution of air quality during surges and stable periods of the epidemic for areas experiencing severe and mild epidemic conditions by adopting the ensemble empirical mode decomposition (EEMD). To our knowledge, there is no literature to discuss the change of air quality under the outbreak from the perspective of signal analysis. This article adopts this method mainly considering the following three aspects: characteristics of data of air pollutants, purpose of research and advantages of methods. Specifically, (1) The data form of air pollutants is non-stationary daily time series data and its sample size is grater than 300 for each year, meeting the data requirements of EEMD method, which is suitable for non-stationary signal analysis. (2) This paper aims to analyze the long term change characteristics of air pollutants, and EEMD could effectively extract this characteristic. (3) The results of EMD may be aliasing, resulting in unclear distinction of components in different time scales. EEMD proposed by Wu and Huang (2009) and used in this paper could make up the shortcoming of EMD and provide more more accurate long time scale information about the air quality.

For the first innovation, the empirical process is as follows: (1) Based on data availability, this paper analyzes and compares trends in six pollutants (PM2.5, PM10, SO2, NO2, CO and O3) in 328 cities between 2018-2019 and 2019-2020. (2) To discuss differences in air quality conditions between cities with severe and mild epidemic situations, we selected 12 cities that experienced severe epidemic conditions in Hubei Province in China and 24 cities with few confirmed cases from other provinces and used the air quality index (AQI) to measure air quality from Jan 1, 2018, to Dec 27, 2020. These cities were chosen to include differing epidemic conditions for comparison. These 12 cities are under severe epidemic conditions because they are all in Hubei province and have more confirmed COVID-19 cases compared with the cities in other provinces, causing widespread concern in the community. We classify the above 24 cities as areas under mild epidemic conditions because the confirmed epidemic cases in these cities are almost zero or few. They all experienced periods of outbreak and stabilization, providing sufficient AQI data samples for analysis. (3) To our knowledge, no investigation has addressed changes in AQI following recurring outbreaks. Whether the impact of COVID-19 on AQI will be weakened with future outbreaks and stabilization of the epidemic situation attracted our interest. Hence, we further selected 5 cities, Harbin, Dalian, Urumqi, Chengdu and Beijing, for investigation of this point and comparison of changes in air quality before and after secondary outbreaks. The epidemic situation was controlled after the outbreak in these cities, new confirmed cases were detected again a few months after the confirmed cases were no longer increased, and the blockade measures in these cities were further strengthened.

5: Methodology: There are many methodological gaps in this study. You did not mention in the abstract where your pollution data were taken from, thereby leaving the gap in your methodological choices. I have not found this information in your methodology either. Thus, it is critically unclear where did you obtain your PM2.5, PM10, SO2, NO2, CO and O3 data? Second, as mentioned, you introduced a term “difference method” in the abstract without explaining what is this. Third, although there are many AQIs, used in the literature, you have not specified which AQI type you applied in this study in the introduction. Fourth, you classified the cities based on some classes, but neither an academic reference, confirming such methodological choice, nor a quantitative evaluation of this classification were presented. Fifth, you introduced such terms as “cities experiencing severe epidemic” and “cities with mild epidemic conditions” but you have not explained the numerical criteria/criterion, allowing to classify the cities as one type or another. Fifth, you introduced a term ?? (t), but I have not found any explanation what does this term mean except the fact that it is obtained during the Gaussian white noise sequencing. Sixth, you have not explained what is “the Wind database”, where neither an academic, nor DOI-based reference was provided. The latter gap makes your study virtually irreproducible, thereby undermining any further prospects at this peer review.

RESPONSE: Thank you for your suggestions. To analyze and compare the evolution of levels of different pollutants during the periods before and after the COVID-19 pandemic, data for this paper on six kinds of pollutants, including PM2.5, PM10, SO2, NO2, CO and O3, and the air quality index (AQI) of 348 cities from Jan 1, 2018, to Dec 27, 2020, were collected from the Ministry of Ecological Environment of China. In order to ensure that the difference results of the growth trend are calculated according to the corresponding date, some missing data are deleted, and the sample size are all 360 for 2018, 2019 and 2020 without missing data. We are very sorry that we did not report these contents in the data section, but put them at the beginning of the results section. We have adjusted the order of these contents.

We have added an simple explanation of the difference method in Abstract. In fact, this method is not DID, but preprocessed the data, and made two differences in the concentration of air pollutants in different years to study the difference in the variation trend of pollutants each year.

We explain the calculation process of AQI index in detail in the supplementary material.

According to the purpose of our study, this paper presents for the first time a method to classify cities during the epidemic.

As for the criterion of city classification, there is no suitable criterion for city classification based on our research topic, and the three types of criteria currently used are proposed in this paper for the first time. And we elaborate the reasons for using such classification criteria in the introduction section. To discuss differences in air quality conditions between cities with severe and mild epidemic situations, we selected 12 cities that experienced severe epidemic conditions in Hubei Province in China and 24 cities with few confirmed cases from other provinces and used the air quality index (AQI) to measure air quality from Jan 1, 2018, to Dec 27, 2020. These cities were chosen to include differing epidemic conditions for comparison. These 12 cities are under severe epidemic conditions because they are all in Hubei province and have more confirmed COVID-19 cases compared with the cities in other provinces, causing widespread concern in the community. We classify the above 24 cities as areas under mild epidemic conditions because the confirmed epidemic cases in these cities are almost zero or few. They all experienced periods of outbreak and stabilization, providing sufficient AQI data samples for analysis. (3) To our knowledge, no investigation has addressed changes in AQI following recurring outbreaks. Whether the impact of COVID-19 on AQI will be weakened with future outbreaks and stabilization of the epidemic situation attracted our interest. Hence, we further selected 5 cities, Harbin, Dalian, Urumqi, Chengdu and Beijing, for investigation of this point and comparison of changes in air quality before and after secondary outbreaks. The epidemic situation was controlled after the outbreak in these cities, new confirmed cases were detected again a few months after the confirmed cases were no longer increased, and the blockade measures in these cities were further strengthened.

The numbers of confirmed cases of COVID-19 in cities with severe epidemic conditions, with few confirmed cases and experiencing secondary outbreaks were obtained from the Wind database. We chose these cities mainly because, compared with other Chinese regions, COVID-19 first broke out in Wuhan, Hubei Province. Therefore, these 12 cities in Hubei Province, including Wuhan, where the epidemic was relatively serious, as a group of samples, and cities with few confirmed cases of the epidemic in the same period as the second group of samples. In addition, during the sample period, five cities including Harbin and Dalian showed obvious secondary outbreaks of the epidemic in China. Areas that were not severely affected were classified on the basis that there were few confirmed cases during the outbreak.

We add a white noise with a standard normal distribution to the original signal to produce a new signal . We have added an explanation about it in Section 2.1.

Wind (https://www.wind.com.cn/default.html) is a financial data and analysis tool service provider. Information is a financial data, information and software service enterprise in mainland China, headquartered in Lujiazui Financial Center, Shanghai. In the domestic market, Wind's customers include more than 90% of China's securities companies, fund management companies, insurance companies, banks and investment companies and other financial enterprises; In the international market, 75 per cent of qualified Foreign institutional investors (QFII) that have been approved by the China Securities Regulatory Commission are Wind's customers. At the same time, most well-known domestic financial academic research institutions and authoritative regulatory institutions are also its customers. A large number of Chinese and English media, research reports and academic papers often cite the data provided by Wind Information. The detailed revisions related this comment are as follows:

2.5. The data of air quality and COVID-19

Following He et al. (2020), the air quality index (AQI) was selected in this paper to measure the air quality of cities in China. We considered three kinds of cities: cities with severe epidemic conditions, cities with few confirmed cases, and cities experiencing secondary outbreaks (Table 1). The data for this paper on six kinds of pollutants, including PM2.5, PM10, SO2, NO2, CO and O3, and the air quality index (AQI) of 348 cities from Jan 1, 2018, to Dec 27, 2020, were collected from the Ministry of Ecological Environment of China (The data of February 6, 2019 is missing, and there is no data of February 29, 2019 and 2018. We delete the data of these two days.). In order to facilitate the presentation of empirical results and based on the consideration of data availability, we collected data from three years and differentiated the samples by year. The sample covers the periods before and after the outbreak of COVID-19. Therefore, sample size was sufficient for empirical analysis of the impacts of secondary outbreaks. Beyond that, the discussion on pollutant emissions is mainly based on the emissions monitored on that day, and does not involve in-depth discussion on the causes and chemical reactions of pollutants.

Abstract: This paper investigates the various impact of COVID-19 began in January 2020 on air quality in different types of cities. We analyze the different degrees of improvement of concentrations of six air pollutants (PM2.5, PM10, SO2, NO2, CO and O3) obtained from the Ministry of Ecology and Environment, PRC in different types of Chinese cities from 2018 to 2020 with difference method (the trend of pollutants in different years was differenced) and ensemble empirical mode decomposition (EEMD), and then adopt the recursive plots (RPs) and recursive quantitative analysis (RQA) for the first time to discuss whether air quality is more difficult to predict during the outbreak. The empirical results indicate that: (1) After the initial outbreak, only the emissions of NO2, CO and PM2.5 declined for the first 1-3 months, and during the fourth to fifth months the emissions of six air pollutants were elevated in most cities; (2) For the cities with serious epidemic situations in the Hubei province (China), the air quality is improved, but for the cities experiencing a second outbreak, the air quality was first enhanced and then deteriorated, and the sensitivity of air quality to COVID-19 re-outbreak is decreasing. For Dalian, Harbin and Urumqi, AQI is improved by approximately 12.91, 6.28 and 0.36, respectively. (3) In comparison, the predictability of AQI has declined in cities with serious epidemic situations in Hubei, but AQI achieves a stable state sooner in cities with mild epidemic. The conclusions may facilitate analysis of differences in air quality evolution characteristics and fluctuations before and after outbreaks from a quantitative perspective.

6: Structure: The article is imperfectly structured as you describe some evidently methodological information in the section, called “Empirical Results” (Lines 260-282). I would suggest you to move this information to the methodology. An uncommonly large number of illustrative figures were places in appendix instead of the main article. Why this is the case is not clear for me because the placement of too many figures in the appendix hampers clear comprehension of your article. Specifically, instead of quickly scrolling down to a required figure below a paragraph, a reader should switch between the documents and scrupulously search for a required figure in this impressive corpus of figures, inserted in the appendix. The contents of the abstract are not harmonized with the contents of conclusions. In the current conditions it looks like you reported different findings in these sections. For instance, the abstract states that “only the emissions of NO2, CO and PM2.5 declined” in the first period, while in conclusions you said that “changes in the levels of CO and O3 declined”. Are you reporting different findings or this is a manifestation of the inconsistency?

RESPONSE: Thank you for your suggestions. We have adjusted the structure of the article and shifted some discussion about methods in the conclusion to the methods section. Given the large number of diagrams that need to be presented in this article, some non-essential diagrams are included in the supplementary material. Beyond that, we have revised the abstract and conclusions to ensure they are consistent. Emissions of NO2, CO and PM2.5 decline after pandemic, and the conclusion section mainly discusses the change in the proportion of cities with improved air quality after the outbreak. The detailed revisions related this comment are as follows:

Abstract: This paper investigates the various impact of COVID-19 began in January 2020 on air quality in different types of cities. We analyze the different degrees of improvement of concentrations of six air pollutants (PM2.5, PM10, SO2, NO2, CO and O3) obtained from the Ministry of Ecology and Environment, PRC in different types of Chinese cities from 2018 to 2020 with difference method (the trend of pollutants in different years was differenced) and ensemble empirical mode decomposition (EEMD), and then adopt the recursive plots (RPs) and recursive quantitative analysis (RQA) for the first time to discuss whether air quality is more difficult to predict during the outbreak. The empirical results indicate that: (1) After the initial outbreak, only the emissions of NO2, CO and PM2.5 declined for the first 1-3 months, and during the fourth to fifth months the emissions of six air pollutants were elevated in most cities; (2) For the cities with serious epidemic situations in the Hubei province (China), the air quality is improved, but for the cities experiencing a second outbreak, the air quality was first enhanced and then deteriorated, and the sensitivity of air quality to COVID-19 re-outbreak is decreasing. For Dalian, Harbin and Urumqi, AQI is improved by approximately 12.91, 6.28 and 0.36, respectively. (3) In comparison, the predictability of AQI has declined in cities with serious epidemic situations in Hubei, but AQI achieves a stable state sooner in cities with mild epidemic. The conclusions may facilitate analysis of differences in air quality evolution characteristics and fluctuations before and after outbreaks from a quantitative perspective.

  1. Conclusions

This paper not only investigated the changes in six air pollutants before and after COVID-19 outbreaks in 328 cities in China by extracting the differences in the growth trends between 2018-2019 and 2019-2020 but also applied recurrence plots and recursive quantitative analysis to compare the changes in predictability of AQI, which may be affected by the COVID-19 pandemic. It was confirmed that due to the initial outbreak and restrictive mitigation policies, the concentrations of different pollutants changed, and the predictability of air quality was also affected.

In the three to four months after the outbreak, there is an increasing proportion of cities with reduced O3 concentrations. We hold that in the post-epidemic era, the government should formulate corresponding control policies for different air pollutants.Changes in trends in the impacts of the COVID-19 pandemic on AQI were found to show obvious differences among different classes of cities. Compared with the cities that did not experience severe epidemic conditions, air quality has noticeably improved in the cities in Hubei, and for the cities with secondary outbreaks, air quality first improved and then deteriorated. During the second stable period, the sensitivity of air quality to epidemic conditions decreased. Therefore, it is suggested that in cities with severe epidemic conditions, it is necessary to strengthen the management of residents’ travel, maintain the effect of improving air quality, and ensure the smooth operation of the economy.

7: Line 10 instead of “focuses” should be “focused on”

RESPONSE: Thank you for your suggestions. We have revised the problem. The detailed revisions related this comment are as follows:

Abstract: This paper investigates the various impact of COVID-19 began in January 2020 on air quality in different types of cities. We analyze the different degrees of improvement of concentrations of six air pollutants (PM2.5, PM10, SO2, NO2, CO and O3) obtained from the Ministry of Ecology and Environment, PRC in different types of Chinese cities from 2018 to 2020 with difference method (the trend of pollutants in different years was differenced) and ensemble empirical mode decomposition (EEMD), and then adopt the recursive plots (RPs) and recursive quantitative analysis (RQA) for the first time to discuss whether air quality is more difficult to predict during the outbreak. The empirical results indicate that: (1) After the initial outbreak, only the emissions of NO2, CO and PM2.5 declined for the first 1-3 months, and during the fourth to fifth months the emissions of six air pollutants were elevated in most cities; (2) For the cities with serious epidemic situations in the Hubei province (China), the air quality is improved, but for the cities experiencing a second outbreak, the air quality was first enhanced and then deteriorated, and the sensitivity of air quality to COVID-19 re-outbreak is decreasing. For Dalian, Harbin and Urumqi, AQI is improved by approximately 12.91, 6.28 and 0.36, respectively. (3) In comparison, the predictability of AQI has declined in cities with serious epidemic situations in Hubei, but AQI achieves a stable state sooner in cities with mild epidemic. The conclusions may facilitate analysis of differences in air quality evolution characteristics and fluctuations before and after outbreaks from a quantitative perspective.

8: Line 11 you say “different types of cities” which may sound problematic here. First, in the next lines you say that you analyzed only the cities in China. If this is the case, such methodological choice should be reflected in Line 11 as well. Moreover, if you addressed to different types of cities, you should likely need to specify which type of cities you mean. Namely, what was the basis of their typology? Size? Economy? Please elaborate on this aspect.

RESPONSE: Thank you for your suggestions. We have added an explanation of the classification criteria for cities in Section 2.5. Considering the word limit and content requirement of the abstract, we did not elaborate the classification criteria in the abstract. The detailed revisions related this comment are as follows:

The numbers of confirmed cases of COVID-19 in cities with severe epidemic conditions, with few confirmed cases and experiencing secondary outbreaks were obtained from the Wind database. We chose these cities mainly because, compared with other Chinese regions, COVID-19 first broke out in Wuhan, Hubei Province. Therefore, these 12 cities in Hubei Province, including Wuhan, where the epidemic was relatively serious, as a group of samples, and cities with few confirmed cases of the epidemic in the same period as the second group of samples. Most of these cities have fewer than 10 confirmed cases. In addition, during the sample period, five cities including Harbin and Dalian showed obvious secondary outbreaks of the epidemic in China. The number of confirmed cases in these cities is rising again after a period of stability. These cities are the third group.

9: Line 13 what is difference method? Finite Difference Method or something else? This should be quickly clarified in the abstract. Note that every uncommon method, being applied in a study, should be ideally explained, or at least, specified.

RESPONSE: Thank you for your suggestions. We illustrate this point in the abstract. The detailed revisions related this comment are as follows:

Abstract: This paper investigates the various impact of COVID-19 began in January 2020 on air quality in different types of cities. We analyze the different degrees of improvement of concentrations of six air pollutants (PM2.5, PM10, SO2, NO2, CO and O3) obtained from the Ministry of Ecology and Environment, PRC in different types of Chinese cities from 2018 to 2020 with difference method (the trend of pollutants in different years was differenced) and ensemble empirical mode decomposition (EEMD), and then adopt the recursive plots (RPs) and recursive quantitative analysis (RQA) for the first time to discuss whether air quality is more difficult to predict during the outbreak. The empirical results indicate that: (1) After the initial outbreak, only the emissions of NO2, CO and PM2.5 declined for the first 1-3 months, and during the fourth to fifth months the emissions of six air pollutants were elevated in most cities; (2) For the cities with serious epidemic situations in the Hubei province (China), the air quality is improved, but for the cities experiencing a second outbreak, the air quality was first enhanced and then deteriorated, and the sensitivity of air quality to COVID-19 re-outbreak is decreasing. For Dalian, Harbin and Urumqi, AQI is improved by approximately 12.91, 6.28 and 0.36, respectively. (3) In comparison, the predictability of AQI has declined in cities with serious epidemic situations in Hubei, but AQI achieves a stable state sooner in cities with mild epidemic. The conclusions may facilitate analysis of differences in air quality evolution characteristics and fluctuations before and after outbreaks from a quantitative perspective.

10: Line 18 You are submitting your article to a journal with general readership. Thus, you should specify here that “Hubei” is a province of China: “the Hubei province (China)”. If you implied some city with the identical name, this should be specified as well.

RESPONSE: Thank you for your suggestions. We have revised this point. The detailed revisions related this comment are as follows:

Abstract: This paper investigates the various impact of COVID-19 began in January 2020 on air quality in different types of cities. We analyze the different degrees of improvement of concentrations of six air pollutants (PM2.5, PM10, SO2, NO2, CO and O3) obtained from the Ministry of Ecology and Environment, PRC in different types of Chinese cities from 2018 to 2020 with difference method (the trend of pollutants in different years was differenced) and ensemble empirical mode decomposition (EEMD), and then adopt the recursive plots (RPs) and recursive quantitative analysis (RQA) for the first time to discuss whether air quality is more difficult to predict during the outbreak. The empirical results indicate that: (1) After the initial outbreak, only the emissions of NO2, CO and PM2.5 declined for the first 1-3 months, and during the fourth to fifth months the emissions of six air pollutants were elevated in most cities; (2) For the cities with serious epidemic situations in the Hubei province (China), the air quality is improved, but for the cities experiencing a second outbreak, the air quality was first enhanced and then deteriorated, and the sensitivity of air quality to COVID-19 re-outbreak is decreasing. For Dalian, Harbin and Urumqi, AQI is improved by approximately 12.91, 6.28 and 0.36, respectively. (3) In comparison, the predictability of AQI has declined in cities with serious epidemic situations in Hubei, but AQI achieves a stable state sooner in cities with mild epidemic. The conclusions may facilitate analysis of differences in air quality evolution characteristics and fluctuations before and after outbreaks from a quantitative perspective.

11: Line 19 What is the second outbreak? How did you define a second outbreak in your study?

RESPONSE: Thank you for your suggestions. We have added an explanation of the classification criteria for cities in Section 2.5. The detailed revisions related this comment are as follows:

The numbers of confirmed cases of COVID-19 in cities with severe epidemic conditions, with few confirmed cases and experiencing secondary outbreaks were obtained from the Wind database. We chose these cities mainly because, compared with other Chinese regions, COVID-19 first broke out in Wuhan, Hubei Province. Therefore, these 12 cities in Hubei Province, including Wuhan, where the epidemic was relatively serious, as a group of samples, and cities with few confirmed cases of the epidemic in the same period as the second group of samples. Most of these cities have fewer than 10 confirmed cases. In addition, during the sample period, five cities including Harbin and Dalian showed obvious secondary outbreaks of the epidemic in China. The number of confirmed cases in these cities is rising again after a period of stability. These cities are the third group.

12: Line 18 “improved significantly”. Ideally, the word “significant” in peer-review should be justified by the indicators of statistical significance (p-value for instance).

RESPONSE: Thank you for your suggestions. We have revised this point. The detailed revisions related this comment are as follows:

Abstract: This paper investigates the various impact of COVID-19 began in January 2020 on air quality in different types of cities. We analyze the different degrees of improvement of concentrations of six air pollutants (PM2.5, PM10, SO2, NO2, CO and O3) obtained from the Ministry of Ecology and Environment, PRC in different types of Chinese cities from 2018 to 2020 with difference method (the trend of pollutants in different years was differenced) and ensemble empirical mode decomposition (EEMD), and then adopt the recursive plots (RPs) and recursive quantitative analysis (RQA) for the first time to discuss whether air quality is more difficult to predict during the outbreak. The empirical results indicate that: (1) After the initial outbreak, only the emissions of NO2, CO and PM2.5 declined for the first 1-3 months, and during the fourth to fifth months the emissions of six air pollutants were elevated in most cities; (2) For the cities with serious epidemic situations in the Hubei province (China), the air quality is improved, but for the cities experiencing a second outbreak, the air quality was first enhanced and then deteriorated, and the sensitivity of air quality to COVID-19 re-outbreak is decreasing. For Dalian, Harbin and Urumqi, AQI is improved by approximately 12.91, 6.28 and 0.36, respectively. (3) In comparison, the predictability of AQI has declined in cities with serious epidemic situations in Hubei, but AQI achieves a stable state sooner in cities with mild epidemic. The conclusions may facilitate analysis of differences in air quality evolution characteristics and fluctuations before and after outbreaks from a quantitative perspective.

13: Line 37 EEA is an acronym and it was explained here, but NASA is also an acronym. Why it was not explained like EEA then?

RESPONSE: Thank you for your suggestions. We have added relevant explanations. The detailed revisions related this comment are as follows:

  1. Introduction

The outbreak and spread of COVID-19 had a significant impact on the lives of people and the operation of the economy (Bauwens et al., 2020; Venter et al., 2020; Briz-Redón, et al., 2021; McKee and Stuckler, 2020; Guan et al., 2020; Choi and Brindley, 2021). Governments have implemented various measures to prevent the spread of the disease, such as controls on movement and personnel isolation measures (Chang et al., 2020; Tian et al., 2020) and case isolation and home quarantine (Shi and Brasseur, 2020; Silver et al., 2020; Filonchyk et al., 2020; Cole et al., 2020), which has led to significant reductions in travel and industrial products. Therefore, the negative impact of urban residents’ living activities on urban ambient air quality may be weakened, and air quality could be improved (Chen et al., 2021), which is the main purpose of this paper. According to the air quality change data and satellite images released by National Aeronautics and Space Administration (NASA) and the European Environment Agency (EEA) before and after the epidemic, due to the outbreak of COVID-19, the carbon dioxide and nitrogen dioxide contents in the air in various regions showed a downward trend. This provided a new perspective from which to study changes in air quality during the outbreak resulting from these regulations (Le et al., 2020; Tollefson, 2020; Forster et al., 2020) and the influence of measures for curbing the spread of COVID-19 on these changes and the improvement of environmental quality (Liu et al., 2020a; Liu et al., 2020b; Huang et al., 2021).

14: Lines 58-59 “This negative influence was more significant…” this statement does require an academic reference for justification. The same is applied to the next sentence about difference of emission reduction depending on countries and their measures against COVID-19.

RESPONSE: Thank you for your suggestions. This statement is a quotation from Liu et al. (2020) in this paper and we have revised this point. The detailed revisions related this comment are as follows:

From the perspective of energy demand and residents’ consumption, Liu et al. (2020) pointed out that energy use and carbon dioxide emissions have decreased during the COVID-19 pandemic, and this negative influence was more significant than that observed during previous economic downturns.

15: Line 66: COVID-19 is an acronym, so it cannot be written as “covid-19”

RESPONSE: Thank you for your suggestions. We checked the articles to make sure they were capitalized.

16: Lines 72-79 lack any references despite the facts you stated here might seem controversial. Do you have any academic references in mind to justify these statements?

RESPONSE: Thank you for your suggestions. ‘emissions will rebound’ is a quotation from Zheng et al. (2020) and lines 74 to 79 provide a review of current research, we hold that no studies have yet looked at the impact of a secondary outbreak on air quality, which serves to introduce the next paragraph on the innovation of this paper.

17: Line 109: Mention this study period in the abstract as well, please.

RESPONSE: Thank you for your suggestions. We have revised this point. The detailed revisions related this comment are as follows:

Abstract: This paper investigates the various impact of COVID-19 began in January 2020 on air quality in different types of cities. We analyze the different degrees of improvement of concentrations of six air pollutants (PM2.5, PM10, SO2, NO2, CO and O3) obtained from the Ministry of Ecology and Environment, PRC in different types of Chinese cities from 2018 to 2020 with difference method (the trend of pollutants in different years was differenced) and ensemble empirical mode decomposition (EEMD), and then adopt the recursive plots (RPs) and recursive quantitative analysis (RQA) for the first time to discuss whether air quality is more difficult to predict during the outbreak. The empirical results indicate that: (1) After the initial outbreak, only the emissions of NO2, CO and PM2.5 declined for the first 1-3 months, and during the fourth to fifth months the emissions of six air pollutants were elevated in most cities; (2) For the cities with serious epidemic situations in the Hubei province (China), the air quality is improved, but for the cities experiencing a second outbreak, the air quality was first enhanced and then deteriorated, and the sensitivity of air quality to COVID-19 re-outbreak is decreasing. For Dalian, Harbin and Urumqi, AQI is improved by approximately 12.91, 6.28 and 0.36, respectively. (3) In comparison, the predictability of AQI has declined in cities with serious epidemic situations in Hubei, but AQI achieves a stable state sooner in cities with mild epidemic. The conclusions may facilitate analysis of differences in air quality evolution characteristics and fluctuations before and after outbreaks from a quantitative perspective.

18: Line 112: It is unclear from the current formulation which air quality index you meant here? There are multiple AQIs, applied in research literature; they vary depending on the country, type of pollutants (aerosol or a certain gas) and the methodology of the calculation. See the following paper for checking these indices in details (https://link.springer.com/article/10.1007/s11869-016-0435-y).

RESPONSE: Thank you for your suggestions. We explain the calculation process of AQI index in detail in the supplementary material.

19: Lines 114-115: It is not a common knowledge for readers to link the location of cities in the Hubei province and the abundance of COVID-19 cases. First, you rather need to mention that Hubei suffered from many COVID-19 cases and only then, to mention this.

RESPONSE: Thank you for your suggestions. For our sample, we divided cities according to the number of local COVID-19 cases counted in each city, without distinguishing the travel trajectory prior to each confirmed case. All confirmed COVID-19 data were obtained from the WIND database.

20: Lines 128-141: This paragraph takes a completely different tangent by creating a new research question that will be addressed in this study. For clarity, I would strongly

recommend to emphasize your research aim in the end of the introduction, as mentioned in the major comment about the introduction above.

RESPONSE: Thank you for your suggestions. The second innovation aims to describe the effect of the COVID-19 pandemic on the predictability of urban air quality, , which could facilitate analysis of differences in air quality evolution characteristics and fluctuations before and after outbreaks from a quantitative perspective. At the end of the introduction, we elaborate the research purpose. The detailed revisions related this comment are as follows:

Second, whether the COVID-19 pandemic could affect the predictability of urban air quality in different cities has not to our knowledge been addressed in previous literature. This paper adopts recurrence plots (RPs) and recurrence quantification analysis (RQA) to compare the characteristics of fluctuation in AQI before and after outbreaks, which could be used to describe the effect of the COVID-19 pandemic on the predictability of urban air quality. Specifically, in this paper we first selected the three classes of cities mentioned before: cities experiencing severe epidemic, cities with mild epidemic conditions, and cities experiencing secondary outbreaks. The recurrence plots of the mean values of AQI in these cities are reported to discuss the characteristics of fluctuation in AQI in these different kinds of cities. Second, the recurrence plots and recurrence quantification analysis for mean values of AQI in 2018, 2019 and 2020 are listed for comparison. This may help to accurate prediction of urban air quality, facilitate analysis of differences in air quality evolution characteristics and fluctuations before and after outbreaks from a quantitative perspective and formulate air control measures in advance.

21: Lines 150-151. What does it mean that the local government annual efforts to control air pollution are relatively close? What are the policy human factors? This sentence is extremely vaguely formulated despite the assumption you are making here might have strongly affected your results. I would advise you devote much more space in the methodology to explain this assumption. Explain in few words, who is responsible in enacting, imposing and enforcing the COVID-19-related restrictions in China. Moreover, explain what are the policy human factors and give some examples for a reader to understand why these factors are important. Third, give some illustrative example why we can neglect the differences in the way how different cities in China coped with COVID-19, thereby confirming the sanity of your assumption. Otherwise, I would consider that you made rather a blind assumption for maximum simplification of your approach.

RESPONSE: Thank you for your suggestions. We assume that the Chinese government's efforts to combat air pollution will not vary too much from year to year to interfere with the impact of lockdown policies. Policy differences can be eliminated in the difference process, this treatment is similar to the regression equations do difference elimination of various effects. Without this assumption, it is impossible to divide cities. In fact, we surveyed the air governance policies published on the Policy Document Library of Chinese government website (https://www.gov.cn/) and the government websites of various cities in the past three years, and concluded that for our sample cities, these governance measures would not affect our analysis of the impact of the epidemic on air quality.

22: Lines 152-153 Trajectory periodicity of what?

RESPONSE: We show the trajectory of AQI time series on a two-dimensional plane by using dimensionality raising technique.

23: Line 161-162: Please specify where the exact equation can be checked in this paper, thus, reducing the probability that a reader is confused by this vague formulation here.

RESPONSE: Thank you for your suggestions. We added a formula for that. The detailed revisions related this comment are as follows:

2.1. Ensemble Empirical Mode Decomposition

For cities experiencing severe epidemic and those with mild epidemic conditions, the difference in AQI between 2020 and 2019 minus the difference in AQI between 2019 and 2018 was evaluated in this paper. The change in trends for these pollutants caused by COVID-19 was calculated by

                (1)

where ,  and  represent the pollutants in 2020, 2019 and 2018, respectively. The results for each city were averaged and recorded as . Then, the EEMD method was adopted to obtain the long-term impact of COVID-19 on AQI. EEMD could make up the shortcoming of EMD. In the process of decomposing the IMF with EMD, many iterations are required, and the conditions for stopping the iteration lack a standard, so the IMFs obtained by different conditions for stopping the iteration are also different. This shortcomings has been solved by EEMD proposed by Wu and Huang (2009) and used in this paper. The EEMD decomposition principle is: when the additional white noise is evenly distributed in the whole time-frequency space, the time-frequency space is composed of different scale components divided by the filter bank. In this paper, EEMD is adopted to analyze the differences in air pollutant changes. The other IMFs have its own meaning separately, and IMF1, IMF2,…, IMF12 indicate the time scale is getting smaller and smaller according to the calculation formula of IMFs. Since the last IMF is the closest IMF to the long-term trend, it is used in this paper. Specifically, EEMD algorithm steps are as follows:

24: Line 204: Why AQI is emphasized by bold text here? Please check if there is any reason behind this emphasis.

RESPONSE: Thank you for your suggestions. The bold here represents the set of multiple one-dimensional vectors.

25: Line 212: The alignment of Equation 6 is likely broken, see the left margin of the equation.

RESPONSE: Thank you for your suggestions. We centered the formulas and double-checked the alignment of all the formulas.

26: Lines 247-248: As mentioned, there are multiple AQIs applied in research literature. Thus, consider either elaborating what does this particular AQI reflects or providing an equation which can demonstrate how your particular AQI was calculated.

RESPONSE: Thank you for your suggestions. We have provided the detailed AQI formula in the supplementary material.

27: Line 257: What was the quantitative criterion/criteria to classify a city as “severe”, for instance? Without such explanation, this choice looks rather arbitrary.

RESPONSE: Thank you for your suggestions. We have added an explanation of city classification criteria to the article. The detailed revisions related this comment are as follows:

The numbers of confirmed cases of COVID-19 in cities with severe epidemic conditions, with few confirmed cases and experiencing secondary outbreaks were obtained from the Wind database. We chose these cities mainly because, compared with other Chinese regions, COVID-19 first broke out in Wuhan, Hubei Province. Therefore, these 12 cities in Hubei Province, including Wuhan, where the epidemic was relatively serious, as a group of samples, and cities with few confirmed cases of the epidemic in the same period as the second group of samples. Most of these cities have fewer than 10 confirmed cases. In addition, during the sample period, five cities including Harbin and Dalian showed obvious secondary outbreaks of the epidemic in China. The number of confirmed cases in these cities is rising again after a period of stability. These cities are the third group.

28: Line 260: The methodology is finished here, but I have not found any information on where did you obtain your PM2.5, PM10, SO2, NO2, CO and O3 data?

RESPONSE: Thank you for your suggestions. We added an explanation of the source of the data in the article and abstract. The detailed revisions related this comment are as follows:

Abstract: This paper investigates the various impact of COVID-19 began in January 2020 on air quality in different types of cities. We analyze the different degrees of improvement of concentrations of six air pollutants (PM2.5, PM10, SO2, NO2, CO and O3) obtained from the Ministry of Ecology and Environment, PRC in different types of Chinese cities from 2018 to 2020 with difference method (the trend of pollutants in different years was differenced) and ensemble empirical mode decomposition (EEMD), and then adopt the recursive plots (RPs) and recursive quantitative analysis (RQA) for the first time to discuss whether air quality is more difficult to predict during the outbreak. The empirical results indicate that: (1) After the initial outbreak, only the emissions of NO2, CO and PM2.5 declined for the first 1-3 months, and during the fourth to fifth months the emissions of six air pollutants were elevated in most cities; (2) For the cities with serious epidemic situations in the Hubei province (China), the air quality is improved, but for the cities experiencing a second outbreak, the air quality was first enhanced and then deteriorated, and the sensitivity of air quality to COVID-19 re-outbreak is decreasing. For Dalian, Harbin and Urumqi, AQI is improved by approximately 12.91, 6.28 and 0.36, respectively. (3) In comparison, the predictability of AQI has declined in cities with serious epidemic situations in Hubei, but AQI achieves a stable state sooner in cities with mild epidemic. The conclusions may facilitate analysis of differences in air quality evolution characteristics and fluctuations before and after outbreaks from a quantitative perspective.

2.5. The data of air quality and COVID-19

Following He et al. (2020), the air quality index (AQI) was selected in this paper to measure the air quality of cities in China. We considered three kinds of cities: cities with severe epidemic conditions, cities with few confirmed cases, and cities experiencing secondary outbreaks (Table 1). The data for this paper on six kinds of pollutants, including PM2.5, PM10, SO2, NO2, CO and O3, and the air quality index (AQI) of 348 cities from Jan 1, 2018, to Dec 27, 2020, were collected from the Ministry of Ecological Environment of China (The data of February 6, 2019 is missing, and there is no data of February 29, 2019 and 2018. We delete the data of these two days.). In order to facilitate the presentation of empirical results and based on the consideration of data availability, we collected data from three years and differentiated the samples by year. The sample covers the periods before and after the outbreak of COVID-19. Therefore, sample size was sufficient for empirical analysis of the impacts of secondary outbreaks. Beyond that, the discussion on pollutant emissions is mainly based on the emissions monitored on that day, and does not involve in-depth discussion on the causes and chemical reactions of pollutants.

29: Lines 258-260. What is “the Wind database”. Kindly note that without an academic reference to this dataset or a DOI-based reference to this dataset, it is impossible to find what can of data you used, which makes your study unreproducible. Note that the irreproducibility of the study is a very solid reason of rebuttal even if other aspects of your study had perfect quality.

RESPONSE: Wind (https://www.wind.com.cn/default.html) is a financial data and analysis tool service provider. Information is a financial data, information and software service enterprise in mainland China, headquartered in Lujiazui Financial Center, Shanghai. In the domestic market, Wind's customers include more than 90% of China's securities companies, fund management companies, insurance companies, banks and investment companies and other financial enterprises; In the international market, 75 per cent of qualified Foreign institutional investors (QFII) that have been approved by the China Securities Regulatory Commission are Wind's customers. At the same time, most well-known domestic financial academic research institutions and authoritative regulatory institutions are also its customers. A large number of Chinese and English media, research reports and academic papers often cite the data provided by Wind Information.

30: Lines 262-282: This information would better fit the methodological description, not the empirical results. Once again, one of the key aspects of your data acquisition remained unclarified. Where have you obtained all these PM2.5, PM10, SO2, NO2, CO and O3 data from?

RESPONSE: Thank you for your suggestions. We added an explanation of the source of the data in the article and abstract. The detailed revisions related this comment are as follows:

Abstract: This paper investigates the various impact of COVID-19 began in January 2020 on air quality in different types of cities. We analyze the different degrees of improvement of concentrations of six air pollutants (PM2.5, PM10, SO2, NO2, CO and O3) obtained from the Ministry of Ecology and Environment, PRC in different types of Chinese cities from 2018 to 2020 with difference method (the trend of pollutants in different years was differenced) and ensemble empirical mode decomposition (EEMD), and then adopt the recursive plots (RPs) and recursive quantitative analysis (RQA) for the first time to discuss whether air quality is more difficult to predict during the outbreak. The empirical results indicate that: (1) After the initial outbreak, only the emissions of NO2, CO and PM2.5 declined for the first 1-3 months, and during the fourth to fifth months the emissions of six air pollutants were elevated in most cities; (2) For the cities with serious epidemic situations in the Hubei province (China), the air quality is improved, but for the cities experiencing a second outbreak, the air quality was first enhanced and then deteriorated, and the sensitivity of air quality to COVID-19 re-outbreak is decreasing. For Dalian, Harbin and Urumqi, AQI is improved by approximately 12.91, 6.28 and 0.36, respectively. (3) In comparison, the predictability of AQI has declined in cities with serious epidemic situations in Hubei, but AQI achieves a stable state sooner in cities with mild epidemic. The conclusions may facilitate analysis of differences in air quality evolution characteristics and fluctuations before and after outbreaks from a quantitative perspective.

2.5. The data of air quality and COVID-19

Following He et al. (2020), the air quality index (AQI) was selected in this paper to measure the air quality of cities in China. We considered three kinds of cities: cities with severe epidemic conditions, cities with few confirmed cases, and cities experiencing secondary outbreaks (Table 1). The data for this paper on six kinds of pollutants, including PM2.5, PM10, SO2, NO2, CO and O3, and the air quality index (AQI) of 348 cities from Jan 1, 2018, to Dec 27, 2020, were collected from the Ministry of Ecological Environment of China (The data of February 6, 2019 is missing, and there is no data of February 29, 2019 and 2018. We delete the data of these two days.). In order to facilitate the presentation of empirical results and based on the consideration of data availability, we collected data from three years and differentiated the samples by year. The sample covers the periods before and after the outbreak of COVID-19. Therefore, sample size was sufficient for empirical analysis of the impacts of secondary outbreaks. Beyond that, the discussion on pollutant emissions is mainly based on the emissions monitored on that day, and does not involve in-depth discussion on the causes and chemical reactions of pollutants.

31: Line 267: The procedure of deleting the data and the corresponding criteria for filtering out the data should have been described either in the methodology or in the supplementary material as well

RESPONSE: Thank you for your suggestions. The data of February 6, 2019 is missing, and there is no data of February 29, 2019 and 2018. We delete the data of these two days. We added this explanation. The detailed revisions related this comment are as follows:

2.5. The data of air quality and COVID-19

Following He et al. (2020), the air quality index (AQI) was selected in this paper to measure the air quality of cities in China. We considered three kinds of cities: cities with severe epidemic conditions, cities with few confirmed cases, and cities experiencing secondary outbreaks (Table 1). The data for this paper on six kinds of pollutants, including PM2.5, PM10, SO2, NO2, CO and O3, and the air quality index (AQI) of 348 cities from Jan 1, 2018, to Dec 27, 2020, were collected from the Ministry of Ecological Environment of China (The data of February 6, 2019 is missing, and there is no data of February 29, 2019 and 2018. We delete the data of these two days.). In order to facilitate the presentation of empirical results and based on the consideration of data availability, we collected data from three years and differentiated the samples by year. The sample covers the periods before and after the outbreak of COVID-19. Therefore, sample size was sufficient for empirical analysis of the impacts of secondary outbreaks. Beyond that, the discussion on pollutant emissions is mainly based on the emissions monitored on that day, and does not involve in-depth discussion on the causes and chemical reactions of pollutants.

32: Line 269: As your appendix is very lengthy, it is advisable to refer to the exact figure/table/section of the appendix. Kindly specify where a reader can find such information if the appendix.

RESPONSE: Thank you for your suggestions. We have examined the charts in the supplementary material, and ‘appendix’ is replaced with ‘supplementary material’.

33: Line 272: This formula should have been likely presented in the methodological description once you mentioned it in the text for the first time (see Lines 161 and 162).

RESPONSE: Thank you for your suggestions. We provide the relevant formula in Section 2.1. The detailed revisions related this comment are as follows:

2.1. Ensemble Empirical Mode Decomposition

For cities experiencing severe epidemic and those with mild epidemic conditions, the difference in AQI between 2020 and 2019 minus the difference in AQI between 2019 and 2018 was evaluated in this paper. The change in trends for these pollutants caused by COVID-19 was calculated by

                (1)

where ,  and  represent the pollutants in 2020, 2019 and 2018, respectively. The results for each city were averaged and recorded as . Then, the EEMD method was adopted to obtain the long-term impact of COVID-19 on AQI. EEMD could make up the shortcoming of EMD. In the process of decomposing the IMF with EMD, many iterations are required, and the conditions for stopping the iteration lack a standard, so the IMFs obtained by different conditions for stopping the iteration are also different. This shortcomings has been solved by EEMD proposed by Wu and Huang (2009) and used in this paper. The EEMD decomposition principle is: when the additional white noise is evenly distributed in the whole time-frequency space, the time-frequency space is composed of different scale components divided by the filter bank. In this paper, EEMD is adopted to analyze the differences in air pollutant changes. The other IMFs have its own meaning separately, and IMF1, IMF2,…, IMF12 indicate the time scale is getting smaller and smaller according to the calculation formula of IMFs. Since the last IMF is the closest IMF to the long-term trend, it is used in this paper. Specifically, EEMD algorithm steps are as follows:

34: Line 291: Please specify the months which you mean here right in the text.

RESPONSE: Thank you for your suggestions. We have added relevant instructions. The detailed revisions related this comment are as follows:

As shown in Table 2, Fig. 2 and Supplementary Materials, Fig. S1, within two months (Jan and Feb) after the COVID-19 outbreak, the emissions of CO, NO2, PM10 and PM2.5 were reduced, but the emissions of SO2 were elevated. Considering that O3 is secondary pollutant and its emissions could be affected by meteorological factors such as lightning, therefore it is hard to evaluate whether the negative impact of the outbreak on O3 emissions is significant. From Apr to Jun 2020, strict home isolation and business shutdown policies (such as the traffic control measures issued in response to the epidemic) may have begun to be successful. Therefore, the rates of growth of SO2 levels began to decrease. After Aug 2020, the emissions of most pollutants declined. During this stable period, although the epidemic in China was effectively controlled, the government did not relax control of transportation. In Supplementary Materials, Fig. S1, different colors represent the increase or decrease of the last IMF of various pollutant emissions.

35: Line 292-293: This sentence does not have any sense and is seemingly incomplete. Kindly see that you left an open bracket, that was never closed by the end of the sentence.

RESPONSE: Thank you for your suggestions. The close bracket follows ‘peroxyacetylnitrate’. We have deleted this sentence. The detailed revisions related this comment are as follows:

As shown in Table 2, Fig. 2 and Supplementary Materials, Fig. S1, within two months (Jan and Feb) after the COVID-19 outbreak, the emissions of CO, NO2, PM10 and PM2.5 were reduced, but the emissions of SO2 were elevated. Considering that O3 is secondary pollutant and its emissions could be affected by meteorological factors such as lightning, therefore it is hard to evaluate whether the negative impact of the outbreak on O3 emissions is significant. From Apr to Jun 2020, strict home isolation and business shutdown policies (such as the traffic control measures issued in response to the epidemic) may have begun to be successful. Therefore, the rates of growth of SO2 levels began to decrease. After Aug 2020, the emissions of most pollutants declined. During this stable period, although the epidemic in China was effectively controlled, the government did not relax control of transportation. In Supplementary Materials, Fig. S1, different colors represent the increase or decrease of the last IMF of various pollutant emissions.

36: Line 294: This statement requires an academic reference to some O3-based fundamental study.

RESPONSE: Thank you for your suggestions. The statement here is not relevant to the topic of this article and we have removed it. The detailed revisions related this comment are as follows:

As shown in Table 2, Fig. 2 and Supplementary Materials, Fig. S1, within two months (Jan and Feb) after the COVID-19 outbreak, the emissions of CO, NO2, PM10 and PM2.5 were reduced, but the emissions of SO2 were elevated. Considering that O3 is secondary pollutant and its emissions could be affected by meteorological factors such as lightning, therefore it is hard to evaluate whether the negative impact of the outbreak on O3 emissions is significant. From Apr to Jun 2020, strict home isolation and business shutdown policies (such as the traffic control measures issued in response to the epidemic) may have begun to be successful. Therefore, the rates of growth of SO2 levels began to decrease. After Aug 2020, the emissions of most pollutants declined. During this stable period, although the epidemic in China was effectively controlled, the government did not relax control of transportation. In Supplementary Materials, Fig. S1, different colors represent the increase or decrease of the last IMF of various pollutant emissions.

37: Lines 294-299. There are several problems in this speculation. First, you have mentioned the word “significant” and that it is hard to evaluate the significance here. Statistical significance is a very strictly defined term, which can demonstrate that a result from data generated by testing or experimentation is related to a specific cause or not. If such significance cannot be proven for O3, I assumed that you addressed the statistical significance of negative impact of the outbreak on other pollutant emissions. If this is the case, where a reader can see the results? Moreover, you speculated that O3 emissions could be affected by meteorological factors. The problem is that the emissions of NO2, CO and, especially, particulate matter, are also affected by meteorological and by their own unique factors (NO2 – transportation and sub-daily dynamics or vehicles, CO – combustion and its sub-daily dynamics as well as by seasonality, for instance). Most critically, how did you disentangle seasonal effects of CO? While it is theoretically possible to assess monthly-scale temporal trends of aerosols or anthropogenically-driven gaseous pollutants such as NO2, it is impossible to assess suchtrends for CO without detrending or disentangling its seasonal effects because CO is normally low in summer due to enhanced land carbon sink and strong in winter due to weakness of this sink and decomposition of plants/biological material in soil. If your data were taken from inventories about pollutant emissions, this comment is not important, though. This is critical only if you CO estimates were taken from measurements. Moreover, the decrease of all the pollutants you analyzed except O3 had been already reported for Chinese cities by (https://link.springer.com/article/10.1007/s10668-022-02353-z). Perhaps, you can check their study to learn about the potential drivers of this contrast.

RESPONSE: Thank you for your suggestions. We have changed the statement about significance. Beyond that, we did not measure the relevant pollutant data, the data came from the emission inventory of China's Ministry of Environmental Protection. We were very inspired by their research, and we cited their conclusions in the introduction. In fact, our study differs from theirs. This paper further adopts recurrence plots (RPs) and recurrence quantification analysis (RQA) to compare the characteristics of fluctuation in AQI before and after outbreaks, which could be used to describe the effect of the COVID-19 pandemic on the predictability of urban air quality. Specifically, in this paper we first selected the three classes of cities mentioned before: cities experiencing severe epidemic, cities with mild epidemic conditions, and cities experiencing secondary outbreaks. The recurrence plots of the mean values of AQI in these cities are reported to discuss the characteristics of fluctuation in AQI in these different kinds of cities. Second, the recurrence plots and recurrence quantification analysis for mean values of AQI in 2018, 2019 and 2020 are listed for comparison. This may help to accurate prediction of urban air quality, facilitate analysis of differences in air quality evolution characteristics and fluctuations before and after outbreaks from a quantitative perspective and formulate air control measures in advance. The detailed revisions related this comment are as follows:

  1. Introduction

(…)

Current studies have confirmed that COVID-19 has exerted significant and positive impacts on air quality and assessed these impacts from different aspects (He et al., 2020; Wang et al., 2021b; Elsaid et al., 2021). In China, since the outbreak of COVID-19, the measures to deal with the epidemic situation have also become the focus of public attention (Feng et al., 2020). Zhao et al.(2020) analyzed changes in air quality in Shanghai, Wuhan and Tangshan before and after the outbreak and found that control measures improved air quality in Wuhan. Wang et al. (2021) quantified the impact of reduced intra-city mobility on air quality in 325 Chinese cities from the perspective of population mobility. When variables such as temperature were included, the air quality index (AQI) was 12.2 percent lower in cities with lockdown policies. Ming et al. (2020) used a difference in Differences (DID) model to estimate the impact of the COVID-19 pandemic on air quality and found a significant improvement in air quality during the COVID-19 pandemic.

38: Line 300: “may have been successful” is an exceedingly modal, suggestive sentence, that is not supported by any numerical arguments. In other words, can you present any arguments, corroborating the efficiency of shutdown policies? Moreover, a reader is not aware about what is “shutdown policy”. Perhaps, it has sense to elaborate the main principles of this policy, while pointing out the time when this policy was enacted? Otherwise, this statement looks unjustified as well.

RESPONSE: Thank you for your suggestions. This is just a subjective analysis based on our conclusions. It is precisely because of the effect of the relevant lockdown policies that air quality began to improve, which is also consistent with the conclusions of the current research mentioned in the introduction. The detailed revisions related this comment are as follows:

3.1. Effects of epidemic context on different pollutants in 348 cities

(…)

As shown in Table 2, Fig. 2 and Supplementary Materials, Fig. S1, within two months (Jan and Feb) after the COVID-19 outbreak, the emissions of CO, NO2, PM10 and PM2.5 were reduced, but the emissions of SO2 were elevated. Considering that O3 is secondary pollutant and its emissions could be affected by meteorological factors such as lightning, therefore it is hard to evaluate whether the negative impact of the outbreak on O3 emissions is significant. From Apr to Jun 2020, strict home isolation and business shutdown policies (such as the traffic control measures issued in response to the epidemic) may have begun to be successful. Therefore, the rates of growth of SO2 levels began to decrease. After Aug 2020, the emissions of most pollutants declined. During this stable period, although the epidemic in China was effectively controlled, the government did not relax control of transportation. In Supplementary Materials, Fig. S1, different colors represent the increase or decrease of the last IMF of various pollutant emissions. For example, yellow represents the increased IMFs, indicating that pollutants have an upward trend in long-term time scale. In addition, the sizes of color boxes represent degrees of change. As shown in Supplementary Materials, Fig. S1, during the initial outbreak period (Jan, Feb, Mar and Apr), the emissions of PM2.5, PM10 and NO2 declined, but changes in CO emissions were not noticeable. Furthermore, percentages of cities with improved air quality for different pollutants are analyzed in this paper, and the results are provided in Table 3 and Supplementary Materials, Fig. S2. Beyond that, the average of the monthly city-wide air pollutant emission data for primary pollutants are provided in Supplementary Materials, Tables S1, S2 and S3 and the time trends of the Monthly city-wide mean value for different primary pollutants is reported in Supplementary Materials, Fig. S14.

39: Lines 301-302: Please support the statement “declined significantly” by providing the arguments of statistical significance here.

RESPONSE: Thank you for your suggestions. We have changed the statement about significance and deleted ‘significantly’.

40: Lines 302-303: Did you evaluate the control of COVID-19 epidemic in China in your study? If yes, please provide the arguments, justifying this statement. Otherwise, please provide a reference to any previous study, which proved that these controls were effective. Also, you have not actually mentioned that you meant COVID-19 pandemic here.

RESPONSE: Thank you for your suggestions. This is not a policy assessment of pandemic lockdowns, but rather an assessment and comparison of changes in air quality before and after the pandemic. Instead of using a policy evaluation approach, we use a signal analysis approach. In addition, we use the number of confirmed cases in each city as a proxy for the severity of COVID-19 occurrence and classify cities based on these data. These data comes from Wind database.

41: Lines 308-309: Once again, CO experiences seasonal changes and the same treatment of CO as particulate matter in temporal dimension would undesirably bias your results. I strongly recommend to remove seasonal component from CO before starting this analysis.

RESPONSE: Thank you for your suggestions. We believe that the difference between the growth levels of pollutants in 2019 and 2020 has excluded the influence of seasonal factors. Instead of differentiating the concentration of pollutants, we first subtract the concentration of 2018 from the concentration of 2019, and subtract the concentration of 2019 from the concentration of 2020, and then differentiate the results. This difference method can exclude the influence of seasonality, because the corresponding spring and autumn of each year will be differenced.

42: Line 317: These patterns do vary depending on the pollutant, but isn’t it possible to estimate overall AQ based on all these pollutants using AQI? I assumed that this was the reason why you introduced AQI, to be able to express the overall air quality conditions based on a set of representative pollutants for Chinese cities.

RESPONSE: Thank you for your suggestions. We want to not only analyze the changes in AQI, but also briefly discuss the changes in various kinds of pollution.

43: Line 318: Why not to report the exact percentage with the exact number of cities in absolute dimension in the brackets here? Why you decided to choose a range-based reporting here?

RESPONSE: Thank you for your suggestions. The exact percentage with the exact number of cities are reported in Table 3. We want to conduct an overall analysis. There are small differences in the percentage data of different pollutants in different months. For the sake of succinct narration, only a unified range is given, otherwise the content will be too redundant.

44: Lines 323: A reference is required to prove that the epidemic has generated widespread concern. Moreover, did you mean widespread concern in a city, in China or globally? On top of that, you have mentioned a significant change of some parameter without providing a single statistical proof of significance in the sentence below.

RESPONSE: Thank you for your suggestions. We were referring to the coronavirus outbreak in China at the time. We have added references and deleted the ‘significant’. The detailed revisions related this comment are as follows:

As shown in Table 3 and Supplementary Materials, Fig. S2, the proportion of cities with improved air quality varied by pollutant. Specifically, the percentages of cities with decreasing O3 and SO2 were less than 30% based on the emissions monitored on that day instead of considering its chemical reactions. Nevertheless, the percentages of cities showing improvement in other pollutants exceeded 50%, and there was no noticeable change in the percentage of cities with decreasing AQI. This indicates that at the beginning of the COVID-19 outbreak, improvement in air quality was limited. However, the harm caused by the epidemic has generated widespread concern (Le et al., 2020; Tollefson, 2020; Forster et al., 2020), and the travel behavior of residents in many cities has not been limited. In fact, according to the data released by the National Bureau of statistics in China1, the average monthly passenger flow of railway, highway, water transportation and shipping in 2020 decreased by 39.8%, 47.1, 45.03 and 36.71 respectively compared with 2019. Beyond that, in 2020, the number of motor vehicles in China reached 372 million, and 33.28 million new motor vehicles could be registered, an increase of 1.14 million over 2019 based on the statistics of China’s Ministry of public security2. According to the platform travel data released by Didi Taxi3 in 2020, due to the epidemic, the demand for online car Hailing reached a trough in February 2020, but only four months later, the user travel demand in June has returned to the level of the same period in 2019. To avoid being infected when using shared means of transport, some residents may prefer to driving private cars instead of using public transport, resulting in increased air pollutant emissions.

45: Line 328: You used future simple tense, while speaking about 2020 which passed two years ago here.

RESPONSE: Thank you for your suggestions. We have corrected this syntax error. The detailed revisions related this comment are as follows:

As shown in Table 3 and Supplementary Materials, Fig. S2, the proportion of cities with improved air quality varied by pollutant. Specifically, the percentages of cities with decreasing O3 and SO2 were less than 30% based on the emissions monitored on that day instead of considering its chemical reactions. Nevertheless, the percentages of cities showing improvement in other pollutants exceeded 50%, and there was no noticeable change in the percentage of cities with decreasing AQI. This indicates that at the beginning of the COVID-19 outbreak, improvement in air quality was limited. However, the harm caused by the epidemic has generated widespread concern (Le et al., 2020; Tollefson, 2020; Forster et al., 2020), and the travel behavior of residents in many cities has not been limited. In fact, according to the data released by the National Bureau of statistics in China1, the average monthly passenger flow of railway, highway, water transportation and shipping in 2020 decreased by 39.8%, 47.1, 45.03 and 36.71 respectively compared with 2019. Beyond that, in 2020, the number of motor vehicles in China reached 372 million, and 33.28 million new motor vehicles could be registered, an increase of 1.14 million over 2019 based on the statistics of China’s Ministry of public security2. According to the platform travel data released by Didi Taxi3 in 2020, due to the epidemic, the demand for online car Hailing reached a trough in February 2020, but only four months later, the user travel demand in June has returned to the level of the same period in 2019. To avoid being infected when using shared means of transport, some residents may prefer to driving private cars instead of using public transport, resulting in increased air pollutant emissions.

46: Line 335: You have not presented any result above, it rather looks like a hasty adjustment of some data, not related to your study directly (like DiDi data or the data from Chinese national inventories) to the required explanation, offered by Chang et al., (2021). To give you an example, how the number of the new vehicles issued in 2020 prove that people shifted to private cars? Is this number significantly larger than the increase between 2018 and 2019, for instance. Even so the increase is significant, do you have any arguments, which can prove that the concern about epidemy, but not the growth of affluence of Chinese citizens, for instance, has caused this increase in the vehicle demand? Moreover, in Line 336 you are speaking about air pollution in general without actually adhering to AQI results from your study. Where a reader can see the estimated interplay between the concerns to be infected, the increase of use of personal vehicles/taxis and the dynamics of AQI in Chinese cities? If this analysis was limited to what has been said in lines 323-335, such arguments are rather preliminary and, as mentioned above, require further argumentation.

RESPONSE: Thank you for your suggestions. Here are economic theoretical analyses based on our results, and the results of these analyses have been confirmed by the previous literature. If the analysis of the transformation of private car use is added, the topic of the article will be unclear.

47: Line 338: Once again, what is “strict home isolation and travel control policies”. These policies have never been explained in this study, the dynamics of home isolation were neither shown, nor compared with AQI to make such conclusions.

RESPONSE: Thank you for your suggestions. Some examples of strict home isolation and travel control policies have been provided in current studies mentioned in this paper (Chang et al., 2020; Tian et al., 2020). This is not a policy assessment of pandemic lockdowns, but rather an assessment and comparison of changes in air quality before and after the pandemic. Instead of using a policy evaluation approach, we use a signal analysis approach. In addition, we use the number of confirmed cases in each city as a proxy for the severity of COVID-19 occurrence and classify cities based on these data. These data comes from Wind database.

48: Line 339. The word “significantly” is not supported by any statistical arguments.

RESPONSE: Thank you for your suggestions. We removed the unreasonable point about ‘significant’.

49: Lines 343-345. It is hard not to agree with it, but why to mention such negative aspect of your methodology rather than directly stating how you can proceed to more robust analysis?

RESPONSE: Thank you for your suggestions. That’s one of the reasons we have group discussions about cities. Due to the small number of cities with secondary outbreaks within a few months after the outbreak of the epidemic, it is difficult to conduct a robustness test on the samples due to the limited number of samples.

50: Line 345: A combination of highly-self critical statement above and the decision to analyze 12 cities in Hubei creates an impression that the analysis, performed for other Chinese cities outside Hubei was not robust, therefore undermining the last signs of credibility of the analysis in Section 3.1

RESPONSE: Thank you for your suggestions. In the months following the outbreak, these cities had limited sample sources, as we need to point out in our article.

51: Lines 352-362. The word “Significantly” is not supported by any arguments here. Moreover, “may have resulted” is a highly suggestive sentence without being supported by any arguments. If I would state that the air quality in these cities have improved by favorable meteorological conditions, what would be the main arguments, corroborating your hypothesis about home isolation policy in such discussion? The same can be applied to the phrase “may have resulted from implementation of…”

RESPONSE: Thank you for your suggestions. The main argument for the stay-at-home policy hypothesis is based on the results we provide that show air improvement in the month in which the corresponding quarantine policy was issued. We use the difference method, so we can eliminate the interference of meteorological conditions. We believe that the economic reasoning we have made is sound.

52: Line 379. Kindly explain what is “the degree of deterioration weakened”. Does it mean that the deterioration of air quality slowed down?

RESPONSE: Yes.

53: Line 382: As mentioned, the statement about “good control of pandemic” should be supported by either numerical arguments or by references to previous studied, which proved or indicated it was the case.

RESPONSE: Thank you for your suggestions. The information comes from Chinese transport authorities (https://www.mot.gov.cn/), Chinese government websites (https://www.gov.cn/) and People’s Daily (http://paper.people.com.cn/rmrb/html/2022-08/13/nbs.D110000renmrb_01.htm). These sources of information have been added. The detailed revisions related this comment are as follows:

In Supplementary Materials, Fig. S5, blue and yellow represent the improvement and deterioration of air quality, respectively. Air quality deteriorated from Jan 2020 to May 2020, corresponding to the initial outbreak of COVID-19, in most of these 24 cities, including Heyuan, Aksu, and Baise. However, the degree of deterioration in air quality in these cities weakened after Aug 2020. In addition, air quality in most cities with few confirmed COVID-19 cases has not improved during the pandemic. In fact, in these cities, because the epidemic is well controlled and the number of infected individuals is minimal, many cities have not implemented rigorous traffic control, business shutdown, or home isolation policies (Based on relevant information on Chinese government websites, https://www.gov.cn/).

54: Line 385: The word “significant” is not supported by any statistical arguments here.

RESPONSE: Thank you for your suggestions. We have deleted this word.

55: Line 389: Once again, why not to report an exact estimate instead of approximate estimate (~40%?).

RESPONSE: Thank you for your suggestions. The exact percentage with the exact number of cities are reported in Table 3. We want to conduct an overall analysis. There are small differences in the percentage data of different pollutants in different months. For the sake of succinct narration, only a unified range is given, otherwise the content will be too redundant.

56: Lines 396-397: Where a reader can get familiarized with the appearance of cases of COVID-19 you are talking about here? Is there any figure, demonstrating these dynamics?

RESPONSE: Thank you for your suggestions. The latest updates on the COVID-19 outbreak in China can be found on https://news.ifeng.com/c/special/7tPlDSzDgVk.

57: Lines 403-404: The air quality can improve and deteriorate, but not increase or decrease.

RESPONSE: Thank you for your suggestions. We have revised this point. The detailed revisions related this comment are as follows:

After the initial outbreak period, the values of the changes in the last IMF were negative in Dalian, Harbin and Urumqi, indicating that during this period, air quality improved in these cities by approximately 12.91, 6.28 and 0.36, respectively. However, air quality deteriorated in Beijing and Chengdu during this period by approximately 0.77 and 5.04, respectively. During the second outbreak of COVID-19, the values of changes in the last IMF were less than zero in Harbin (-1.07), Dalian (-1.64), Chengdu (-5.93) and Beijing (-4.05), but air quality deteriorated in Urumqi (1.98). The degree of change in the last IMF before and after the COVID-19 outbreak was reduced, indicating that the sensitivity of air quality to epidemic conditions is decreasing.

58: Lines 411-416. The language of these sentences is rather confusing from grammar and content points of view. Grammatically, why you have used future simple here (“will affect…”), did you provide any forecasts or predictions in this study? I am afraid I have not noticed that. From content point of view, what does it mean to “reduce the difficulty” (I assumed, it means “to simplify”) and most critically, “to reduce the accuracy” of the estimates? How the outbreak can even theoretically reduce the accuracy of air quality estimates and where have you previously mentioned such objective in your study? If you refer to the difficulties in lines 416-419, most commonly used air quality indices should be insensitive to the COVID outbreaks and, therefore, realistically reflect air quality or am I wrong here? Why one should take into account any differences in air quality methodology before and after the outbreak?

RESPONSE: Thank you for your suggestions. We have revised the tenses of the sentences. For ‘How the outbreak can even theoretically reduce the accuracy of air quality estimates’, we have mentioned it in our second innovation in introduction. Beyond that, there are differences in the evolution of air quality before and after the outbreak of COVID-19, which may lead to the change in the regularity of air quality data. The detailed revisions related this comment are as follows:

  1. The predictability of the evolution of AQI trends caused by COVID-19

Beyond that, previous studies involving the air quality prediction used the econometric or numerical simulation methods, and aimed to provide the specific air quality estimates for the next few days or weeks. This paper does not intend to estimate the concentration of air pollutants in the future and payed more attention on whether the outbreak of COVID-19 could affect the varying intensity of AQI. In other words, it aims to evaluate whether the outbreak of the epidemic reduce the difficulty and accuracy of the estimation of future value of air quality or not. Namely, there are differences in the evolution of air quality before and after the outbreak of COVID-19, which may lead to the change in the regularity of air quality data.

59: Line 425: There are no arguments, supporting statistical association, let alone, the

significance of this association between the air quality and home isolation measures in this paragraph. Thus, such statement is not justified.

RESPONSE: Thank you for your suggestions. We matched the two factors in terms of time. The main argument for the stay-at-home policy hypothesis is based on the results we provide that show air improvement in the month in which the corresponding quarantine policy was issued. We use the difference method, so we can eliminate the interference of meteorological conditions or other factors. We believe that the economic reasoning we have made is sound.

60: Line 437: What is the criterion of significant instability of epidemic (or AQI), so a reader can get familiarized with this phenomenon?

RESPONSE: Thank you for your suggestions. Recursive quantitative analysis of various indicators can solve this problem, and detailed elaboration is reported in the method. For example, LAM: Laminarity is another important indicator that can be used to measure the stability of an AQI system.

61: Lines 470-471: Please indicate where we can evaluate the relationship between consumption behavior, increase of energy and the complicated pattern of AQI?

RESPONSE: That’s the further reasonable economic analysis we make from the results, and we have deleted these subjective analysis. The detailed revisions related this comment are as follows:

In the 24 cities with few COVID-19 cases, air quality improved overall because of restrictive epidemic control policy or environmental control measures (Supplementary Materials, Fig. S10). Areas of uncertainty were observed in the middle of the sample, indicating that retaliatory consumption led to the increasing fluctuation of the impact of the epidemic on air quality (Supplementary Materials, Fig. S11). However, in the stable stage of the epidemic, the long-term change trend in AQI gradually stabilized, as reflected by the blue area in the lower right corner in Fig. 4, indicating that residents’ retaliatory consumption only lasted for a few months before the long-term trend in air quality became regionally stable. RQA indicators for cities where the epidemic situation was not serious is illustrated in Fig. 4.

62: Line 480: What is “serious” difference in RQA? Is it a quantitative criterion or arbitrarily defined description? What would be “not serious” difference in RQA, for instance?

RESPONSE: That means cities with fewer confirmed cases, which has been mentioned in the city classification criteria.

63: Lines 489-503. This description is entirely suggestive. In particular, “may have led to deterioration” is not supported by any numerical arguments, thereby critically undermining your conclusion. Once again, the statement about the efficiency of control measures is not supported either. No academic reference, nor the actual results of your study demonstrate this efficiency. Moreover, there are statements about not quantified phenomena such as “panic” of citizens or the magnitude of their “daily activities”. Which parameters were used to judge about these behavioral patterns of citizens?

RESPONSE: That’s the further reasonable economic analysis we make from the results, and we have deleted these subjective analysis. The detailed revisions related this comment are as follows:

4.3. Five cities experiencing secondary outbreaks

       To investigate the evolution of air quality trends resulting from the COVID-19 pandemic in cities with secondary outbreaks, temporal trends in AQI were analyzed and recurrence plots of the evolution of trends in AQI in these cities were constructed (shown in Supplementary Materials, Fig. S12 and Fig. S13).

In the cities that experienced secondary outbreaks, the air quality showed a pattern of initial improvement and subsequent deterioration (Supplementary Materials, Fig. S12). It could be deduced that in the first stage of the epidemic, the strict travel restriction policy inhibited the residents’ movement and reduced the residents’ consumption levels. Hence, the emissions of various air pollutants showed downward trends. Therefore, although the trend of deteriorating air quality was reduced, it still existed. The long-term change in AQI was stable in early 2020, corresponding to the outbreak stage, indicating that during the outbreak stage, the deterioration of air quality was slow (Supplementary Materials, Fig. S13). However, the first wave of the epidemic was stable. Even in the second outbreak, the fluctuation of the air quality trend was not affected. The RQA indicators for the recurrence plot of the evolution of trends in AQI are reported in Fig. 5.

64: Line 519. You once again mention the word “significant/significantly”, but I have not noticed any arguments, demonstrating or refuting significance of any analyzed parameters in this study.

RESPONSE: Thank you for your suggestions. We have deleted this word.

65: Line 521: I do not generally understand how the change in something can decline. Do you mean that the variability of CO and O3 weakened? If yes, such dynamics can be judged by referring to statistical terms of variability such as variance and standard deviation.

RESPONSE: It means a decrease in concentration, and we’ve revised this point. The detailed revisions related this comment are as follows:

  1. Conclusions

This paper not only investigated the changes in six air pollutants before and after COVID-19 outbreaks in 328 cities in China by extracting the differences in the growth trends between 2018-2019 and 2019-2020 but also applied recurrence plots and recursive quantitative analysis to compare the changes in predictability of AQI, which may be affected by the COVID-19 pandemic. It was confirmed that due to the initial outbreak and restrictive mitigation policies, the concentrations of different pollutants changed, and the predictability of air quality was also affected.

In the three to four months after the outbreak, there is an increasing proportion of cities with reduced O3 concentrations. We hold that in the post-epidemic era, the government should formulate corresponding control policies for different air pollutants.Changes in trends in the impacts of the COVID-19 pandemic on AQI were found to show obvious differences among different classes of cities. Compared with the cities that did not experience severe epidemic conditions, air quality has noticeably improved in the cities in Hubei, and for the cities with secondary outbreaks, air quality first improved and then deteriorated. During the second stable period, the sensitivity of air quality to epidemic conditions decreased. Therefore, it is suggested that in cities with severe epidemic conditions, it is necessary to strengthen the management of residents’ travel, maintain the effect of improving air quality, and ensure the smooth operation of the economy.

66: Lines 522-532. First of all, even in the pre-pandemic worlds, there were myriads of air quality studies, which demonstrated that each pollutant should be controlled differently depending on the harm it can inflict either on environment or on human health. Why this paradigm should be emphasized in this study once again only after COVID-19? Can you elaborate this aspect here? Moreover, this paragraph does not shed the light on the predictability of urban air quality (as stated in the title). Meanwhile, the patterns about air quality

RESPONSE: Thank you for your suggestions. In the post-pandemic era, various air pollutants may change differently and be affected differently by the epidemic than before, so cities with different epidemic status need to be treated separately. The predictability of urban air quality has been provided in the end of conclusion.

67: Line 525. There were no statistical arguments or causal arguments, demonstrating that particularly CO and O3 were driven by energy consumption. What if CO was driven by seasonal changes or the change in the efficiency of vehicular combustion? All these hypotheses have right to exist and only those, supported by numerical arguments, should be included in a peer-review study like yours. Furthermore, is not the finding about the insignificant change of NO2, PM2.5 and PM10 in 328 cities you analyzed in not important enough finding to be reported in the abstract? Does it mean that the inclusion of aerosols an NO2 into your analysis did not bring any palpable benefits in providing new insights on AQ change in COVID-19-affected cities?

RESPONSE: Thank you for your suggestions. Factors such as seasonal variations have been eliminated by differential methods. For the impact of energy consumption, that's the further economic inference we make from the results.

68: Line 533: Several questions arise here. What is “social economy”? Is it a common term for Chinese cities you analyzed or it is a rather improper formulation here? Moreover, doesn’t your suggestion imply that transportation should not be improved, AQ should remain as bad as it is and smooth operation of “social economy” are not required in the cities, not affected by COVID-19 severely?

RESPONSE: Thank you for your suggestions. We have deleted ‘social’. We emphasize the need to focus on air quality in the outbreak cities, not against other types of urban air control measures. In addition, we do not have data for cities outside the sample, so we cannot make arbitrary recommendations.

69: Line 536: You have not explained the difference between the cities with effective epidemic control and the cities with not effective epidemic control in this paper. Hence, this conclusion is based on assumption about efficiency of anti-epidemic measures in some cities of China, but is not supported by results.

RESPONSE: We have explained the difference between the cities with effective epidemic control and the cities with not effective epidemic control in Section 4.

70: Lines 540-542. Let’s imagine the situation where a population of some city is considerably increased, thereby increasing the amount of vehicles, which, in turn result in deterioration of AQ. Alternatively, some huge industry is being developed in a city and corresponding emissions are being generated as a result of new industrial production. Is it true that for such cities, seemingly unaffected by COVID-19, one can forecast future AQ based on past AQ indices, before significant increase of cars/industrial production occurred? How accurate will be these predictions, not accounting such important sources of pollution? Kindly note that this is a rhetoric question, debunking your suggestion about the ability to predict future AQ of the city just based on historical AQ in this city.

RESPONSE: It is with this in mind that we adopt the difference method. All sorts of factors cannot necessarily be included in the model, and we have explained it in Section 2. The detailed revisions related this comment are as follows:

  1. Methods and Data

       The empirical study is divided into two parts. First, in Part 1 of Fig. 1, the second-order difference method was used to calculate the differences in air pollutant changes, and then the EEMD method was adopted to obtain and analyze the long-term trends in these differences. For the feasibility of empirical research, it is assumed that for each city, the local government's annual efforts to control air pollution are relatively close, and the difference can eliminate policy human factors to a certain extent. In fact, this paper first uses the difference method to calculate the growth of air pollutants before and after the outbreak of the epidemic, and then uses the signal decomposition method to decompose its components. The difference method is not a DID model, which does not involve explanatory variables and explained variables. Second, a recursive graph was used to allow visualization of trajectory periodicity through phase space. By drawing a recursive graph, we can study some aspects of the m-dimensional phase space trajectory of the AQI through a two-dimensional representation, which reflects the predictability of air quality. Then, recursive quantitative analysis was used for the quantitative analysis of changes in predictability. The specific steps are shown in Fig. 1.

71: Line 544 “More comprehensive detection”. What does this term mean? Why the current detection is not comprehensive enough for allowing residents to make rational travel plans? As far as I know, in most Chinese cities, there are systems for reporting AQ in the city based on either measurements or AQ modelling. Why such systems are imperfect? I have not noticed any evaluations of these systems in your study.

RESPONSE: Thank you for your suggestions. We have deleted this conclusion. The detailed revisions related this comment are as follows:

In addition, according to the results of RPs, in the stable period of the epidemic, there was instability in the cities with severe COVID-19 outbreaks, indicating that the impact of COVID-19 is challenging to predict. Nevertheless, in cities with fairly effective epidemic control, AQI trends stabilized more rapidly. According to the RQA results, the predictability and stability of the AQI system were improved in cities where the epidemic was not serious, but in cities with severe COVID-19 outbreaks in Hubei, the opposite was true. This implies that in cities where the epidemic was not serious, the government can estimate the future AQI index based on past air quality to formulate relevant policies to effectively improve regional air quality in the post-epidemic era.

72: Line 545: You are speaking about the gases here, but you also supposed to analyze PM2.5 and PM10 (if I judge by your abstract). Thus, why your suggestion mentions only gaseous pollutants here?

RESPONSE: PM10 refers to particulate matter with aerodynamic equivalent diameter less than or equal to 10 microns in ambient air, also known as inhalable particles Content. PM10 coarse particles in the air can slowly settle, and easy to be blocked in the nasal cavity and mouth, part of the sputum And so out of the body, so relatively little harm to human health. PM2.5 refers to particulate matter with aerodynamic equivalent diameter less than or equal to 2.5 microns in ambient air, also known as fine particulate matter. It can be suspended in the air for a long time, and the higher its concentration in the air, the more serious the air pollution. We use air pollutants to refer to them generically.

73: Lines 550-551. If your method cannot take into account the transportation (horizontal and vertical) of pollutants depending on meteorological conditions, how could you ensure that the decrease of some pollutants was related to COVID-19 or related measures, not to meteorological conditions? What has constrained you from applying the spatial weight matrix in this study to nail down your conclusions more effectively?

RESPONSE: Thank you for your suggestions. Considering that the cities studied in this paper are not completely bordering on each other, an appropriate weight matrix cannot be found for research, and the spatial econometric model also needs appropriate independent variables. Of course, this is what we plan to do later.

74: Line 555: “too difficult to analyze meteorological factors…”. Please note that in peer review, difficulties in studying, addressing some phenomena do not imply that you one can just damp such recommendation into the limitations of the study. Rather, there should be a solid explanation what has constrained you from using such difficult, but probably, fruitful approach, thereby ensuring that meteorological conditions were accounted. In the worst case, you could apply meteorological analysis to 12 cities in Hubei. For instance, meteorological conditions were taken into account precisely for 12 cities in this study (https://link.springer.com/article/10.1007/s40201-020-00564-y) about air quality in China. Moreover, 221 Chinese cities were analyzed in this study (https://www.hindawi.com/journals/complexity/2020/6829142/) where air quality conditions were compared with meteorological conditions. In other words, either use this approach or find more solid justification for not using it rather than stating that it is difficult, please.

RESPONSE: Thank you for your suggestions. We apologize for not making this point clear. Because recursive quantitative analysis is not an econometric regression model, the relationship between multiple variables cannot be considered at the same time. We have revised this point and added more explanation. The detailed revisions related this comment are as follows:

  1. Limitations and future research directions

We hold that this study has two main deficiencies. First, some pollutants are secondarily produced in the atmosphere, how to distinguish them from the evolution of primary pollutants based on COVID-19 condition could not be solved. Second, the pollutants emitted in a city are transported to other regions within a few hours and days, which corresponds to the spatial effect of air pollutants. The EEMD method could not effectively depict this spatial effect. Therefore, in the subsequent work, we intend to further improved the EEMD method and introduce the spatial weight matrix to investigate this spatial effect. Beyond that, recursive quantitative analysis is not an econometric regression model, the relationship between multiple variables cannot be considered at the same time, such as the meteorological factors. Therefore, in the follow-up empirical analysis, we intend to construct a flexible regression model to depict its effects.

We find that the reviewer’s comments have been quite helpful in improving the paper, and we have revised our paper point by point. Once again, thank you very much for your comments and suggestions!

Best Regards!

Yours sincerely!

1 https://data.stats.gov.cn/easyquery.htm?cn=A01

2 https://www.mps.gov.cn/

3 https://www.didiglobal.com/

1 https://data.stats.gov.cn/easyquery.htm?cn=A01

2 https://www.mps.gov.cn/

3 https://www.didiglobal.com/

Reviewer 4 Report

The presentation quality of this manuscript in not satisfactory. This paper investigated the changes in air pollutants before and after COVID-19 and it is a well-known fact that if there is no economic and traffic activity in the city, the pollution will be less. Therefore, a more appropriate title would be:  Changes in air quality during the work period of COVID-19. This manuscript is more of an overview article. The manuscrpit did not explain why all data were presented as mean values. The manuscript does not contain hypotheses or the goal of the research, and the results showed that the method is not suitable for this type of research due to shortcomings the can significantly affect the results.

Author Response

We would like to thank the editor and reviewers for their careful and thorough reading of this manuscript and for the thoughtful comments and constructive suggestions, which have helped us improve the manuscript and reach a better scientific level. All of the comments are valuable and have been very helpful during the revision process and have provided important guiding significance to our research. We have carefully revised the manuscript entitled “Improvement and predictability of urban air quality under different stages of the COVID-19 pandemic” (ijerph-1849181) according to the editor and reviewers’ comments. The responses to the reviewers’ comments and the revised portions are reported below each question and are marked in blue and red, respectively. We tried our best to improve the manuscript by making the suggested changes. These changes did not influence the content and framework of the paper. We appreciate the editor’s and reviewers’ thoughtful work earnestly and hope that the corrections will meet with your approval. Thank you for the time and effort that was expended on this paper. The responses to the reviewers’ and editor’s comments are as follows:

Response to the Comments of Reviewer 4

Reviewer’s comments:

1: The presentation quality of this manuscript in not satisfactory. This paper investigated the changes in air pollutants before and after COVID-19 and it is a well-known fact that if there is no economic and traffic activity in the city, the pollution will be less. Therefore, a more appropriate title would be:  Changes in air quality during the work period of COVID-19. This manuscript is more of an overview article. The manuscrpit did not explain why all data were presented as mean values. The manuscript does not contain hypotheses or the goal of the research, and the results showed that the method is not suitable for this type of research due to shortcomings the can significantly affect the results.

RESPONSE: Thank you for your suggestions. The title has been revised: Changes in air quality during the work period of COVID-19. We have also revised Abstract, Introduction and Limitations and future research directions to make our work and innovation more clear. The main purpose of our use of mean values is to unify the analysis results, and we compare different types of cities between groups. Beyond that, We have changed the statement about significance in this paper. The detailed revisions related this comment are as follows:

Abstract: This paper investigates the various impact of COVID-19 began in January 2020 on air quality in different types of cities. We analyze the different degrees of improvement of concentrations of six air pollutants (PM2.5, PM10, SO2, NO2, CO and O3) obtained from the Ministry of Ecology and Environment, PRC in different types of Chinese cities from 2018 to 2020 with difference method (the trend of pollutants in different years was differenced) and ensemble empirical mode decomposition (EEMD), and then adopt the recursive plots (RPs) and recursive quantitative analysis (RQA) for the first time to discuss whether air quality is more difficult to predict during the outbreak. The empirical results indicate that: (1) After the initial outbreak, only the emissions of NO2, CO and PM2.5 declined for the first 1-3 months, and during the fourth to fifth months the emissions of six air pollutants were elevated in most cities; (2) For the cities with serious epidemic situations in the Hubei province (China), the air quality is improved, but for the cities experiencing a second outbreak, the air quality was first enhanced and then deteriorated, and the sensitivity of air quality to COVID-19 re-outbreak is decreasing. For Dalian, Harbin and Urumqi, AQI is improved by approximately 12.91, 6.28 and 0.36, respectively. (3) In comparison, the predictability of AQI has declined in cities with serious epidemic situations in Hubei, but AQI achieves a stable state sooner in cities with mild epidemic. The conclusions may facilitate analysis of differences in air quality evolution characteristics and fluctuations before and after outbreaks from a quantitative perspective.

  1. Introduction

(…)

Second, whether the COVID-19 pandemic could affect the predictability of urban air quality in different cities has not to our knowledge been addressed in previous literature. This paper adopts recurrence plots (RPs) and recurrence quantification analysis (RQA) to compare the characteristics of fluctuation in AQI before and after outbreaks, which could be used to describe the effect of the COVID-19 pandemic on the predictability of urban air quality. Specifically, in this paper we first selected the three classes of cities mentioned before: cities experiencing severe epidemic, cities with mild epidemic conditions, and cities experiencing secondary outbreaks. The recurrence plots of the mean values of AQI in these cities are reported to discuss the characteristics of fluctuation in AQI in these different kinds of cities. Second, the recurrence plots and recurrence quantification analysis for mean values of AQI in 2018, 2019 and 2020 are listed for comparison. This may help to accurate prediction of urban air quality, facilitate analysis of differences in air quality evolution characteristics and fluctuations before and after outbreaks from a quantitative perspective and formulate air control measures in advance.

  1. Limitations and future research directions

We hold that this study has two main deficiencies. First, some pollutants are secondarily produced in the atmosphere, how to distinguish them from the evolution of primary pollutants based on COVID-19 condition could not be solved. Second, the pollutants emitted in a city are transported to other regions within a few hours and days, which corresponds to the spatial effect of air pollutants. The EEMD method could not effectively depict this spatial effect. Therefore, in the subsequent work, we intend to further improved the EEMD method and introduce the spatial weight matrix to investigate this spatial effect. Beyond that, recursive quantitative analysis is not an econometric regression model, the relationship between multiple variables cannot be considered at the same time, such as the meteorological factors. Therefore, in the follow-up empirical analysis, we intend to construct a flexible regression model to depict its effects.

We find that the reviewer’s comments have been quite helpful in improving the paper, and we have revised our paper point by point. Once again, thank you very much for your comments and suggestions!

Best Regards!

Yours sincerely!

Round 2

Reviewer 1 Report

The authors have made some changes to the paper according to the commments, but the current quality of the paper is not up to the publication standard. 

Reviewer 2 Report

This paper investigates the various impact of COVID-19 began in January 2020 on air quality in different types of cities. But I think there are still many problems in this article.

Comment 1:

The selection of the samples is still confusing, and more explanation is needed to tell us why those cities in the manuscript were chosen instead of other cities with the same characteristics. For instance, why other 5 cities in Hubei can’t be selected and why other cities with few confirmed cases can’t be selected?

Comment 2: 

More algorithms like CEEMD, CEEMDAN, IEEMDAN and other methods should be used for comparison. If the extracted long-term trends are similar, the findings are more robust.

Comment 3: 

The selection of last IMF needs to be explained, why not select more components? Maybe reconstruction with more low-frequency components can better reflect the long-term trend of the series

Comment 4: 

The paper still doesn't effectively explain why the stability of the system can characterize the predictability of AQI and why RQA indicators are more suitable than other indicators such as Permutation Etropy, Sample Entropy and Multiscale Entropy in measuring the stability of time series?

Comment 5: 

To reiterate, the research can't come to the conclusion by taking small samples in 3 years. You may just pick a sample that matches your expected conclusion. Moreover, the differences between the mean values of indicators may be caused by random factors. So the research needs to make the sample size larger and use statistical hypothesis-testing like Student’s t test to determine whether there are statistical differences. The above problems persist throughout the chapter 4.

Comment 6: 

It is puzzling that the Figure 1 is not completely modified yet. The bar graph in Figure 4 is still too dense to be legible and can be divided into several more lines. And the title of Figure 5 should be revised to be consistent to its content. 

Comment 7: 

The interpretation of Figure 5 is still wrong. The trend in the mean values of TT is not decreased. The trend in the mean values of LMAX is decreased.

Comment 8: 

The mean value of last IMF may represent long-term trend of other factors like newly proposed environmental protection policies in 2020, geopolitics and economic downturn. How to exclude the interference of other factors?